# Using a combined power law and log-normal distribution model to simulate particle formation and growth in a mobile aerosol chamber

M. Olin, T. Anttila, and M. Dal Maso

Aerosol Physics Laboratory, Department of Physics, Tampere University of Technology, P.O. Box 692, 33101 Tampere, Finland

*Correspondence to:* M. Olin (miska.olin@tut.fi)

**Abstract.** We present the combined power law and log-normal distribution (PL+LN) model, a computationally efficient model to be used in simulations where the particle size distribution cannot be accurately represented by log-normal distributions, such as in simulations involving the initial steps of aerosol formation, where new particle formation and growth occur simultaneously, or in the case of inverse modelling. The model was evaluated against highly accurate sectional models using input
parameter values that reflect conditions typical to particle formation occurring in the atmosphere and in vehicle exhaust, and tested in the simulation of a particle formation event performed in a mobile aerosol chamber at Mäkelänkatu street canyon measurement site in Helsinki, Finland. The number, surface area, and mass concentrations in the chamber simulation were conserved with the relative errors lower than 2 % using the PL+LN model, whereas a moment-based log-normal model and sectional models with the same computing time as with the PL+LN model caused relative errors up to 17 % and 79 %, respec-
tively.

## 1  Introduction

Particle size distribution is the most important characteristic of nanoparticles, as it controls their deposition to the human respiratory system, their behavior in the atmosphere, and the properties of engineered nanoparticles. The rates of several aerosol processes, such as condensation, coagulation, and deposition, are affected by particle size; thus, the particle size distribution
controls also the evolution of the aerosol. While the rates of the aerosol processes depend on the particle size, different particles within a particle size mode have different rates of aerosol processes and, thus, they evolve with different rates. This causes also the shape of the size distribution to evolve. Because particle size distributions usually contain particles with the diameters of several orders of magnitude rather than being monodisperse, i.e., equally sized, an accurate representation of aerosol properties and evolution requires that particle sizes are expressed as distributions. Due to a high count of particles with different sizes,
shapes, and compositions within a volume of interest, computational costs to model them separately are extremely too high. Therefore, aerosol models typically model one or more parameter of the size distribution, such as particle number or mass concentration of the total particle size range or of several size ranges separately. Simplifications made for size distributions in aerosol models cause unrealistic shapes for the distributions.

Methods that model a particle size distribution the most realistically are sectional methods, in which the size distribution is split into separate size sections. The accuracy of a sectional model can be controlled by the number of the size sections. Increasing the number of sections increases accuracy, but the computational cost is also increased. In multidimensional simulations, such as in Computational Fluid Dynamics (CFD) and in climate simulations, computational efficiency is a key property of the model. Simulations involving inverse modelling (Verheggen and Mozurkewich, 2006), where the values of model input parameters (e.g., new particle formation rate or condensational growth rate) are varied systematically to find out the values that most exactly produce the measured results, may suffer from long computing times even in one-dimensional cases.

Sectional methods vary depending on the conserved property of the aerosol. Only a single property, e.g., particle number, particle surface area, or particle mass concentration, can be conserved in the simulation but other properties will suffer from numerical diffusion, which is seen as the overestimation of the non-conserved properties (Wu and Biswas, 1998). Less numerical diffusion can be obtained, e.g., by using a moving-center fixed-sectional method, in which size sections have fixed boundaries but the centers of the sections are allowed to vary so that number and mass concentrations are conserved better(Jacobson, 1997). However, implementing the moving-center fixed-sectional method in Eulerian simulation, such as in CFD simulation, with simultaneous new particle formation, condensation, coagulation, and transportation is challenging due to discontinuous behavior of the section variables (all particles of a section are transferred to an adjacent section when the center of a section exceeds a section boundary during growth), computationally time-consuming due to the transfer of the particles between the sections, and memory-consuming due to the requirement of storing also the center values of the sections. Wang and Zhang (2012) have modelled simultaneous new particle formation and growth within diesel exhaust plumes using the moving-center fixed-sectional method in three-dimensional CFD simulation and have obtained promising results for particle size distributions compared to the measured distributions with only 8 size sections in a particle diameter decade. However, they did not report comparison between their model and any highly accurate aerosol model; thus, the effect of numerical diffusion to their results is unknown. Another method to decrease numerical diffusion is the TwO-Moment Aerosol Sectional (TOMAS) model, in which both the number and the mass concentrations are stored for all size sections (Adams and Seinfeld, 2002). The TOMAS model provides conservation for both the number and the mass concentrations of the total distribution, but the memory consumption in multidimensional simulations can be too high due to a high number of variables to be stored in every computational cell.

Other approaches to model the particle size distribution are methods based on the moments of the distribution (Whitby and McMurry, 1997), which are both computationally efficient (Mitrakos et al., 2007) and have continuous behavior of the variables. The number of the conserved properties of the aerosol is controlled by the number of the modelled moments; e.g., conserving number, surface area, and mass concentrations can be obtained by modelling the corresponding three moments. The number of the variables being stored during the simulation is the number of the modelled moments, which is significantly less compared to sectional methods, in which the number of the variables can be several hundreds. The major drawback in the methods based on moments is that the size distribution needs to be presented with a pre-defined function, unless the quadrature-method of moments (QMOM, McGraw (1997)) is used. QMOM provides accurate results (Barrett and Webb, 1998) but the reconstruction of the distribution parameters from the moments is not unique (Mitrakos et al., 2007). The typical choice for the size distribution function is the log-normal distribution or the combination of several log-normal distributions. They correspond

well with many laboratory aerosols and aged aerosols, but during the initial steps of the formation and growth of aerosol the size distribution can differ significantly. For example, Tammet and Kulmala (2014) recommend two-power law for the size distribution of atmospheric aerosols measured at least in Northern Europe. Two-power law distribution has four parameters, which implies that four moments are required for the reconstruction of the parameters from the moments, if the distribution is
modelled using the moment method. However, there is no analytical solution for the system of equations of the two-power law approach, and solving the system of equations with four variables numerically is computationally very expensive.

The general dynamic equation (GDE) for the number concentration of a size section $j$, with new particle formation and condensational growth without any other aerosol processes, is (Seinfeld and Pandis, 2006)

$$\frac{\mathrm{d}N_j}{\mathrm{d}t} = \begin{cases} J(t) - \frac{g(t,D_\mathrm{p})}{\Delta D_j}N_j, & j = 1 \\ \frac{g(t,D_\mathrm{p})}{\Delta D_{j-1}}N_{j-1} - \frac{g(t,D_\mathrm{p})}{\Delta D_j}N_j, & j > 1 \end{cases}, \tag{1}$$

where $N_j$ and $\Delta D_j$ are the number concentration and the diameter width of the size section, respectively. $J(t)$ is new particle formation rate as a function of time $t$, and $g(t, D_\mathrm{p})$ is condensational growth rate $\frac{\mathrm{d}D_\mathrm{p}}{\mathrm{d}t}$, where $D_\mathrm{p}$ is the particle diameter. In the case of simultaneous new particle formation and condensation with time- and size-independent rates, the analytical solution for the GDE provides the particle size distribution

$$\frac{\mathrm{d}N}{\mathrm{d}\ln D_\mathrm{p}} = \begin{cases} \frac{J}{g}D_\mathrm{p}, & D_1 \leq D_\mathrm{p} \leq D_2 \\ 0, & \text{otherwise} \end{cases}, \tag{2}$$

where $D_1$ is the diameter of the newly formed particle (assumed constant) and $D_2$ is the largest diameter. Equation (2) is in the form of a power law where the power of $D_\mathrm{p}$ is unity. In a realistic particle formation process, $J(t)$ and $g(t, D_\mathrm{p})$ do not remain constants and other aerosol processes affect also; thus, the power of $D_\mathrm{p}$ can differ and log-normal features will appear in the distribution. Here, we present a method to express the particle size distribution as a combination of a power law and a log-normal distribution. This moment-based combined power law and log-normal distribution model was evaluated against
highly accurate sectional models using theoretical test cases and a real-world case, which represents a simulation of a particle formation event occurred in a mobile aerosol chamber.

## 2 Model description

The combined power law (PL) and log-normal (LN) distribution model (PL+LN) is based on the sum of these distributions. The PL distribution handles the formation and the initial growth of new particles; the LN distribution represents the log-normal
shape of the distribution and it is formed by coagulation and condensation from the PL distribution.

## 2.1 Particle size distributions

### 2.1.1 Power law distribution

The formulation of the PL distribution originates from Eq. (2), where the power of $D_\mathrm{p}$ is allowed to vary,

$$\frac{\mathrm{d}N}{\mathrm{d}\ln D_\mathrm{p}}\bigg|_\mathrm{PL} = \begin{cases} \frac{N_\mathrm{PL}\alpha}{D_2^\alpha - D_1^\alpha} D_\mathrm{p}^\alpha, & D_1 \leq D_\mathrm{p} \leq D_2, \alpha \neq 0 \\[2mm] \frac{N_\mathrm{PL}}{\ln(D_2/D_1)}, & D_1 \leq D_\mathrm{p} \leq D_2, \alpha = 0, \\[2mm] 0, & \text{otherwise} \end{cases} \tag{3}$$

where $N_\mathrm{PL}$ is the total particle number concentration, $\alpha$ is the slope parameter, $D_1$ is the smallest diameter, and $D_2$ is the largest diameter of the PL distribution. In this form, the PL distribution has four parameters, which leads to numerical challenges for the reconstruction of the distribution parameters from four moments. Nevertheless, by fixing one parameter, only three moments are required to be modelled and the reconstruction will simplify. Here, the value of $D_1$ is fixed to the diameter of a newly formed particle, which is also physically sensible because that value is not expected to vary significantly; in atmospheric particle formation, the value is about $1.5 \pm 0.3\,\mathrm{nm}$ (Kulmala et al., 2013).

Three moments required in the modelling of the PL distribution with parameters $N_\mathrm{PL}$, $\alpha$, and $D_2$ are, in this article, number, $N_\mathrm{PL}$, surface area, $S_\mathrm{PL}$, and mass, $M_\mathrm{PL}$, concentrations,

$$N_\mathrm{PL} = N_\mathrm{PL} \tag{4}$$

$$S_\mathrm{PL} = \int_{-\infty}^{\infty} s_\mathrm{p} \frac{\mathrm{d}N}{\mathrm{d}\ln D_\mathrm{p}}\bigg|_\mathrm{PL} \mathrm{d}\ln D_\mathrm{p}$$

$$= s_1 N_\mathrm{PL} \frac{\alpha}{\alpha+2} \frac{\left(\frac{D_2}{D_1}\right)^{\alpha+2} - 1}{\left(\frac{D_2}{D_1}\right)^{\alpha} - 1} \tag{5}$$

$$M_\mathrm{PL} = \int_{-\infty}^{\infty} m_\mathrm{p} \frac{\mathrm{d}N}{\mathrm{d}\ln D_\mathrm{p}}\bigg|_\mathrm{PL} \mathrm{d}\ln D_\mathrm{p}$$

$$= m_1 N_\mathrm{PL} \frac{\alpha}{\alpha+3} \frac{\left(\frac{D_2}{D_1}\right)^{\alpha+3} - 1}{\left(\frac{D_2}{D_1}\right)^{\alpha} - 1}, \tag{6}$$

where $s_\mathrm{p}$ and $m_\mathrm{p}$ are the surface area and the mass of a particle, respectively, and $s_1$ and $m_1$ are the surface area and the mass of a newly formed particle, respectively. All particles are assumed to be spherical. Equations (5) and (6) have singularities at $\alpha$ values of -3, -2, and 0. In those cases, the equations have different formulations, and from now on, the singularity equations are not shown here due to the fact that $\alpha$ will never equal a singularity value precisely in a simulation. To model the composition of particles can be done by separating the mass concentration to different components using the assumption that the particles are internally mixed, i.e., the composition does not vary with particle diameter. Modelling of the particle composition is, however, outside of the scope of this article.

The reconstruction of the distribution parameters from the moments $N_{\text{PL}}$, $S_{\text{PL}}$, and $M_{\text{PL}}$ is performed as follows. The zeroth moment $N_{\text{PL}}$ is already one of the distribution parameters, but $S_{\text{PL}}$ and $M_{\text{PL}}$ are not. The latter are converted to the system of equations of two unknown variables $\alpha$ and $d = D_2/D_1$

$$
\begin{cases}
\dfrac{M_{\text{PL}}}{N_{\text{PL}}} \dfrac{1}{m_1} = \left(\dfrac{\alpha}{\alpha+3}\right)\left(\dfrac{d^{\alpha+3}-1}{d^{\alpha}-1}\right) \\[2ex]
\dfrac{M_{\text{PL}}}{S_{\text{PL}}} \dfrac{s_1}{m_1} = \left(\dfrac{\alpha+2}{\alpha+3}\right)\left(\dfrac{d^{\alpha+3}-1}{d^{\alpha+2}-1}\right)
\end{cases}.
\tag{7}
$$

However, there is no analytical solution for this system of equations, but solving two variables numerically is sufficiently fast for this purpose. A pre-calculated interpolation table is used in the numerical solution, with which a more rapid calculation is obtained. The interpolation table increases the memory cost of the model, but as the table is unique (independent on temporal or spatial coordinate) it needs to be stored in one memory location only.

### 2.1.2 Log-normal distribution

The LN distribution is expressed by the equation

$$
\left.\frac{\mathrm{d}N}{\mathrm{d}\ln D_{\text{p}}}\right|_{\text{LN}} = \frac{N_{\text{LN}}}{\sqrt{2\pi}\ln\sigma}\exp\left[-\frac{\ln^2\left(D_{\text{p}}/D_{\text{g}}\right)}{2\ln^2\sigma}\right],
\tag{8}
$$

where $N_{\text{LN}}$ is the total particle number concentration, $\sigma$ the geometric standard deviation, and $D_{\text{g}}$ the geometric mean diameter of the LN distribution. The LN distribution is also modelled as three moments, $N_{\text{LN}}$, $S_{\text{LN}}$, and $M_{\text{LN}}$. Following the method of Whitby and McMurry (1997), the reconstruction of the distribution parameters from the moments can be performed using the
equations

$$
N_{\text{LN}} = N_{\text{LN}}
\tag{9}
$$
$$
D_{\text{g}} = 6^{-2/3}\pi^{-5/6}\rho^{2/3}N_{\text{LN}}^{-5/6}S_{\text{LN}}^{3/2}M_{\text{LN}}^{-2/3}
\tag{10}
$$
$$
\ln^2\sigma = \ln\left(6^{2/3}\pi^{1/3}\rho^{-2/3}N_{\text{LN}}^{1/3}S_{\text{LN}}^{-1}M_{\text{LN}}^{2/3}\right),
\tag{11}
$$

where $\rho$ is the particle density.

### 2.1.3 Connection between the distributions

The combined particle distribution is modelled as the superposition of the PL and the LN distributions

$$
\left.\frac{\mathrm{d}N}{\mathrm{d}\ln D_{\text{p}}}\right|_{\text{PL+LN}} = \left.\frac{\mathrm{d}N}{\mathrm{d}\ln D_{\text{p}}}\right|_{\text{PL}} + \left.\frac{\mathrm{d}N}{\mathrm{d}\ln D_{\text{p}}}\right|_{\text{LN}}.
\tag{12}
$$

Figure 1 presents examples of the PL+LN distribution. PL distributions with different values of $\alpha$ are shown in the left pane; $N = 10^6\,\text{cm}^{-3}$, $D_1 = 1.5\,\text{nm}$, and $D_2 = 5\,\text{nm}$ are equal in all four distributions. The right pane shows the PL distribution with
values $N = 10^6\,\text{cm}^{-3}$, $\alpha = 1$, $D_1 = 1.5\,\text{nm}$, and $D_2 = 3\,\text{nm}$, the LN distribution with values $N = 5 \times 10^5\,\text{cm}^{-3}$, $D_{\text{g}} = 4\,\text{nm}$, and $\sigma = 1.1$, and the combination of them.

A schematic presentation of the connections between the distributions is shown in Fig. 2. Particles in the PL distribution, formed by new particle formation and grown by condensation and coagulation (Fig. 3), are transferred to the LN distribution through three intermodal processes: coagulational transfer, intermodal coagulation, and condensational transfer. The coagulational transfer is accounted by intramodal coagulation, i.e., self-coagulation, which is basically an intramodal process, but in this model it is used to initiate the LN distribution by transferring the coalesced resultant particles larger than $D_2$ to the LN distribution. The coagulational transfer is described in more detail in Sect. 2.2.5. After the LN distribution is initiated, particles of the both distributions begin to collide intermodally (the intermodal coagulation). In that case, the resultant particles are always assigned to the LN distribution, which is thought to consist of larger particles than the PL distribution.

The coagulational transfer remains the only process initiating the formation of the LN distribution if the condensational transfer is neglected. Therefore, in the case of low particle number concentration, i.e., low intramodal coagulation rate, the formation rate of the LN distribution is slow; thus, the combined distribution would be mainly in a power law form. However, in realistic particle formation events, log-normal features in the size distribution are widely observed (Hinds, 1999). This is due to the facts that the aerosol processes have normally time- and size-dependent rates and that the particles can be multicomponent, and due to the intramodal coagulation that eventually results in self-preserving log-normal distribution (Friedlander, 2000). The model described here connects the formation of the LN distribution with the intramodal coagulation only. Therefore, log-normal features can be generated artificially to the PL+LN distribution by transferring some of the particles from the PL distribution to the LN distribution. This transfer is calculated through condensation (the condensational transfer). Particles that are to be grown beyond the diameter $D_2$ are transferred to the LN distribution by the condensational transfer instead of keeping them in the PL distribution and increasing the value of $D_2$. The condensational transfer is described in more detail in Sect. 2.2.5.

## 2.2 Aerosol processes

The general dynamic equation for a particular moment $X$ $(= N_{\mathrm{PL}}, S_{\mathrm{PL}}, M_{\mathrm{PL}}, N_{\mathrm{LN}}, S_{\mathrm{LN}},$ or $M_{\mathrm{LN}})$ in a one-dimensional (temporal coordinate only) simulation is

$$\frac{\mathrm{d}X}{\mathrm{d}t} = \mathrm{npf}_X + \mathrm{cond}_X + \mathrm{coag}_X + \mathrm{loss}_X^{\mathrm{coag}} + \mathrm{loss}_X^{\mathrm{dep}} + \mathrm{transfer}_X^{\mathrm{coag}} + \mathrm{transfer}_X^{\mathrm{cond}}, \tag{13}$$

where terms on the right hand side denote new particle formation, condensation, coagulation, coagulational losses, depositional losses, coagulational transfer, and condensational transfer, respectively. The formulation of the terms is described next.

### 2.2.1 New particle formation

New particle formation is modelled by a term

$$
\mathrm{npf}_X =
\begin{cases}
J(t), & X = N_{\mathrm{PL}} \\
J(t)\, s_1, & X = S_{\mathrm{PL}} \\
J(t)\, m_1, & X = M_{\mathrm{PL}} \\
0, & X = N_{\mathrm{LN}},\, S_{\mathrm{LN}},\, \text{or } M_{\mathrm{LN}}
\end{cases}
, \tag{14}
$$

where $J(t)$ can be calculated, e.g., through any nucleation theory, in which $J(t)$ depend also on vapor concentrations and temperature, for example. However, finding the correct formulation for $J(t)$ is outside of the scope of this article; thus, we decided to use formation rate as an input parameter that can be either a constant, $J$, or a time-dependent function, $J(t)$.
Additionally, inverse modelling is done to obtain $J(t)$ from measured particle number concentrations.

The sizes of a newly formed particle ($D_1$, $s_1$, and $m_1$) can be obtained from nucleation theories, but they are assumed to be constants here. In the case where $J(t)$ suddenly drops to zero but condensation still continues, concentration of particles with diameters around $D_1$ would subsequently decrease down to zero due to the growth of newly formed particles to larger diameters. Therefore, $D_1$, as a parameter of the PL distribution, should be a variable to model the distribution accurately.
However, this would change the distribution back to a more complex four-parameter distribution that is outside of the scope of this article. In that case or with smoothly decreasing $J(t)$, $\alpha$ has a tendency to increase.

### 2.2.2 Condensation

Condensation rate [$\mathrm{kg\, m^{-3} s}$] of vapor $v$ on a particle distribution, PL or LN (denoted with $i$), can be modelled as (Olin et al., 2015)

$$
\mathrm{cond}_{M_i} = \int_{-\infty}^{\infty} \frac{\mathrm{d}m_{\mathrm{p},v}}{\mathrm{d}t} \left.\frac{\mathrm{d}N}{\mathrm{d}\ln D_{\mathrm{p}}}\right|_i \mathrm{d}\ln D_{\mathrm{p}}, \tag{15}
$$

where $\frac{\mathrm{d}m_{\mathrm{p},v}}{\mathrm{d}t}$ is the mass growth rate of a particle [$\mathrm{kg\, s^{-1}}$] due to vapor $v$ (Lehtinen and Kulmala, 2003),

$$
\frac{\mathrm{d}m_{\mathrm{p},v}}{\mathrm{d}t} = 2\pi(D_{\mathrm{p}} + D_v)(\mathcal{D}_{\mathrm{p}} + \mathcal{D}_v)(C_{v,\infty} - C_{v,\mathrm{p}}) \frac{\mathrm{Kn}+1}{0.377\mathrm{Kn}+1+\frac{4}{3\phi}(\mathrm{Kn}^2 + \mathrm{Kn})}, \tag{16}
$$

where $D_v$ is the diameter of a vapor molecule. $\mathcal{D}_{\mathrm{p}}$ and $\mathcal{D}_v$ are the diffusion coefficients of a particle and of a vapor molecule, respectively. $C_{v,\infty}$ and $C_{v,\mathrm{p}}$ are the mass concentration of the vapor in the far-field and over the particle surface, respectively.
Kn and $\phi$ are the Knudsen number and the mass accommodation coefficient, respectively. The concentration $C_{v,\mathrm{p}}$ is

$$
C_{v,\mathrm{p}} = \Gamma_v\, C_{v,\mathrm{sat}} \exp\left(\frac{4\mathcal{S}m_v}{k_{\mathrm{B}}T\rho D_{\mathrm{p}}}\right), \tag{17}
$$

where $\Gamma_v$, $C_{v,\mathrm{sat}}$, and $m_v$ are activity, the saturation concentration, and the molecule mass of the vapor, respectively, $\mathcal{S}$ is surface tension, $k_{\mathrm{B}}$ is the Boltzmann constant, and $T$ is temperature.

If all the parameters in Eq. (16) do not depend on the spatial location, as is the case in a one-dimensional simulation, the mass growth rate can be considered a function of time and the particle diameter only. Here, the mass growth rate (single-component case) is expressed using condensational growth rate $g(t, D_\mathrm{p})$

$$\frac{\mathrm{d}m_\mathrm{p}}{\mathrm{d}t}(t, D_\mathrm{p}) = \frac{\mathrm{d}m_\mathrm{p}}{\mathrm{d}D_\mathrm{p}} \cdot \frac{\mathrm{d}D_\mathrm{p}}{\mathrm{d}t}(t, D_\mathrm{p}) = \frac{\pi}{2} \rho D_\mathrm{p}^2 g(t, D_\mathrm{p}). \tag{18}$$

Hence, the condensation rate for a particle distribution becomes

$$\mathrm{cond}_{M_i} = \frac{\pi}{2} \rho \int\limits_{-\infty}^{\infty} D_\mathrm{p}^2 g(t, D_\mathrm{p}) \left.\frac{\mathrm{d}N}{\mathrm{d}\ln D_\mathrm{p}}\right|_i \mathrm{d}\ln D_\mathrm{p}, \tag{19}$$

which has an analytical solution for the both distributions, PL and LN, when $g(t, D_\mathrm{p})$ can be expressed with a polynomial of $D_\mathrm{p}$. The mass growth rate is proportional to $D_\mathrm{p}^2$ if the following conditions are met: 1) the particle size is in free-molecular regime, 2) $D_p \gg D_v$, 3) $C_{v,\infty} \gg C_{v,\mathrm{p}}$. The last one applies when the particle size is large or when the vapor has low saturation vapor pressure. Since particle sizes near the molecular size are modelled in this article, only the first condition applies satisfactorily. Nevertheless, this article concentrates mainly in cases where the mass growth rate is assumed to be proportional to $D_\mathrm{p}^2$. Additionally, a single test case, where the mass growth rate is calculated using Eqs. (16) – (17), is presented. The main point in this article is not to provide the correct formulation for $g(t, D_\mathrm{p})$, but to compare different models, and additionally to perform inverse modelling to obtain $g(t)$ from the time evolution of measured aerosol size distributions. Due to the assumption of the proportionality of the mass growth rate, the condensational growth rate becomes size-independent, and finally, the condensation terms used in Eq. (13) become

$$\mathrm{cond}_X = \begin{cases} 0, & X = N_i \\ 2\pi \, g(t) \int_{-\infty}^{\infty} D_\mathrm{p} \, \mathrm{d}N_i, & X = S_i \\ \frac{\pi}{2} \rho \, g(t) \int_{-\infty}^{\infty} D_\mathrm{p}^2 \, \mathrm{d}N_i, & X = M_i \end{cases} \tag{20}$$

where $\mathrm{d}N_i$ is an abbreviation of

$$\left.\frac{\mathrm{d}N}{\mathrm{d}\ln D_\mathrm{p}}\right|_i \mathrm{d}\ln D_\mathrm{p}. \tag{21}$$

The analytical solution for Eq. (20) is

$$\mathrm{cond}_X = X \, g(t) \cdot \begin{cases} 0, & X = N_{\mathrm{PL}} \\ \frac{2}{D_1}\left(\frac{\alpha+2}{\alpha+1}\right)\left(\frac{d^{\alpha+1}-1}{d^{\alpha+2}-1}\right), & X = S_{\mathrm{PL}} \\ \frac{3}{D_1}\left(\frac{\alpha+3}{\alpha+2}\right)\left(\frac{d^{\alpha+2}-1}{d^{\alpha+3}-1}\right), & X = M_{\mathrm{PL}} \\ 0, & X = N_{\mathrm{LN}} \\ \frac{2}{D_\mathrm{g}} \exp\left(-\frac{3}{2}\ln^2\sigma\right), & X = S_{\mathrm{LN}} \\ \frac{3}{D_\mathrm{g}} \exp\left(-\frac{5}{2}\ln^2\sigma\right), & X = M_{\mathrm{LN}} \end{cases} \tag{22}$$

when $\alpha$ is not -3, -2, or -1.

When the mass growth rate is calculated from the vapor concentrations and the properties of the vapor and the particles using Eqs. (16) – (17), it rarely can be expressed with a polynomial of $D_\mathrm{p}$, unless polynomial fits are done for the function. However, if the vapor concentrations or the other properties are allowed to vary during the simulation, fits for the mass growth rate function may become inconvenient. In that case, the integral in Eq. (19) cannot be solved analytically. Therefore, numerical integration is required, in which Eq. (19) is calculated in a form of

$$\mathrm{cond}_{M_i} = \frac{\pi}{2}\rho \sum_{j=1}^{n} D_j^2 \, g(t, D_j) \left. \frac{\mathrm{d}N}{\mathrm{d}\ln D_\mathrm{p}} \right|_i \ln \frac{D_{j+1}}{D_j}, \tag{23}$$

where $D_j$ is the particle diameter of the size section $j$ used in numerical integration when the particle diameter range is split into $n$ sections. Computational cost of numerical integration is, however, higher compared to analytical solution of the integrals. Therefore, Gaussian quadratures are used here to reduce the associated computing time; they provide the optimal particle diameters and their weights for efficient evaluation of the integrals. The details of the Gaussian quadratures are described in Appendix A.

### 2.2.3 Coagulation

Coagulation is modelled as intramodal coagulation within the PL distribution and within the LN distribution, and as intermodal coagulation from the PL distribution to the LN distribution. The coagulation terms derived from the equations of Whitby and

McMurry (1997) are

$$\text{coag}_{N_{\text{PL}}} = -\frac{1}{2} \int\limits_{-\infty}^{\infty} \int\limits_{-\infty}^{\infty} \beta(D_{\text{p}}, D'_{\text{p}}) \, \mathrm{d}N_{\text{PL}} \, \mathrm{d}N'_{\text{PL}}$$

$$- \int\limits_{-\infty}^{\infty} \int\limits_{-\infty}^{\infty} \beta(D_{\text{p}}, D'_{\text{p}}) \, \mathrm{d}N_{\text{PL}} \, \mathrm{d}N'_{\text{LN}} \tag{24}$$

$$\text{coag}_{S_{\text{PL}}} = -\frac{1}{2} \int\limits_{-\infty}^{\infty} \int\limits_{-\infty}^{\infty} \left[ 2s_{\text{p}} - \left( s_{\text{p}}^{\frac{3}{2}} + s_{\text{p}}'^{\frac{3}{2}} \right)^{\frac{2}{3}} \right] \beta(D_{\text{p}}, D'_{\text{p}}) \, \mathrm{d}N_{\text{PL}} \, \mathrm{d}N'_{\text{PL}}$$

$$- \int\limits_{-\infty}^{\infty} \int\limits_{-\infty}^{\infty} s_{\text{p}} \, \beta(D_{\text{p}}, D'_{\text{p}}) \, \mathrm{d}N_{\text{PL}} \, \mathrm{d}N'_{\text{LN}} \tag{25}$$

$$\text{coag}_{M_{\text{PL}}} = - \int\limits_{-\infty}^{\infty} \int\limits_{-\infty}^{\infty} m_{\text{p}} \, \beta(D_{\text{p}}, D'_{\text{p}}) \, \mathrm{d}N_{\text{PL}} \, \mathrm{d}N'_{\text{LN}} \tag{26}$$

$$\text{coag}_{N_{\text{LN}}} = -\frac{1}{2} \int\limits_{-\infty}^{\infty} \int\limits_{-\infty}^{\infty} \beta(D_{\text{p}}, D'_{\text{p}}) \, \mathrm{d}N_{\text{LN}} \, \mathrm{d}N'_{\text{LN}} \tag{27}$$

$$\text{coag}_{S_{\text{LN}}} = -\frac{1}{2} \int\limits_{-\infty}^{\infty} \int\limits_{-\infty}^{\infty} \left[ 2s_{\text{p}} - \left( s_{\text{p}}^{\frac{3}{2}} + s_{\text{p}}'^{\frac{3}{2}} \right)^{\frac{2}{3}} \right] \beta(D_{\text{p}}, D'_{\text{p}}) \, \mathrm{d}N_{\text{LN}} \, \mathrm{d}N'_{\text{LN}}$$

$$+ \int\limits_{-\infty}^{\infty} \int\limits_{-\infty}^{\infty} \left[ \left( s_{\text{p}}^{\frac{3}{2}} + s_{\text{p}}'^{\frac{3}{2}} \right)^{\frac{2}{3}} - s'_{\text{p}} \right] \beta(D_{\text{p}}, D'_{\text{p}}) \, \mathrm{d}N_{\text{PL}} \, \mathrm{d}N'_{\text{LN}} \tag{28}$$

$$\text{coag}_{M_{\text{LN}}} = \int\limits_{-\infty}^{\infty} \int\limits_{-\infty}^{\infty} m_{\text{p}} \, \beta(D_{\text{p}}, D'_{\text{p}}) \, \mathrm{d}N_{\text{PL}} \, \mathrm{d}N'_{\text{LN}}, \tag{29}$$

where $\beta(D_{\text{p}}, D'_{\text{p}})$ is the coagulation coefficient of particles with the diameters of $D_{\text{p}}$ and $D'_{\text{p}}$ calculated with the equation

$$\beta(D_{\text{p}}, D'_{\text{p}}) = 2\pi(D_{\text{p}} + D'_{\text{p}})(\mathcal{D}_{\text{p}} + \mathcal{D}'_{\text{p}}) f(\text{Kn}_{\text{coag}}), \tag{30}$$

where $f(\text{Kn}_{\text{coag}})$ is the transition regime function of Dahneke (1983)

$$f(\text{Kn}_{\text{coag}}) = \frac{1 + \text{Kn}_{\text{coag}}}{1 + 2\text{Kn}_{\text{coag}} + 2\text{Kn}_{\text{coag}}^2}, \tag{31}$$

where $\text{Kn}_{\text{coag}}$ is the Knudsen number for coagulation

5    $$\text{Kn}_{\text{coag}} = \frac{4(\mathcal{D}_{\text{p}} + \mathcal{D}'_{\text{p}})}{(D_{\text{p}} + D'_{\text{p}})\sqrt{\bar{c}^2 + \bar{c}'^2}}, \tag{32}$$

where $\bar{c}$ and $\bar{c}'$ are the mean thermal velocities of particles with the diameters of $D_{\text{p}}$ and $D'_{\text{p}}$.

The integrals in Eqs. (24) – (29) cannot be solved analytically in the transition regime because Eq. (31) cannot be presented in a polynomial form. Therefore, the integrals are calculated numerically or by using quadratures in the same manner as with the condensation term described in Appendix A.

## 2.2.4 Particle losses

The losses due to coagulation of the particles in the PL+LN distribution to the background distribution excluded from the PL+LN distribution are considered the coagulational losses. Particles in the background distribution are assumed to be significantly larger than the particles in the PL+LN distribution. Therefore, the particle diameters of the background distribution can be approximated with a single diameter value, e.g., $\mathrm{CMD_{bg}}$ (count median diameter). According to Kerminen and Kulmala (2002), the coagulation coefficient will then become

$$\beta(D_\mathrm{p}, \mathrm{CMD_{bg}}) \approx \beta(D_1, \mathrm{CMD_{bg}}) \left(\frac{D_\mathrm{p}}{D_1}\right)^{l_\mathrm{bg}}, \tag{33}$$

where $l_\mathrm{bg}$ is the exponent depending on $\mathrm{CMD_{bg}}$. The value of $l_\mathrm{bg}$ ranges between -2 and -1 (Lehtinen et al., 2007). The coagulational loss term, e.g., for a number concentration is

$$\mathrm{loss}_{N_i}^\mathrm{coag} = -N_\mathrm{bg} \int_{-\infty}^{\infty} \beta(D_\mathrm{p}, \mathrm{CMD_{bg}}) \mathrm{d}N_i \approx -N_\mathrm{bg} \beta(D_1, \mathrm{CMD_{bg}}) D_1^{-l_\mathrm{bg}} \int_{-\infty}^{\infty} D_\mathrm{p}^{l_\mathrm{bg}} \, \mathrm{d}N_i, \tag{34}$$

in which the last integral can be solved analytically. The analytical solutions for the coagulational loss terms are

$$\mathrm{loss}_X^\mathrm{coag} = -X N_\mathrm{bg} \cdot \begin{cases} \beta(D_1, \mathrm{CMD_{bg}}) \left(\frac{\alpha}{\alpha + l_\mathrm{bg}}\right) \left(\frac{d^{\alpha + l_\mathrm{bg}} - 1}{d^\alpha - 1}\right), & X = N_\mathrm{PL} \\ \beta(D_1, \mathrm{CMD_{bg}}) \left(\frac{\alpha + 2}{\alpha + 2 + l_\mathrm{bg}}\right) \left(\frac{d^{\alpha + 2 + l_\mathrm{bg}} - 1}{d^{\alpha + 2} - 1}\right), & X = S_\mathrm{PL} \\ \beta(D_1, \mathrm{CMD_{bg}}) \left(\frac{\alpha + 3}{\alpha + 3 + l_\mathrm{bg}}\right) \left(\frac{d^{\alpha + 3 + l_\mathrm{bg}} - 1}{d^{\alpha + 3} - 1}\right), & X = M_\mathrm{PL} \\ \beta(D_\mathrm{g}, \mathrm{CMD_{bg}}) \exp\left[\frac{1}{2} l_\mathrm{bg}^2 \ln^2 \sigma\right], & X = N_\mathrm{LN} \\ \beta(D_\mathrm{g}, \mathrm{CMD_{bg}}) \exp\left[\left(\frac{1}{2} l_\mathrm{bg}^2 + 2 l_\mathrm{bg}\right) \ln^2 \sigma\right], & X = S_\mathrm{LN} \\ \beta(D_\mathrm{g}, \mathrm{CMD_{bg}}) \exp\left[\left(\frac{1}{2} l_\mathrm{bg}^2 + 3 l_\mathrm{bg}\right) \ln^2 \sigma\right], & X = M_\mathrm{LN} \end{cases} \tag{35}$$

when $\alpha$ is not 0 or $-l_\mathrm{bg}$.

The losses to walls due to diffusion of particles are considered the depositional losses. They are modelled with the method of Hussein et al. (2009), in which the deposition rate of particles in a test chamber is

$$\lambda = \frac{1}{V} \sum_w A_w u \tag{36}$$

where $V$ is the volume of the chamber, $A_w$ is the surface area of the wall $w$, and $u$ is the deposition velocity of particles. A simple approximation for the deposition velocity is used here

$$u \propto D_\mathrm{p}^{-1}, \tag{37}$$

which is valid for particles smaller than $100\,\mathrm{nm}$ according to Lai and Nazaroff (2000). The depositional loss term, e.g, for a number concentration now become

$$\mathrm{loss}_{N_i}^{\mathrm{dep}} = -\int_{-\infty}^{\infty} \lambda \mathrm{d}N_i = -k_{\mathrm{dep}} \int_{-\infty}^{\infty} D_{\mathrm{p}}^{-1} \mathrm{d}N_i \tag{38}$$

where all effects, except the effect of the diameter, are included in the deposition coefficient $k_{\mathrm{dep}}$. The last integral can be solved analytically, from which the depositional loss terms become

$$\mathrm{loss}_X^{\mathrm{dep}} = -X k_{\mathrm{dep}} \cdot \begin{cases} D_1^{-1}\left(\frac{\alpha}{\alpha-1}\right)\left(\frac{d^{\alpha-1}-1}{d^{\alpha}-1}\right), & X = N_{\mathrm{PL}} \\[4pt] D_1^{-1}\left(\frac{\alpha+2}{\alpha+1}\right)\left(\frac{d^{\alpha+1}-1}{d^{\alpha+2}-1}\right), & X = S_{\mathrm{PL}} \\[4pt] D_1^{-1}\left(\frac{\alpha+3}{\alpha+2}\right)\left(\frac{d^{\alpha+2}-1}{d^{\alpha+3}-1}\right), & X = M_{\mathrm{PL}} \\[4pt] D_{\mathrm{g}}^{-1}\exp\left(\frac{1}{2}\ln^2\sigma\right), & X = N_{\mathrm{LN}} \\[4pt] D_{\mathrm{g}}^{-1}\exp\left(-\frac{3}{2}\ln^2\sigma\right), & X = S_{\mathrm{LN}} \\[4pt] D_{\mathrm{g}}^{-1}\exp\left(-\frac{5}{2}\ln^2\sigma\right), & X = M_{\mathrm{LN}} \end{cases} \tag{39}$$

when $\alpha$ is not 0 or 1.

The effect of particle losses on the PL distribution is seen as decreased $\alpha$. In the trivial case, as in Eq. (1), $\alpha$ becomes less than zero when $k_{\mathrm{dep}} > g$. This effect is due to increased losses with increasing particle diameters because larger particles have longer residence times from the moment since their formation. However, Eq. (37) counteracts in this effect by decreasing the deposition velocity with increasing particle size, but with small $g$, the effect of increased residence time dominates over the effect of decreased deposition velocity. Additionally, $\alpha$ is further decreased due to coagulational losses.

### 2.2.5 Intermodal particle transfer

The intermodal coagulation is included together with the intramodal coagulation in the coagulation terms ($\mathrm{coag}_X$) seen in Eqs. (24) – (29). The coagulational ($\mathrm{transfer}_X^{\mathrm{coag}}$) and condensational ($\mathrm{transfer}_X^{\mathrm{cond}}$) transfer are modelled as follows.

Particles with the diameter higher than the cut diameter

$$D_{\mathrm{coag}} = \left(D_2^3 - D_{\mathrm{p}}'^3\right)^{1/3} \tag{40}$$

form particles with the diameter higher than $D_2$ after coagulating with a particle having a diameter of $D_{\mathrm{p}}'$, assuming full coalescence (Fig. 2). Those resultant particles are transferred from the PL distribution to the LN distribution, because their particle diameters will correspond with the form of a log-normal distribution rather than a power law distribution, which can be observed using a highly accurate sectional model. The coagulational transfer terms are negative for the PL distribution and

positive for the LN distribution to conserve the moments, and they are expressed as

$$\text{transfer}_{N_{\text{PL}}} = -\text{transfer}_{N_{\text{LN}}} = -\frac{1}{2} \int\limits_{-\infty}^{\infty} \int\limits_{\ln D_{\text{coag}}}^{\infty} \beta(D_{\text{p}}, D_{\text{p}}') \, \mathrm{d}N_{\text{PL}} \, \mathrm{d}N_{\text{PL}}' \tag{41}$$

$$\text{transfer}_{S_{\text{PL}}} = -\text{transfer}_{S_{\text{LN}}} = -\frac{1}{2} \int\limits_{-\infty}^{\infty} \int\limits_{\ln D_{\text{coag}}}^{\infty} \left( s_{\text{p}}^{\frac{3}{2}} + s_{\text{p}}'^{\frac{3}{2}} \right)^{\frac{2}{3}} \beta(D_{\text{p}}, D_{\text{p}}') \, \mathrm{d}N_{\text{PL}} \, \mathrm{d}N_{\text{PL}}' \tag{42}$$

$$\text{transfer}_{M_{\text{PL}}} = -\text{transfer}_{M_{\text{LN}}} = -\frac{1}{2} \int\limits_{-\infty}^{\infty} \int\limits_{\ln D_{\text{coag}}}^{\infty} \left( m_{\text{p}} + m_{\text{p}}' \right) \beta(D_{\text{p}}, D_{\text{p}}') \, \mathrm{d}N_{\text{PL}} \, \mathrm{d}N_{\text{PL}}', \tag{43}$$

which are calculated using the quadrature or numerical integration as in the case of the coagulation terms.

Considering a time step of $\Delta t$ in a Lagrangian simulation, particles with the diameters larger than $D_{\text{cond}} = D_2 - g \, \Delta t$ will grow due to condensation to have the diameters larger than $D_2$ (Fig. 2). Modelling condensation only, the value of $D_2$ at the next time step would increase to $D_2 + g \, \Delta t$. The condensational transfer is used to transfer the particles in the PL distribution with the diameters of $D_{\text{cond}} < D_{\text{p}} < D_2$ to the LN distribution. However, if the condensational transfer is modelled fully, $D_2$ would never increase and all condensation would affect the LN distribution only. In that case, the distributions would separate from each other. For this reason, the effect of condensational transfer is dampened using a factor $\gamma$ as a multiplier in the condensational transfer equations. The factor can obtain values between zero and unity, and it describes how the particles will be distributed between the PL and the LN distributions. The value $\gamma = 0$ produces a distribution that will be mainly in a power law form; the value $\gamma = 1$ produces a log-normal distribution only. To choose a suitable value for $\gamma$ for a simulation, the user should consider how well does the aerosol formation event follow the approximations of the theory described here. The value 0 is suitable only when the aerosol processes follow the theory exactly. To simulate a realistic particle formation event, the value has to be increased towards unity using the following guidelines. The more the following conditions are met, the higher $\gamma$ should be used: (1) particle formation or growth are multicomponent processes, (2) the particle formation rate or the condensation growth rate vary significantly with time, (3) the condensational growth rate varies significantly with the particle size, (4) the background aerosol acting as a coagulation sink does not remain in a nearly constant state during the time domain of the simulation, (5) particle sizes in the background aerosol are not significantly higher than in the PL+LN distribution, (6) the depositional losses cannot be approximated with as simple form as described here, e.g., in the case of complex geometry or turbulent flow. In real atmospheric particle formation events, $\gamma$ should rarely has the value of less than 0.5, which can also be used as an initial guess if figuring the previous guidelines is problematic. If the shapes of the distributions to be modelled are initially known, the value of $\gamma$ can be adjusted to obtain a proper model output, e.g., in the case of inverse modelling. The factor $\gamma$ can also be considered a time-dependent function or a spatially varying variable, but here we concentrate only to constant values of $\gamma$ because the theory behind the value of $\gamma$ is currently unknown.

The number of particles in the PL distribution to be transferred to the LN distribution due to the condensational transfer in the time step of $\Delta t$ is

$$N_{\text{PL}\to\text{LN}} = \gamma \int_{\ln D_{\text{cond}}}^{\ln D_2} \left.\frac{dN}{d\ln D_{\text{p}}}\right|_{\text{PL}} d\ln D_{\text{p}}. \tag{44}$$

Considering infinitesimally small time step ($\Delta t \to 0$), $D_{\text{cond}}$ approaches $D_2$ and $N_{\text{PL}\to\text{LN}}$ approaches

$$\gamma \frac{g\Delta t}{D_2} \left.\frac{dN}{d\ln D_{\text{p}}}\right|_{\text{PL},\, D_{\text{p}}=D_2}. \tag{45}$$

The transferred amounts for $S$ and $M$ are obtained in the same approach as in Eq. (20), and they are negative for the PL distribution and positive for the LN distribution. Hence, the term for the condensational transfer becomes

$$\text{transfer}_X^{\text{cond}} = \gamma \frac{g(t)}{D_2} \left.\frac{dN}{d\ln D_{\text{p}}}\right|_{\text{PL},\, D_{\text{p}}=D_2} \cdot \begin{cases} -1, & X = N_{\text{PL}} \\ -s_2, & X = S_{\text{PL}} \\ -m_2, & X = M_{\text{PL}} \\ +1, & X = N_{\text{LN}} \\ +s_2, & X = S_{\text{LN}} \\ +m_2, & X = M_{\text{LN}} \end{cases}, \tag{46}$$

where $s_2$ and $m_2$ are the surface area and the mass of the particle with the diameter of $D_2$. The condensational transfer does not alter the moments of the total distribution because particles are not altered in the transfer, it only transfers the particles between the distributions; therefore, the value of $\gamma$ has a minor effect only on the moments, but a noticeable effect on the shape of the PL+LN distribution.

## 3  Simulation setup for the evaluation of the PL+LN model

The PL+LN model was evaluated with the simulations of theoretical test cases and a real particle formation case. The evaluation was done against sectional models that yield accurate results due to a high number of size sections. Two types of sectional models were used: fixed-sectional (FS) and moving-center fixed-sectional (MC) models. The models are further subdivided depending on the amount of size sections they use. FS models provide the best accuracy for the particle number concentration and MC models for the mass concentration, when a high number of size sections is modelled. The results from different models and from measurement data are examined by comparing the distributions, the moments ($N$, $S$, and $M$) and the variables, GMD (geometric mean diameter) and GSD (geometric standard deviation). GMD and GSD can be calculated from a continuous or a

discrete total distribution with the equations

$$\ln \text{GMD} = \frac{1}{N} \int\limits_{-\infty}^{\infty} \ln D_\text{p} \, dN = \frac{1}{N} \sum_j N_j \ln D_{\text{p},j} \tag{47}$$

$$\ln^2 \text{GSD} = \frac{1}{N} \int\limits_{-\infty}^{\infty} \ln^2 \left( \frac{D_\text{p}}{\text{GMD}} \right) dN = \frac{1}{N} \sum_j N_j \ln^2 \left( \frac{D_{\text{p},j}}{\text{GMD}} \right), \tag{48}$$

where $D_{\text{p},j}$ is the geometric average particle diameter of the size section $j$. Relative errors of the moments and the variables compared to the reference models are calculated with

$$\delta_X = \frac{X - X_\text{ref}}{X_\text{ref}}, \tag{49}$$

where $X$ and $X_\text{ref}$ are the moment or the variable from the model in examination and from the reference model, respectively. FS models are considered the reference models, with the exception of the mass moment, $M$, in a real particle formation case, for which the reference model is an MC model, because it provides the best mass-conservation.

The capability of the PL+LN model in inverse modelling is also tested using the real measurement data from the particle formation event. The best estimates of the new particle formation rates, $J(t)$, and the condensational growth rates, $g(t)$, obtained from the different models, are compared with each other. These values for $J(t)$ and $g(t)$ are later used in the simulation that is used to examine the output accuracies and computational costs of different models.

The diameter of a newly formed particle was assumed to be a constant, $D_1 = 1.6 \, \text{nm}$, in all cases. The value was chosen because it is in the range of a relevant size of a particle from which atmospheric aerosol formation starts (Kulmala et al., 2007) and of a size of a smallest particle that can be detected with the Particle Size Magnifier (PSM) with the detection efficiency of nearly unity (Vanhanen et al., 2011). Single-component modelling was performed assuming a mixture with the particle bulk density of $\rho = 1.4 \, \text{g cm}^{-3}$ as the component. The value was chosen because it is a relevant density of small particles in the atmosphere (Kannosto et al., 2008).

## 3.1 Theoretical test cases

Theoretical test cases were used to compare the PL+LN model output with a highly accurate FS model. The FS model had 1000 size sections between 1.6 and $10 \, \text{nm}$ (FS1000) which is sufficiently dense to produce accurate results. Additionally, the PL+LN model was compared with the model having a log-normal distribution only (LN). All cases were simulated using constant and equal time steps to obtain a reliable comparison; the simulated time domains ($t_\text{max}$) were split into 3000 time steps.

The input parameters of the test cases are presented in Tab. 1. The Atm and Exh cases represent particle formation cases using input parameter values that reflect conditions typical to the atmosphere and to vehicle exhaust, respectively. Typical new particle formation rates in the atmosphere range from 0.01 to $10 \, \text{cm}^{-3} \, \text{s}^{-1}$ and condensational growth rates from 0.1 to $20 \, \text{nm h}^{-1}$ (Kulmala et al., 2004). In vehicle exhaust, new particle formation rates can reach up to $10^{10} \, \text{cm}^{-3} \, \text{s}^{-1}$ and condensational growth rates up to $20 \, \text{nm s}^{-1}$ (Rönkkö et al., 2006; Uhrner et al., 2007; Olin et al., 2015). To test the PL+LN model in a wide range of $J$ and $g$, low values for Atm cases ($J = 0.1 \, \text{cm}^{-3} \, \text{s}^{-1}$, $g = 1 \, \text{nm h}^{-1}$) and high values for the Exh

case ($J = 10^8\,\text{cm}^{-3}\,\text{s}^{-1}$, $g = 5\,\text{nm}\,\text{s}^{-1}$) were chosen. In addition to constant $J$ and $g$ values, a case having time-dependent $J(t)$ and a case having time-dependent $J(t)$ and size-dependent $g(D_\text{p})$ were simulated.

The Atm1 case includes simultaneous new particle formation, condensation, intramodal-, and intermodal coagulation. For the Atm2 case, depositional losses were also added. The deposition coefficient $k_\text{dep} = 1.8\,\text{nm}\,\text{h}^{-1}$ was calculated by assuming that particle formation takes place in a test chamber with the dimensions of $3\,\text{m} \times 2\,\text{m} \times 2\,\text{m}$ and with the deposition velocities of salt particles measured by Hussein et al. (2009). For the Atm3 case, a background distribution was added to act as the coagulation sink. The chosen values for the number concentration $N_\text{bg} = 10^3\,\text{cm}^{-3}$ and count median diameter $\text{CMD}_\text{bg} =$

$100\,\text{nm}$ of the background distribution have been observed, e.g., in a boreal forest area (Riipinen et al., 2007). The value for the coagulational loss exponent $l_\text{bg} = -1.6$ was obtained from Lehtinen et al. (2007) using $\text{CMD}_\text{bg} = 100\,\text{nm}$. In the Atm4 case, a bell-shaped time-dependent function for the new particle formation rate (a bell-shaped form in the function of the number concentration between 3 and 6 nm is seen in studies of Sihto et al. (2006); Riipinen et al. (2007)) was modelled with

$$J(t) = J_0\, e^{-\left(\frac{t-t_0}{\tau_\text{J}}\right)^2}, \tag{50}$$

where $t_0 = 1000\,\text{s}$ is the time at which the highest new particle formation rate $J_0 = 0.1\,\text{cm}^{-3}\,\text{s}^{-1}$ occurs and $\tau_\text{J} = 5000\,\text{s}$ represents the width of the bell-shaped curve.

The applicability of the PL+LN model using size-dependent condensational growth rates was evaluated with the Atm5 case, where $g(D_\text{p})$ was modelled using Eqs. (16) – (18) and (23) – (A4). In this case, particles were assumed to consist of the mixture of sulfuric acid and water. The growth is modelled as the growth due to sulfuric acid, $\frac{\text{d}m_\text{p,H}_2\text{SO}_4}{\text{d}t}$, calculated using the sulfuric

acid vapor concentration $[\text{H}_2\text{SO}_4] = 0.8 \times 10^7\,\text{cm}^{-3}$, following the growth due to water vapor,

$$\frac{\text{d}m_\text{p,H}_2\text{O}}{\text{d}t} = \left[\frac{1}{Y_{\text{H}_2\text{SO}_4}(D_\text{p})} - 1\right] \cdot \frac{\text{d}m_\text{p,H}_2\text{SO}_4}{\text{d}t}, \tag{51}$$

where $Y_{\text{H}_2\text{SO}_4}(D_\text{p})$ is the mass fraction of sulfuric acid in a particle in water equilibrium, i.e., a particle having the composition with which no condensation or evaporation of water vapor occurs in temperature of $280\,\text{K}$ and relative humidity of $60\,\%$ when the particle diameter is $D_\text{p}$. The approximation of water equilibrium is reasonable because, with these environmental values,

$\sim 2 \times 10^{10}$ times more water molecules than sulfuric acid molecules exist and thus there are probably a sufficient amount of water molecules to condense on the particle to reach the equilibrium state before the next sulfuric acid molecule condenses on it. The properties of the vapors and the particles were calculated, using the equilibrium composition, as described in Olin et al. (2015). These environmental values were chosen because they are relevant values met in the atmosphere and they cause the condensational growth rate function that is far beyond a constant value in the particle diameter range of this case (from $1.6\,\text{nm}$

to $8\,\text{nm}$), as seen in Fig. 4, which provides a beneficial test to examine how the model behaves with size-dependent $g$.

The Exh case represents simultaneous new particle formation, condensation, intramodal- and intermodal coagulation, coagulational losses, and depositional losses occurring in diesel vehicle exhaust inside the ageing chamber of a laboratory sampling system. The values $N_\text{bg} = 10^6\,\text{cm}^{-3}$ and $\text{CMD}_\text{bg} = 60\,\text{nm}$ were obtained from the measurements of Rönkkö et al. (2013) and the corresponding $l_\text{bg} = -1.5$ from Lehtinen et al. (2007) using $\text{CMD}_\text{bg} = 60\,\text{nm}$. The deposition coefficient $k_\text{dep} = 0.07\,\text{nm}\,\text{s}^{-1}$ was calculated using the ageing chamber dimensions of $5\,\text{cm}\,(\text{diameter}) \times 100\,\text{cm}\,(\text{length})$ and the deposition velocities of salt particles measured by Hussein et al. (2009).

Because the test cases are purely theoretical, the need of constructing log-normal features to the distributions through the
condensational transfer artificially is minimal. In the Atm4 case, a time-dependent new particle formation rate suggests using
the condensational transfer, but, according to the analysis of the shapes and the moments of the distributions, the output is
not very sensitive to the value of $\gamma$, which is probably due to the bell-shaped function for the new particle formation rate that
produces distributions containing both power law and log-normal features. In the Atm5 case, a size-dependent condensational
growth rate outputs size distributions having features of a different kind, and thus the PL+LN distribution does not fit very
satisfactorily in this case. According to the analysis of the shapes and the moments of the distributions in the Atm5 case, the
best estimate for $\gamma$ is 0.25, which was used in the Atm5 case; $\gamma = 0$ was used in all the other cases.

## 3.2   Mobile aerosol chamber particle formation event

The mobile aerosol chamber is a Teflon bag with the dimensions of $3\,\mathrm{m} \times 2\,\mathrm{m} \times 2\,\mathrm{m}$. The chamber is operated in a batch process,
i.e., firstly, the chamber is filled with the air sample, and secondly, the sample is measured from the chamber. UV lights with
the wavelength of $254\,\mathrm{nm}$ (UVC) are used in the chamber to initiate new particle formation and to boost the aging of the
aerosol through photochemical processes. The chamber simulates a particle formation event occurring in the atmosphere, but
with shorter time scale due to the UV lights. The chamber is designed to be mobile; therefore, it is fit to a car trailer.

The particle formation event measurement was performed at a street canyon measurement site of Helsinki Region Envi-
ronmental Services Authority (HSY) located in Mäkelänkatu, Helsinki, Finland. The street had dense traffic during the mea-
surement in 22 April 2015. The chamber was firstly filled with urban air and, once filled, the air sample was sucked with the
measurement devices located in the mobile laboratory vehicle. The details of processing the experimental data to obtain the
moments ($N$, $S$, and $M$) and the variables, GMD and GSD, are described in Appendix B.

### 3.2.1   Obtaining $J(t)$ and $g(t)$ through inverse modelling

Obtaining the values for the new particle formation rate, $J(t)$, and for the condensational growth rate, $g(t)$, for the particle
formation event, occurred in a mobile aerosol chamber, was performed through inverse modelling. A time domain, starting
from $152\,\mathrm{s}$ before switching the UV lights on and ending to $1663\,\mathrm{s}$ after switching the UV lights on, including 13 Nano-
SMPS (Nano Scanning Mobility Particle Sizer) measurement scans, was simulated using different values for $J(t)$ and $g(t)$.
Following the approach of Verheggen and Mozurkewich (2006), the least squares method was used to minimize the errors of
the concentrations $N$, $S$, and $M$ at 13 time moments which represent the middles of the Nano-SMPS scans. The values for
$J(t)$ and $g(t)$ were assumed constants within a time step of a Nano-SMPS scan, $150\,\mathrm{s}$. The condensational growth rate was
assumed also size-independent due to the lack of knowledge of the vapors participating in the condensation process. ODE45
solver was used in the simulations, and it provides the time steps that are sufficiently short to keep the result from altering more
than $1\,\%$ compared to a previous time step but sufficiently long to keep the total computing time convenient.

Coagulation within the nucleation mode was included in the simulations, but the coagulational losses to the background
mode was neglected because its low number concentration would had a minor effect only on the nucleation mode. The particles
formed in this case are possibly multicomponent due to the origin of the vapors, the new particle formation rate seems to vary

significantly with time, and the measured distributions are wide, GSD values were up to 2. Therefore, a high value for the condensational transfer factor $\gamma$ is expected to produce the best results using the PL+LN model. A constant value $\gamma = 0.8$ was used in the simulations because it produces the results that are the closest to the results of highly accurate sectional models, in this case. Due to a high value of $\gamma$, the coagulational transfer would had a minor effect only, and therefore it was neglected in the simulations. The depositional losses were modelled using the deposition coefficient $k_{dep} = 3780\,\mathrm{nm\,h^{-1}}$ which is obtained by fitting the simulated number concentrations with the measured ones after particle formation and growth were quenched ($t > 1500\,\mathrm{s}$).

Firstly, inverse modelling was performed using the PL+LN model. The time series of $J(t)$ and $g(t)$, that produced the most corresponding concentrations compared to the measured ones (Fig. 5), are presented in Figs. 6 and 7. Secondly, inverse modelling was performed using both an FS model having 400 size sections between 1.6 and 100 nm (FS400) and an MC model having 100 size sections between 1.6 and 100 nm (MC100), separately. The computing times of the FS400 and the MC100 models are significantly longer than of the PL+LN model; therefore, the time series of $J(t)$ and $g(t)$, obtained using the PL+LN model, were used as initial guesses when inverse modelling was performed using the FS400 or the MC100 model, to reduce the computing times. The associated computing time of the automatic inverse modelling procedure using the PL+LN model was approximately 2 orders of magnitude shorter than by using the FS400 or the MC100 model, which implies that a significant improvement in the computing time can be obtained using the PL+LN model in the case of inverse modelling. Inverse modelling was also performed using computationally more efficient LN model.

It can be seen from Fig. 6 that there are only minor differences in the values of $J(t)$ between the different models used in inverse modelling. The PL+LN model seems to need higher new particle formation rates compared to the accurate models, the FS400 and the MC100 models. Conversely, the LN model seems to need lower new particle formation rates. These denote that the PL+LN model has a tendency to underestimate the number concentrations, and that the LN model has a tendency to overestimate the number concentrations, because the output number concentrations of the models are nearly equal (Fig. 5). The values of $g(t)$, seen in Fig. 7, have also only minor differences between the different models. The values of $J(t)$ and $g(t)$ before the UV lights ($t < 0\,\mathrm{s}$) seem unphysically high. That is caused because the simulation begins with no particles at $t = -252\,\mathrm{s}$ but the measurement data include some nucleation mode particles at that time, even though the background aerosol distribution was subtracted from the measured distribution. All the particles are not subtracted because the subtraction was done with a purely log-normal distribution, which is exactly not the case of a measured data. Therefore, the simulation time range $-252\,\mathrm{s} < t < -152\,\mathrm{s}$ was used to produce the measured nucleation mode at the time of the first Nano-SMPS scan, $t = -152\,\mathrm{s}$.

The effect of the choice for the lowest particle diameter, $D_1$, for the PL+LN model was also examined by performing inverse modelling with $D_1 = 1\,\mathrm{nm}$ and $D_1 = 3\,\mathrm{nm}$, in addition to $D_1 = 1.6\,\mathrm{nm}$. Approximately 40 % higher values for $J(t)$ were needed to produce the measured concentrations when $D_1 = 1\,\mathrm{nm}$ was used compared to $D_1 = 3\,\mathrm{nm}$. This deviation occurs because smaller particles have higher losses, and with higher value for $D_1$, the smallest particles do not exist. The situation is the same for sectional models, because the smallest particle size needs to be chosen for them too. The choice for $D_1$ does not have significant effect on the condensational growth rate, in this case, due to size-independent $g(t)$.

The effect of the choice for the value of the condensational transfer factor, $\gamma$, for the PL+LN model was also examined by

performing inverse modelling with the values between 0 and 1, in addition to $\gamma = 0.8$. With the values between 0.4 and 0.9, the times series of $J(t)$ obtained from inverse modelling deviate within 3 %, in average. However, with the values outside of that range, the deviation increased up to 36 %. The highest new particle formation rates are needed when the value of $\gamma$ is low because the PL distribution dominates the number of particles, which leads to more small particles, which have high losses. Only minor effect (the deviations within 4 %) was seen on the time series of the condensational growth rate when the value of $\gamma$

was altered between 0 and 1. In conclusion, choosing the value of 0.5 for $\gamma$ if better guess for its value is lacking, as mentioned earlier, would produce reasonable results.

Figure 8 presents the time series of GMD and GSD of the nucleation mode, obtained from the different models used in inverse modelling and from the measured data. It can be observed that the models output these variables relatively well compared to the measured data, although they were not selected as the variables, of which errors are to be minimized, in inverse modelling.

However, underestimations of GSD are seen with the LN model. The measured values of GMD and GSD before the UV lights are inaccurate due to the assumption of a log-normal background aerosol distribution mentioned before.

### 3.2.2    Simulation setup for the examination of the accuracy and the computational cost of the PL+LN model

To examine the accuracy and computational cost of the PL+LN model, the simulations using different models were performed using equal time series of $J(t)$ and $g(t)$ and equal time stepping. The time series obtained using the FS400 model are considered

the best estimates to produce the measurement results due to the highest number of size sections modelled. Therefore, they were used as the time series for all the models used here, PL+LN, LN, FS400, MC100, FS35, and MC10. FS35 and MC10 denote a fixed-sectional model having 35 size sections and a moving-center fixed-sectional model having 10 size sections between 1.6 and 300 nm, respectively. These section number were used because they provide approximately the same computing times as the PL+LN model. Higher upper diameter limit for the FS35 and the MC10 models were chosen due to higher numerical

diffusion associated to these models, which causes higher concentrations in large particle diameters compared to more accurate models. The time series of $J(t)$ and $g(t)$ used here are presented as fits in Figs. 6 and 7. The fitted functions were used rather than the time series because the time series having sharp edges would produce distributions having sharp edges as well, which would be unphysical. Here, the time series have sharp edges due to a very limited number of data points (Nano-SMPS scans) within the time domain. It is obvious that because the fits are not very near the time series, the outputs of the models will not

be very near the measured data. Nevertheless, the fits were used because the purpose here is to examine the accuracy and the computational cost of the PL+LN model, which is done against the highly accurate models, the FS400 and the MC100 models. This comparison is the most properly done when all the models have the same functions for $J(t)$ and $g(t)$. The time domain to be simulated was split into 7953 time steps for all the models. The time splitting was obtained from the ODE45 solver used with the FS400 model. The time steps had the lengths of between 0 and 0.5 s, the shortest ones being in the beginning of the time domain.

The accuracy of the PL+LN model is examined by comparing the relative errors ($\delta_X$) of the moments, $N$, $S$, and $M$, and the variables, GMD and GSD. The reference distributions used in the comparison are the distributions produced by the FS400

model and by the MC100 model, which are considered the models that most accurately conserve the number and the mass concentrations, respectively. The distributions from the FS400 model are used as the reference distribution when calculating $\delta_X$ for all the moments and variables, with the exception of the mass concentration, $M$, for which the MC100 model is used as the reference model. Because the distributions produced by the sectional models are considered here the correct ones rather than the measured distributions, the modelled distributions are used as the reference distributions. In this manner, the differences of the model outputs are caused by the models itself, e.g., due to numerical diffusion or some simplifications used in the model, not by how accurately they correspond with the measured data. The accuracies of the LN, FS35, and MC10 models are also examined.

All the other input parameters were the same as were used with the inverse modelling. The simulations used to examine the model accuracies provide also the possibility of comparing the computational costs of different models, because all the simulations were run using the same computer (Intel Core i5-3470 processor at 3.2 GHz) and had equal time stepping, and therefore equal number of computations of the general dynamic equation per a moment or a size section. Additionally, the sensitivity of the value of $\gamma$ was examined using also the values of 0.1, 0.5, and 0.9, in addition to the value of 0.8.

## 4 Results and discussion

### 4.1 Theoretical test cases

Figure 9 shows the size distributions at the ends of the theoretical test simulations using different models. The distributions of the Atm cases during the whole time domain are presented as a video in the Supplement. It can be observed that the shape of the distribution produced by the PL+LN model is nearly equal to the reference distribution (FS1000). The largest deviations between the PL+LN distributions and the reference distributions are the gap between the PL and the LN distribution and the sharp peak in the PL distribution. These are most clearly seen in the Atm4 case where $\alpha$ is the highest. In the Atm5 case, the shape of the distribution produced by the FS1000 model is different: the distribution of the smallest particles do not follow a power law form due to low condensational growth rates near the particle diameter of $D_1$. Consequently, the PL+LN model is not able to express the distribution correctly at very small particle sizes. The effect of $\gamma$ is also seen with the Atm5 case where the ratio of the concentrations of the LN distribution and of the PL distribution is higher due to higher $\gamma$. Conversely, the distributions produced by the LN model are far beyond the reference distributions. In the Atm4 and the Atm5 cases where new particle formation rates decrease towards the end of the simulation, the LN model begins to act better while the reference distribution transforms towards a log-normal shape.

The effect of the depositional losses can be seen as a decreased $\alpha$ in the Atm2 case compared to the Atm1 case. Because $k_{\mathrm{dep}} > g$, the value of $\alpha$ becomes negative. Comparing the Atm3 case with the Atm2 case, it can be seen that the coagulational losses decrease $\alpha$ further. In the Atm4 and the Atm5 cases, the values of $\alpha$ are again increased compared to the Atm3 case. This occurs because $J(t)$ decreases with increasing time but $g$ remains constant, in time, and thus there will be less small particles with increasing time. The distribution of the Exh case is mainly comparable to the Atm1 case with the exception of higher concentration levels in the Exh case due to higher $\frac{J}{g}$. It can be also observed that the ratio of the concentrations of the LN

distribution and of the PL distribution is higher in the Exh case than in the Atm1 case. This is due to increased coagulational

transfer in the Exh case because it is calculated through the intramodal coagulation, of which rate is proportional to $N_{\mathrm{PL}}^2$. The depositional and coagulational losses do not have significant effect on the distribution in the Exh case because $k_{\mathrm{dep}} \ll g$ and $N_{\mathrm{bg}} \ll N$.

Figure 10 shows the relative errors of the moments ($\delta_X$) in the PL+LN model compared to the reference model, FS1000, as a function of time, and Tab. 2 at the ends of the test simulations. The highest relative errors of the total concentrations, $N$, $S$,

and $M$, are usually met at the ends of the simulated time domains, and they are less than 2 % in all the cases, except the Atm5 case, the total number concentration $N$ being the most accurately conserved moment. In the Atm5 case, $|\delta_X|$ for the moments are 17 % at the highest. The errors of this high level are caused by the reference distribution having features that do not fit well neither with the PL distribution nor with the LN distribution. The form of the size-dependence of the condensational growth rate in the Atm5 case represents, however, one of the worst cases that are to be simulated with the PL+LN model.

For comparison, the parameters of the Atm5 case would cause $|\delta_X|$ to reach the levels of 24 % if the condensational transfer is neglected, the levels of 19 % if only the PL distribution is simulated, and the levels of 90 % if only the LN distribution is simulated. GMD and GSD have $|\delta_X|$ of less than 0.5 % in the cases with the constant parameters, but for the Atm4 and the Atm5 cases, the errors are higher (around $\pm 4$ %). All the cases, with the exception of the Atm5 case, can be simulated with the PL distribution only to achieve the levels of the relative errors as with the PL+LN distribution, but the need of the LN

distribution in addition to the PL distribution arises with the Atm5 case. However, visually inspecting, the LN distribution is needed in all the cases to obtain distributions that have the correct shapes in the highest particle sizes.

The total computing time of the Atm5 case with the PL+LN model compared to the Atm4 case is approximately 2-fold, which is mainly caused by the need of numerical integration in calculation of the condensation terms in the Atm5 case. The associated computing time, and the accuracy, can be controlled by the number of size sections used in numerical integration or

by using a polynomial form for the condensational growth rate. Because condensation is calculated using size sections with the FS1000 model, regardless of the size-dependency of the condensational growth rate, the total computing time increases only about 7 % when switching from size-independent condensational growth rates to size-dependent ones. The increase of the total computing time, in that case, is related to additional computations to obtain the values for the condensational growth rate itself.

### 4.2   Mobile aerosol chamber particle formation event

Particle size distributions obtained from the FS400, the LN, and the PL+LN models are shown as contour plots in Fig. 11 together with the measured distributions. Comparing the plots of the LN and the PL+LN models with the plot of the FS400 model, it can be seen that the PL+LN model behaves better for small diameters than the LN model. However, there is a sharp discontinuity between the PL and the LN distributions in the PL+LN model. In this case, the discontinuity is mainly formed due to the condensational transfer that is separating the distributions from each other. It is also seen that the PL+LN model is capable in vanishing the PL distribution when the aerosol ages and begins to have mainly a log-normal like form. Particle distributions 378 and 978 s after the UV lights were switched on are also shown in Fig. 12. The time $t = 378$ s presents the center of the Nano-SMPS scan where the new particle formation rate is at the highest. At the time $t = 978$ s, new particle

formation was mostly quenched but growth still occurred. The shapes of the distributions produced by the PL+LN model are

near the reference distributions (FS400) with the exception of the gaps between the PL and the LN distributions. The shapes of the distributions at the largest particles produced by the LN model correspond better with the measured distributions than at the smallest particles. In the simulation using the FS35 model, a high numerical diffusion that widens the distribution towards the larger particles is seen. The distributions produced by the MC models, MC100 and MC10, have sharp features but follow the distributions produced by the FS400 model. The number of size sections in the MC10 model is obviously too low to obtain

size distributions that are near the reference distributions. The modelled distributions at the time $t = 978\,\mathrm{s}$ are not very near the measured distribution due to the fitted functions used for $J(t)$ and $g(t)$.

Table 3 presents the computational costs and the accuracies of the models. Computing times are reported relative to the computing time of the PL+LN simulation, $24\,\mathrm{s}$. The PL+LN model has the best accuracy for the total number ($N$) and mass ($M$) concentrations compared to the sectional models with approximately the same computing time (FS35 and MC10) and

to the LN model. The FS35 model is relatively accurate in $N$ output but suffers from high numerical diffusion seen as high relative error (79 %) in $M$. The PL+LN model has also low memory consumption due to a low number of variables. The LN model is, however, the most computationally efficient but the relative errors are high too (up to 17 % in $N$).

The development of $N$, $M$, GMD, GSD, and the relative errors of $N$ and $M$ are shown in Fig. 13. It can be seen that the PL+LN model has nearly the same output for $N$ as the reference models during the whole time domain. The beginning of the

overestimation of $N$ in the LN and in the MC10 models are clearly seen at the region where the new particle formation has the highest rate ($t \approx 400\,\mathrm{s}$). The LN model functions better in $M$, but overestimations are encountered with the MC10 model. In addition to the MC10 model, the FS35 model overestimates $M$ clearly. The PL+LN model outputs $M$ very accurately during the whole time domain. GMD is overestimated slightly with the FS35 and the MC10 models due to numerical diffusion, but the PL+LN and the LN models output it accurately during the whole time domain. The highest error in GMD produced by

the PL+LN model is the underestimation of 1.3 % compared to the FS400 model which occurs at the time of the highest new particle formation rate. The highest deviations between the models are seen in the development of GSD. The PL+LN model underestimates GSD with 3 % at the end of the time domain, but the FS35 model overestimates it significantly and the MC10 model and the LN model underestimate it significantly. Additionally, the MC10 model suffers from uneven behavior due to its low number of size sections. The relative errors of the moments, $\delta_N$ and $\delta_M$, are at the highest levels mostly at the ends of the

simulations. The models having at least the same computational efficiency as the PL+LN model (the FS35, the MC10, and the LN models) fail to produce $N$ and $M$ accurately: the relative errors can be up to tens of percent. The relative errors with the PL+LN model are below 2 % during the whole time domain. In conclusion, the PL+LN model has the best accuracy for the production of $N$, $M$, GMD, and GSD during the whole time domain compared to the other models having at least the same computational efficiency.

Figure 14 presents the particle distribution at the time of $t = 978\,\mathrm{s}$ using the PL+LN model with different values of $\gamma$. It can be seen that the PL distribution in the total distribution is mostly dominating when a low value for $\gamma$ is used (left pane). Conversely, using a high value of $\gamma$ (right pane) produces a more log-normal like form, which, at least in this case, corresponds best with the measured distribution. However, a gap between the PL and the LN distributions is larger in cases of high values

of $\gamma$. The sensitivity of the value of $\gamma$ is also shown in Tab. 4, in which the relative errors of $N$, $M$, GMD, and GSD are reported. It can be seen that a value near 0.8 provides the most accurate results, depending on the variable of the main interest. By comparing the errors of $N$ and $M$ produced by the PL+LN model with the errors produced by different models reported in Tab. 3, it can be seen that the lowest errors for $N$ and $M$ simultaneously are produced by the PL+LN model regardless of the value of $\gamma$ used. Contour plots with different values of $\gamma$ are shown in Fig. 15. By visual inspection, it can be seen that the values 0.5 and 0.8 produce the contour plots being the closest to the measured contour plot in Fig. 11.

## 5    Conclusions

The combined power law and log-normal distribution (PL+LN) model was developed to represent a particle size distribution in simultaneous new particle formation and growth situation, in which log-normal distributions do not represent the aerosol sufficiently well. The PL+LN distribution combines a power law form typical to simultaneous new particle formation and growth situation at the initial steps of aerosol formation with a log-normal form typical to aged aerosols. The PL+LN model is useful in simulations involving the initial steps of aerosol formation where a sectional representation of the size distribution causes too high computational cost, such as in multidimensional simulations or in the case of using inverse modelling to obtain the best estimates for parameters used as input in the model. These parameters can be, e.g., the new particle formation rate or the condensational growth rate that the most accurately produce the distributions as the measured ones. The model uses six moment variables to model the distribution, denoting lower memory consumption compared to sectional models which require tens or hundreds variables. The model includes simultaneous new particle formation, condensation, coagulation, coagulational loss, and depositional loss processes.

The PL+LN model was evaluated using theoretical test simulations and a real-world particle formation event simulation. The test cases represented particle formation events with the parameters related to the atmosphere and to vehicle exhaust. The real-world case was the simulation of a particle formation event measurement performed in a mobile aerosol chamber at Mäkelänkatu street canyon measurement site in Helsinki, Finland. The evaluation was done against highly accurate sectional models using fixed-sectional and moving-center fixed-sectional methods. The accuracy of the total number, surface area, and mass concentrations simulated by the PL+LN model was examined: the relative errors of the concentrations were lower than $2\,\%$ compared to the highly accurate sectional models, with the exception of a theoretical test case having size-dependent condensational growth rate with which the relative errors were up to $17\,\%$ due to the shape of the size distribution produced. The performance of producing geometric mean diameter (GMD) and geometric standard deviation (GSD) of the total distributions using different models was also examined: the highest relative error with the PL+LN model was $3.5\,\%$ for GMD when size-dependent new particle formation rate was modelled in a theoretical test case. The shapes of the distributions produced by the PL+LN model were noticeably more similar to the reference distributions than produced by a simple log-normal distribution model.

Considering the same computing time as the PL+LN model in the chamber event simulation, only 35 size sections for a fixed-sectional model and 10 size sections for a moving-center fixed-sectional model were allowed to be modelled. With these

section numbers, the results for the number and mass concentrations, for GMD, and for GSD were not as accurate as using the

PL+LN model: the relative errors were up to tens of percent. Additionally, a simple log-normal distribution model seemed to output GMD relatively well in this case, but the number concentration was overestimated and GSD was underestimated during almost the whole time domain, especially at times when new particle formation and growth occurred simultaneously.

The new particle formation rates, $J(t)$, and the condensational growth rates, $g(t)$, used in the chamber simulation were obtained through inverse modelling. Firstly, the PL+LN and the LN models were used to obtain the best estimates for $J(t)$

and $g(t)$ that produce the measured concentrations the most accurately. Secondly, the time series of $J(t)$ and $g(t)$ obtained using the PL+LN model were used as initial guesses in the inverse modelling with the highly accurate models. Only minor differences were found in the time series of $J(t)$ and $g(t)$ obtained using different models: the PL+LN model overestimates and the LN model underestimates $J(t)$ slightly. The associated computing times using the highly accurate sectional models are approximately 2 orders of magnitude longer compared to the PL+LN model. Therefore, the PL+LN model provides a rapid

and accurate solution to obtain input parameters, such as new particle formation and condensational growth rates, from the measured data through inverse modelling.

## Appendix A: Gaussian quadratures

The Hermite-Gauss quadrature (Steen et al., 1969) is used in the integrals involving the LN distribution, as the density function of the LN distribution, Eq. (8), is in the form of the weight function of the Hermite-Gauss quadrature $e^{-x^2}$. E.g., an integral

$$\int_{-\infty}^{\infty} D_{\mathrm{p}}^2 g(t, D_{\mathrm{p}}) \left. \frac{\mathrm{d}N}{\mathrm{d}\ln D_{\mathrm{p}}} \right|_{\mathrm{LN}} \mathrm{d}\ln D_{\mathrm{p}} = \frac{N_{\mathrm{LN}}}{\sqrt{2\pi}\ln\sigma} \int_{-\infty}^{\infty} D_{\mathrm{p}}^2 g(t, D_{\mathrm{p}}) \exp\left[ -\frac{\ln^2\left(D_{\mathrm{p}}/D_{\mathrm{g}}\right)}{2\ln^2\sigma} \right] \mathrm{d}\ln D_{\mathrm{p}} \tag{A1}$$

becomes, using the Hermite-Gauss quadrature,

$$\frac{N_{\mathrm{LN}}}{\sqrt{\pi}} \sum_{j=1}^{n} w_j D_{\mathrm{p}}^2 g(t, D_j) \tag{A2}$$

where $D_j$ and $w_j$ are the abscissa and the weight for the bin $j$ obtained from the quadrature, and $n$ is the degree of the quadrature. In this article, the degree of $n = 5$ is used for the LN distribution denoting that the integrals are calculated with

five diameter values. The integrals involved in the PL distribution are in the form of $D_{\mathrm{p}}^\alpha$ which is not a weight function of any specific quadrature; therefore, a Gaussian quadrature for this purpose was developed. E.g., an integral

$$\int_{-\infty}^{\infty} D_{\mathrm{p}}^2 g(t, D_{\mathrm{p}}) \left. \frac{\mathrm{d}N}{\mathrm{d}\ln D_{\mathrm{p}}} \right|_{\mathrm{PL}} \mathrm{d}\ln D_{\mathrm{p}} = \frac{N_{\mathrm{PL}}\alpha}{D_2^\alpha - D_1^\alpha} \int_{\ln D_1}^{\ln D_2} D_{\mathrm{p}}^2 g(t, D_{\mathrm{p}}) D_{\mathrm{p}}^\alpha \, \mathrm{d}\ln D_{\mathrm{p}} \tag{A3}$$

becomes, using the quadrature developed here,

$$\frac{N_{\mathrm{PL}}\ln d^\alpha}{d^\alpha - 1} \sum_{j=1}^{n} w_j D_{\mathrm{p}}^2 g(t, D_j) \tag{A4}$$

where $D_j$ and $w_j$ are the abscissa and the weight for the bin $j$ obtained from the quadrature. The degree of $n = 4$ is used for the PL distribution.

The degree of the quadrature developed here, $n = 4$, is relatively low, but by using higher degrees, internal equations in the quadrature will become more complicated which results in increased computing time. In the cases described in this article, the degree of 4 results in the absolute relative errors of the condensation and the coagulation terms less than $10^{-2}$ compared to a very high degree numerical integration or, if exists, to an analytical solution, but only when $\alpha > 0.5$, in the case of the condensation terms, and when $d < 3$, in the case of the coagulation terms. A drawback of the quadrature with a low degree is that with the values of $\alpha < 0.5$ (condensation) and $d > 3$ (coagulation), errors increase, which causes numerical problems during a simulation. Therefore, the quadrature is used only with the values of $\alpha < 0.5$ (condensation) or $d < 3$ (coagulation), but numerical integration as in Eq. (23) otherwise, or, if exists, an analytical solution for the condensation terms as in Eq. (22). In numerical integration, $n = 200$ size sections are used for condensation calculation, which is required to produce the absolute relative errors less than $10^{-2}$. Calculating coagulation using numerical integration, $n = 20$ is used. However, with only 20 size sections, relative errors increase up to $30\,\%$ when $d$ increases towards 100 and $\alpha$ towards $\pm 5$. Nevertheless, a higher degree is not used due to increasing computing time, which is squarely proportional to the number of size sections due to double integrals in the coagulation terms. Therefore, the degrees are kept low to maintain computational efficiency. To produce the same accuracy by using numerical integration with $n$ size sections as by using the quadrature with the degree of 4 for the PL distribution, with the values of $\alpha$ and $d$ where the quadrature is applicable, about 1 or 2 orders of magnitude longer computing time is consumed. The degree for the Hermite-Gauss quadrature, $n = 5$, can be increased easily without encountering steep increases to computing time; nevertheless, in the cases of this article, $n = 5$ produces the absolute relative errors of less than $10^{-4}$ for the condensation and the coagulation terms.

## Appendix B:  Processing the experimental data

The aerosol sample was measured using Airmodus Particle Size Magnifier (PSM), TSI Ultrafine Condensation Particle Counter (CPC), TSI Nano Scanning Mobility Particle Sizer (Nano-SMPS), TSI Engine Exhaust Particle Sizer (EEPS), and Dekati Electrical Low-Pressure Impactor (ELPI+). PSM in fixed saturator flow setting detects particles with the diameters of higher than about $1.6\,\mathrm{nm}$ (Vanhanen et al., 2011), CPC higher than about $3.6\,\mathrm{nm}$ (Mordas et al., 2008), and Nano-SMPS from about 7 to 64 nm, with the detection efficiency of $50\,\%$ or higher. These cut diameters, $D_{\mathrm{PSM}} = 1.6\,\mathrm{nm}$, $D_{\mathrm{CPC}} = 3.6\,\mathrm{nm}$, and $D_{\mathrm{Nano\text{-}SMPS}} = 7\,\mathrm{nm}$, are used to combine the data of PSM, CPC, and Nano-SMPS to obtain total aerosol size distributions for the diameter range of $1.6 - 64\,\mathrm{nm}$ with

$$\left.\frac{\mathrm{d}N}{\mathrm{d}\ln D_{\mathrm{p}}}\right|_{\text{measured}} = \begin{cases} \frac{\max\{N_{\mathrm{PSM}}-N_{\mathrm{CPC}},0\}}{\ln(D_{\mathrm{CPC}}/D_{\mathrm{PSM}})}, & D_{\mathrm{PSM}} \leq D_{\mathrm{p}} < D_{\mathrm{CPC}} \\[2mm] \frac{\max\{N_{\mathrm{CPC}}-N_{\mathrm{Nano\text{-}SMPS}},0\}}{\ln(D_{\mathrm{Nano\text{-}SMPS}}/D_{\mathrm{CPC}})}, & D_{\mathrm{CPC}} \leq D_{\mathrm{p}} < D_{\mathrm{Nano\text{-}SMPS}} \,, \\[2mm] \left.\frac{\mathrm{d}N}{\mathrm{d}\ln D_{\mathrm{p}}}\right|_{\text{Nano\text{-}SMPS}}, & D_{\mathrm{p}} \geq D_{\mathrm{Nano\text{-}SMPS}} \end{cases} \tag{B1}$$

where $N_{\text{PSM}}$, $N_{\text{CPC}}$, and $N_{\text{Nano-SMPS}}$ are the total number concentrations measured by the devices, and $\left.\frac{\mathrm{d}N}{\mathrm{d}\ln D_{\mathrm{p}}}\right|_{\text{Nano-SMPS}}$ is the particle size distribution measured by Nano-SMPS. The maximum functions prevent the size distribution to become negative. Before the concentrations were input into Eq. (B1), the concentrations output by the devices were synced. Because particle

sizes are well within the range of high detection efficiency of all three devices after the particle formation event, the device outputs would be equal in that moment if the maximum detection efficiencies of all the devices were equal. But because there are differences in the maximum detection efficiencies and the time responses of the devices, the output concentrations were multiplied and the output time vectors were synced so that all the time series of the concentrations are overlapping after the event. Nano-SMPS measures diameters down to 2 nm, but due to its low accuracy for those diameters, all Nano-SMPS data

below 7 nm was neglected. EEPS and ELPI+ having time resolutions of only 1 s were used to ensure the stability of the aerosol distribution during a Nano-SMPS scan lasting 150 s: no rapid changes in the aerosol distribution were observed in the time scales shorter than 150 s.

Initially, the aerosol in the chamber consisted of a background aerosol mode with CMD of 15 nm and the concentration of about 4000 cm$^{-3}$, according to the Nano-SMPS data shown in Fig. 11. No major changes in the distribution were observed until

the UV lights were switched on ($t = 0$ s). After switching the UV lights on, a nucleation mode begins to from, which is seen as the appearance of new particles at small particle diameters. It can be also seen that small particles exist though the growth process proceeds, which implies continuing new particle formation. The total distributions were altered from a power law-shape towards a log-normal-shape. After about 500 s, particle concentration finished increasing (Fig. 5), which occurs because the gaseous precursors initiating new particle formation began to expire. The decreasing trend of particle number concentration

after 500 s was accounted by coagulation and deposition. The particles of the nucleation mode and of the background mode grew to about 25 nm and to about 60 nm during the event, respectively. The total number, surface area, and mass concentrations of the measured nucleation mode were calculated from the total measured size distributions, Eq. (B1), subtracted by log-normal distributions fitted to the background aerosol, assuming spherical particles with the density of 1.4 g cm$^{-3}$. GMD and GSD of the measurement data were calculated also from the nucleation mode size distribution, using Eqs. (47) and (48), although the

accuracy of the measured distribution for the diameters below the Nano-SMPS measurement range is poor.

*Acknowledgements.* This work was funded by the Maj and Tor Nessling Foundation (project number 2014452), by Tampere University of Technology Graduate School, and by the Finnish Funding Agency for Technology and Innovation (Tekes) as a part of the CLEEN MMEA program. Authors acknowledge the personal of the Air Protection Group at the Helsinki Region Environmental Services Authority (HSY) for enabling the measurement campaign at the HSY's street canyon measurement site in Helsinki.

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

**Table 1.** Input parameters for the test cases. Case names with Atm have the parameter sets related to atmospheric particle formation and the Exh case related to particle formation occurring in vehicle exhaust. $J$ and $g$ are the new particle formation rate and the condensational growth rate, respectively. $N_{bg}$ is the concentration of the background aerosol distribution having a count median diameter of $CMD_{bg}$. The coagulational loss exponent $l_{bg}$ depend on the value of $CMD_{bg}$. Wall deposition is modelled using the deposition coefficient $k_{dep}$. The length of the simulated time domain is $t_{max}$.

| Case | $J$ $(\mathrm{cm^{-3}\,s^{-1}})$ | $g$ | $T$ (K) | Coagulation | $N_{bg}$ $(\mathrm{cm^{-3}})$ | $CMD_{bg}$ (nm) | $l_{bg}$ | $k_{dep}$ | $\gamma$ | $t_{max}$ |
|------|------|-----|---------|-------------|-------|--------|----------|-----------|----------|-----------|
| Atm1 | 0.1 | $1\,\mathrm{nm\,h^{-1}}$ | 280 | intra, inter | 0 | – | – | 0 | 0 | 5 h |
| Atm2 | 0.1 | $1\,\mathrm{nm\,h^{-1}}$ | 280 | intra, inter | 0 | – | – | $1.8\,\mathrm{nm\,h^{-1}}$ | 0 | 5 h |
| Atm3 | 0.1 | $1\,\mathrm{nm\,h^{-1}}$ | 280 | intra, inter, bg | $10^3$ | 100 | -1.6 | $1.8\,\mathrm{nm\,h^{-1}}$ | 0 | 5 h |
| Atm4 | Eq. (50) | $1\,\mathrm{nm\,h^{-1}}$ | 280 | intra, inter, bg | $10^3$ | 100 | -1.6 | $1.8\,\mathrm{nm\,h^{-1}}$ | 0 | 5 h |
| Atm5 | Eq. (50) | Eqs. (16) – (18) | 280 | intra, inter, bg | $10^3$ | 100 | -1.6 | $1.8\,\mathrm{nm\,h^{-1}}$ | 0.25 | 5 h |
| Exh | $10^8$ | $5\,\mathrm{nm\,s^{-1}}$ | 500 | intra, inter, bg | $10^6$ | 60 | -1.5 | $0.07\,\mathrm{nm\,s^{-1}}$ | 0 | 1 s |

**Table 2.** Relative errors, $\delta_X$ (%), of the variables at the ends of the test case simulations using the PL+LN model compared to the variables produced by the fixed-sectional model with 1000 size sections. The input parameter sets are shown in Tab. 1. $N$, $S$, and $M$ are the number, the surface area, and the mass concentration of the total particle distribution, respectively. GMD and GSD are the geometric mean diameter and the geometric standard deviation of the distribution.

| Case | $N$ | $S$ | $M$ | GMD | GSD |
|------|-----|-----|-----|-----|-----|
| Atm1 | -0.001 | -0.310 | -0.573 | +0.068 | -0.193 |
| Atm2 | -0.052 | -0.481 | -0.838 | -0.083 | -0.039 |
| Atm3 | +0.173 | -0.884 | -1.296 | -0.086 | -0.505 |
| Atm4 | +0.202 | -0.816 | -1.518 | +0.933 | -1.251 |
| Atm5 | +6.957 | -2.384 | -7.666 | -3.511 | -2.879 |
| Exh | +0.007 | -0.356 | -0.680 | +0.084 | +0.011 |

**Table 3.** Computational costs of different models and relative errors of number ($N$) and mass ($M$) concentrations obtained from the time of 1663 s after the UV lights were switched on in the chamber simulation. The number of variables compared to size sections in MC models is 2-fold because the centers of the size sections need to be stored in addition to the concentrations of the sections. Relative values are calculated using a model with (ref.) as the reference model.

| Model name | Method | Size sections | Variables | Relative computing time | Error in $N$ (%) | Error in $M$ (%) |
|---|---|---|---|---|---|---|
| FS35 | fixed-sectional | 35 | 35 | 1.0 | +1.6 | +79 |
| MC10 | moving-center | 10 | 20 | 1.0 | +18 | +29 |
| LN | log-normal | - | 3 | 0.09 | +17 | +6.4 |
| PL+LN | combined PL and LN | - | 6 | 1.0 (ref.) | +0.48 | -1.2 |
| FS400 | fixed-sectional | 400 | 400 | 170 | 0 (ref.) | +3.4 |
| MC100 | moving-center | 100 | 200 | 200 | +0.31 | 0 (ref.) |

**Table 4.** Relative errors (%) of the variables in the chamber simulation with the PL+LN model using different values for the condensational transfer factor $\gamma$. The errors are compared to the FS400 model, except for $M$ that is compared to the MC100 model.

| $\gamma$ | $N$ | $M$ | GMD | GSD |
|---|---|---|---|---|
| 0.1 | -1.2 | -7.8 | +2.6 | -5.8 |
| 0.5 | +0.75 | -2.7 | -3.7 | +3.4 |
| 0.8 | +0.48 | -1.2 | -0.33 | -2.8 |
| 0.9 | +2.6 | -0.43 | -0.11 | -6.0 |

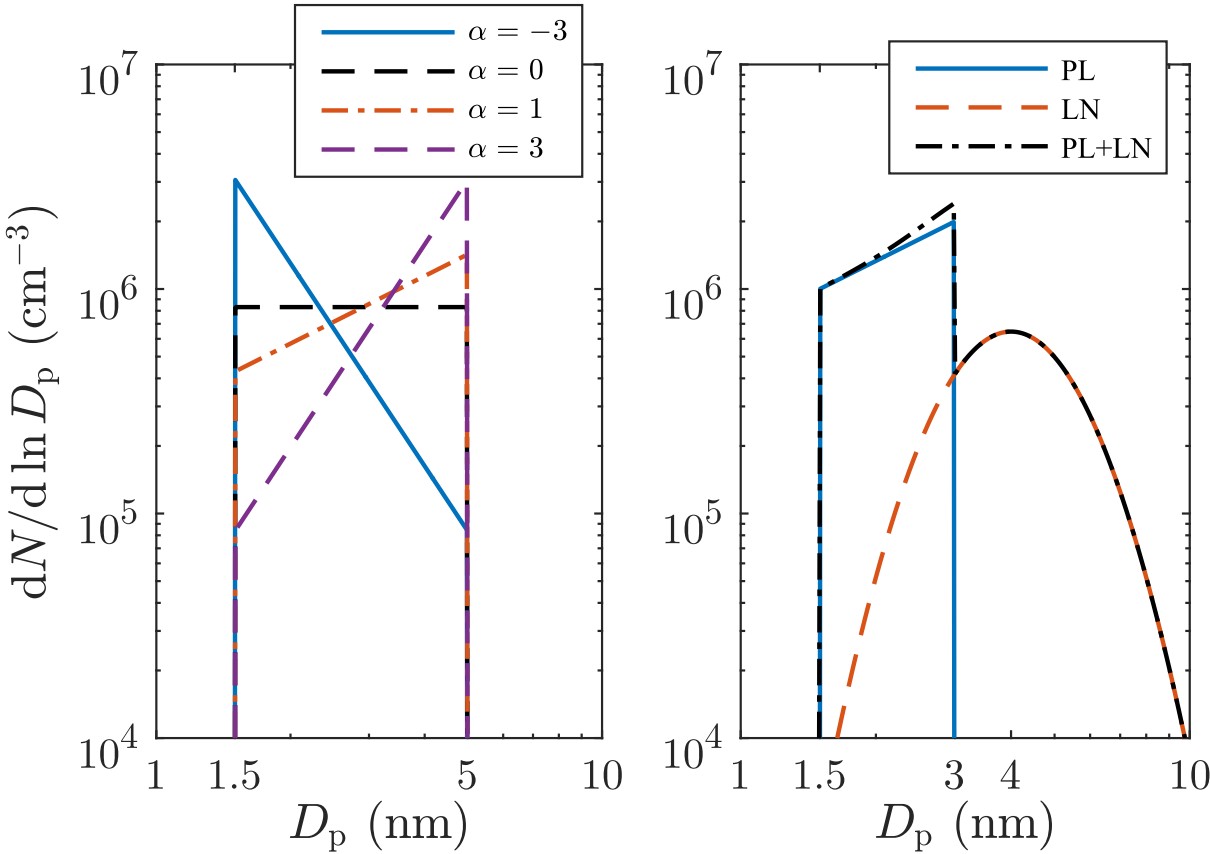

**Figure 1.** Left pane shows examples of power law distributions with different values of the slope parameter $\alpha$. Right pane shows the combination of a power law (PL) and a log-normal (LN) distribution (PL+LN).

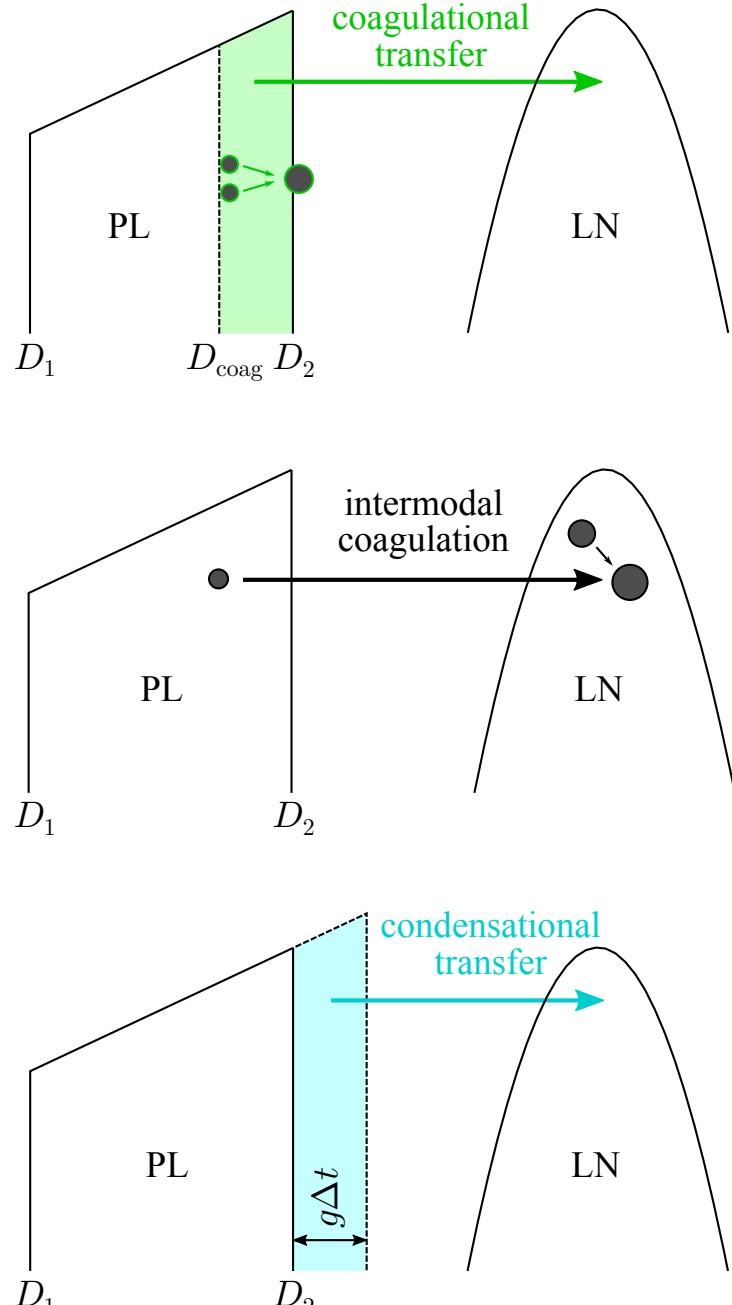

**Figure 2.** Intermodal processes between the PL and the LN distributions. Particles larger than $D_{\mathrm{coag}}$ (green area) form particles larger than $D_2$ by the intramodal coagulation in the PL distribution; the coalesced resultant particles are transferred to the LN distribution. When the LN distribution exists, particles of the both distributions begin to coagulate intermodally; the resultant particles are assigned to the LN distribution. Condensation grows the largest particle diameter by $g\Delta t$ in a time step of $\Delta t$, but the condensational transfer transfers a part of the particles larger than $D_2$ (blue area) to the LN distribution.

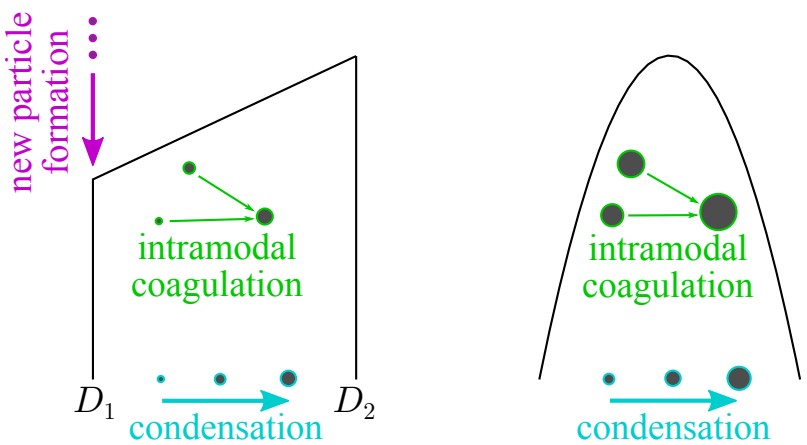

**Figure 3.** Intramodal processes. New particle formation forms particles with the diameter of $D_1$ to the PL distribution. Condensation and intramodal coagulation grow particles within a distribution.

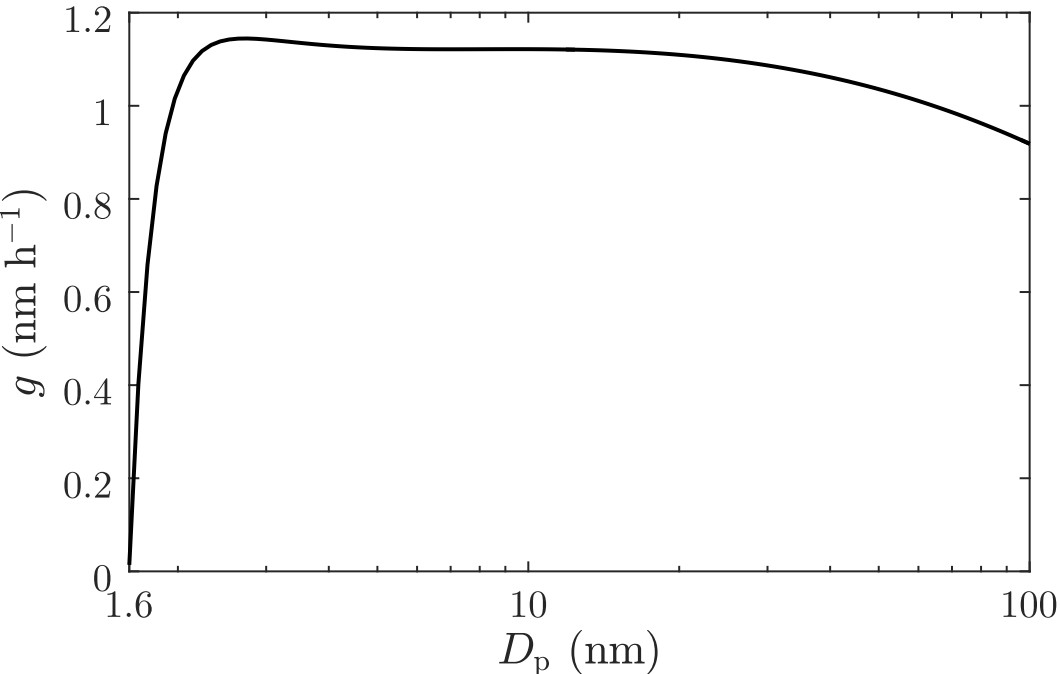

**Figure 4.** Size-dependent condensational growth rate of sulfuric acid-water particles with the sulfuric acid vapor concentration of $0.8 \times 10^7 \, \text{cm}^{-3}$, temperature of $280 \, \text{K}$, and relative humidity of $60 \, \%$ as a function of the particle diameter, used in the Atm5 case.

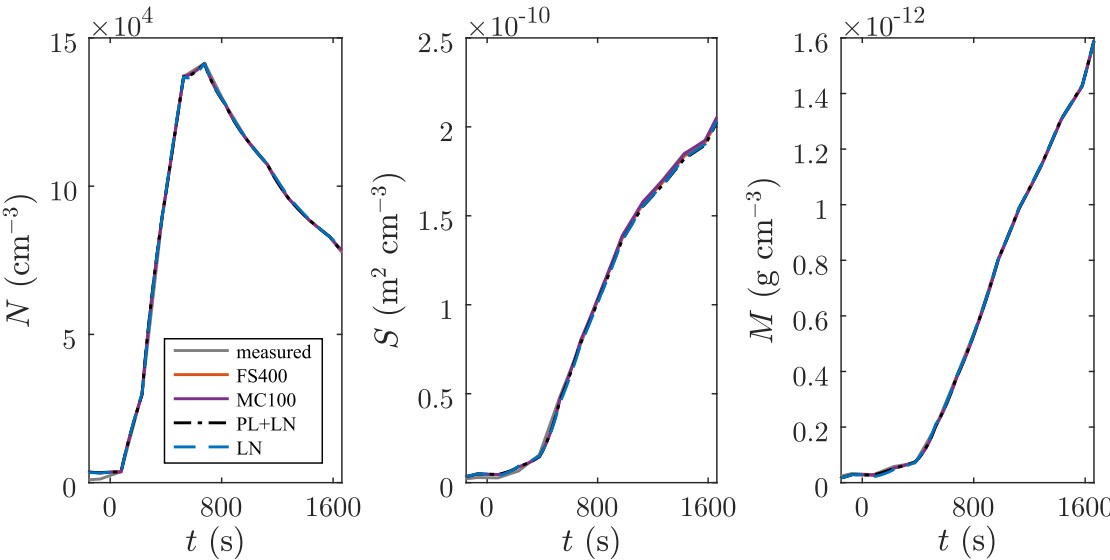

**Figure 5.** Number ($N$), surface area ($S$), and mass ($M$) concentrations of the nucleation mode in the chamber event. The measured concentrations and the concentrations produced during inverse modelling using different models are nearly equal. The data are shown for the centers of the Nano-SMPS scans only, because those values only are used in inverse modelling. FS400 denotes the fixed-sectional model with 400 size sections and MC100 denotes the moving-center fixed-sectional model with 100 size sections.

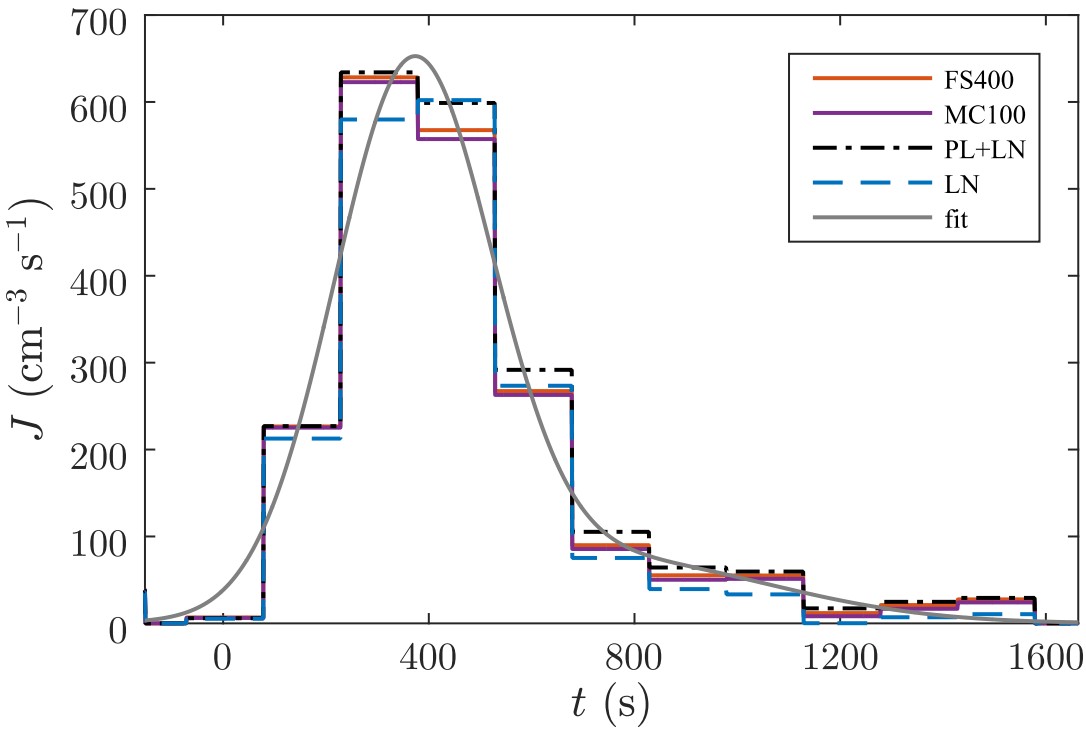

**Figure 6.** Time series for the new particle formation rates in the chamber event that produce the measured concentrations, $N$, $S$, and $M$, the most accurately compared to the measured ones, using different models. The fit denotes a bell-shaped function fitted to the values from the FS400 model.

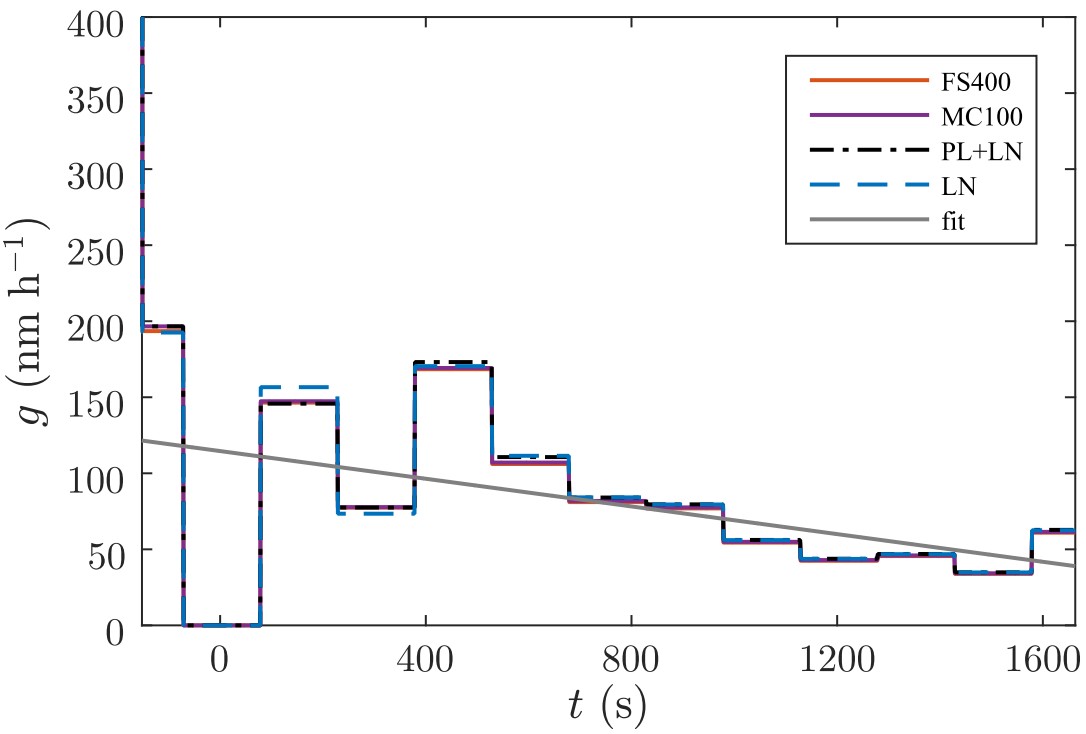

**Figure 7.** Time series for the condensational growth rates in the chamber event that produce the measured concentrations, $N$, $S$, and $M$, the most accurately compared to the measured ones, using different models. The fit denotes a linear function fitted to the values from the FS400 model.

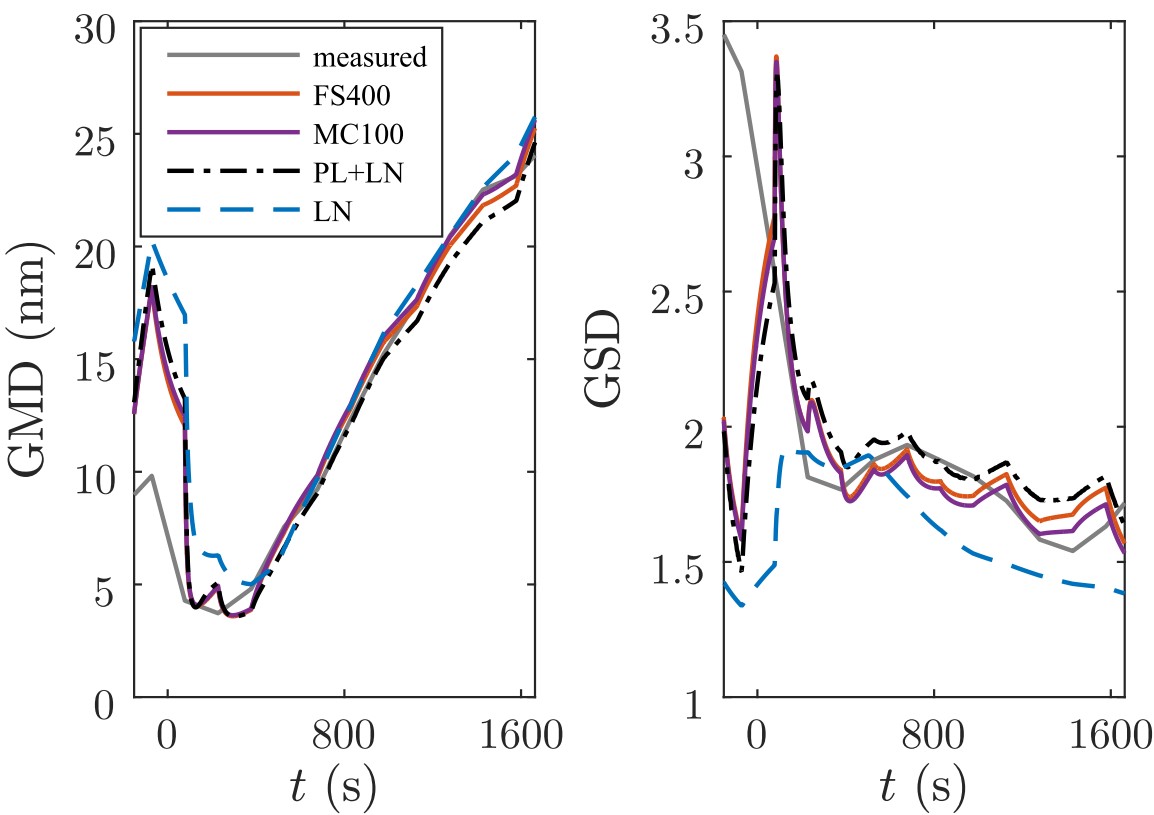

**Figure 8.** Geometric mean diameter (GMD) and geometric standard deviation (GSD) of the nucleation mode in the chamber event, obtained through inverse modelling, using different models.

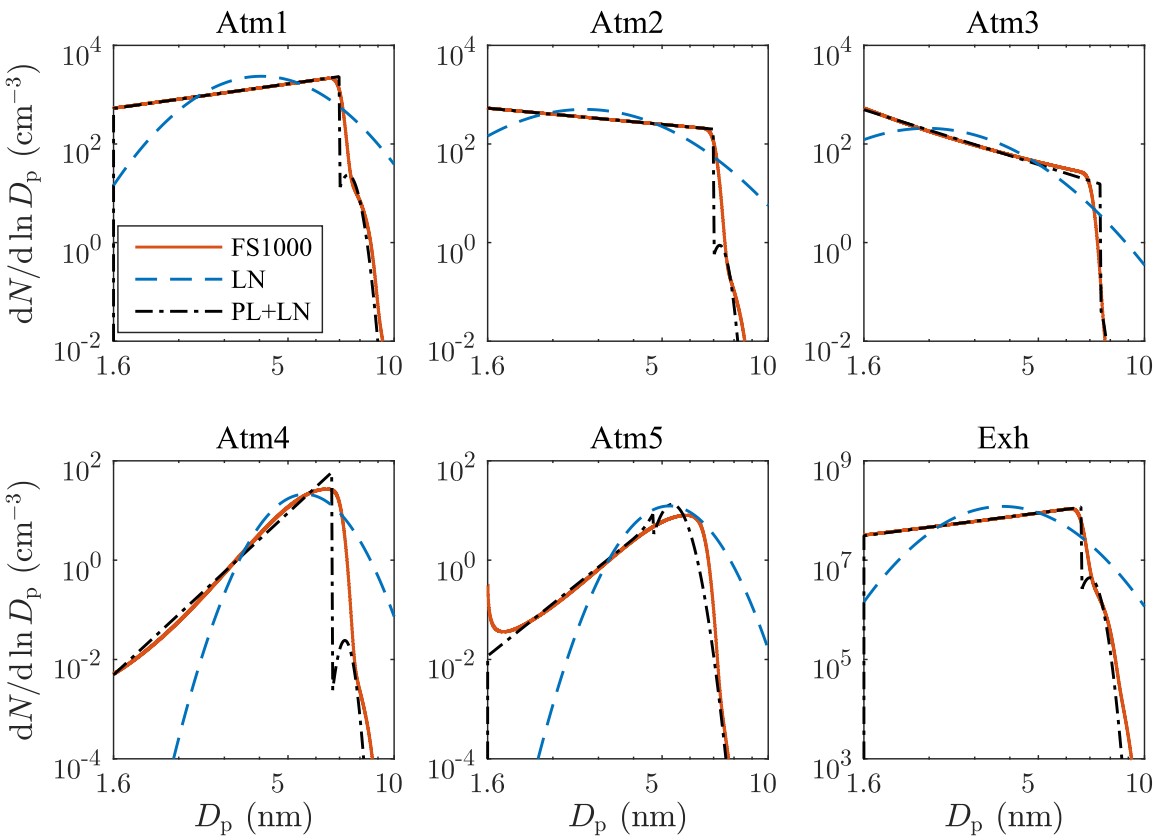

**Figure 9.** Particle size distributions at the ends of the test case simulations produced by different models. The input parameter sets are shown in Tab. 1. FS1000 denotes the fixed-sectional model with 1000 size sections. Note the different scales in the vertical axes on the bottom row.

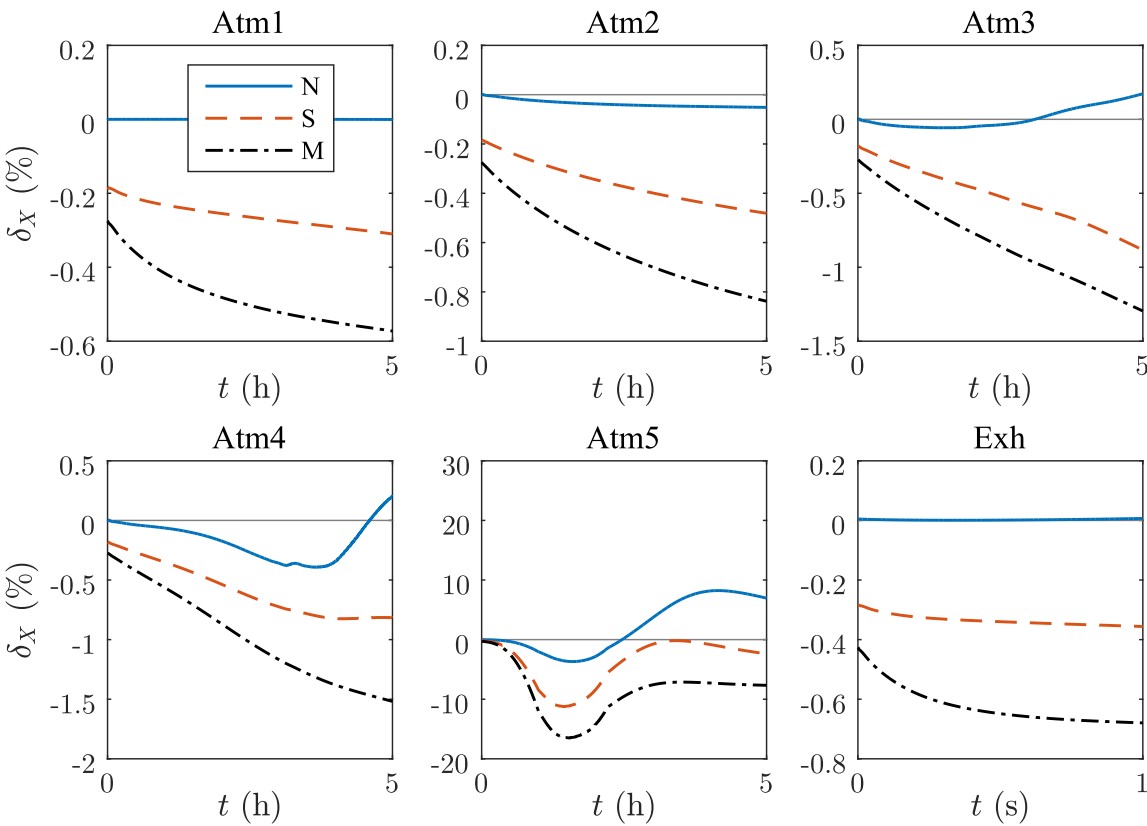

**Figure 10.** The relative errors of the moments ($\delta_X$) in the test cases produced by the PL+LN model. The input parameter sets are shown in Tab. 1.

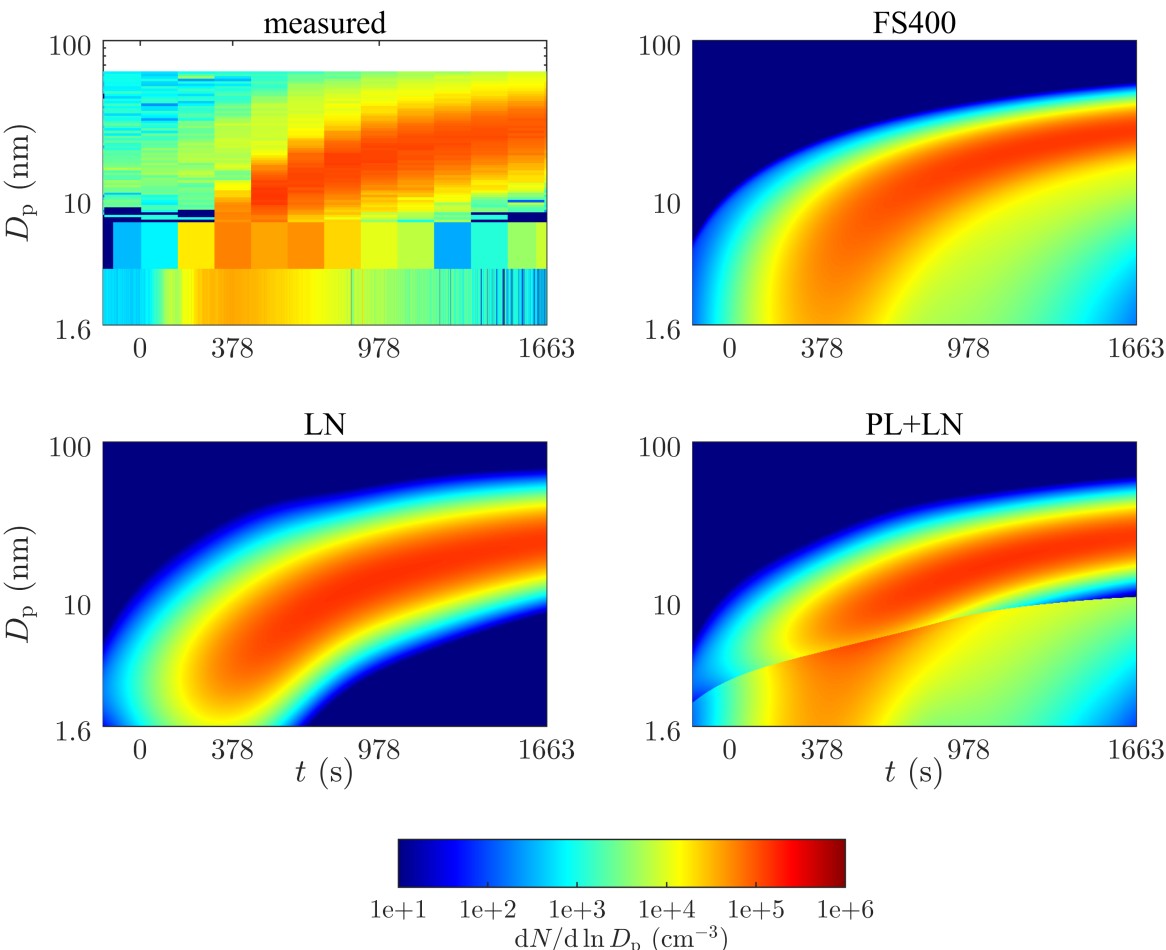

**Figure 11.** Contour plots of particle distributions measured by Airmodus Particle Size Magnifier (PSM), TSI Ultrafine Condensation Particle Counter (CPC), and TSI Nano Scanning Mobility Particle Sizer (Nano-SMPS) and simulated by different models in the chamber event. The value of 0.8 was used for $\gamma$ with the PL+LN model. The UV lights were switched on at time $t = 0\,\mathrm{s}$. Note that the background particle distribution seen in the measured data was excluded from the simulations.

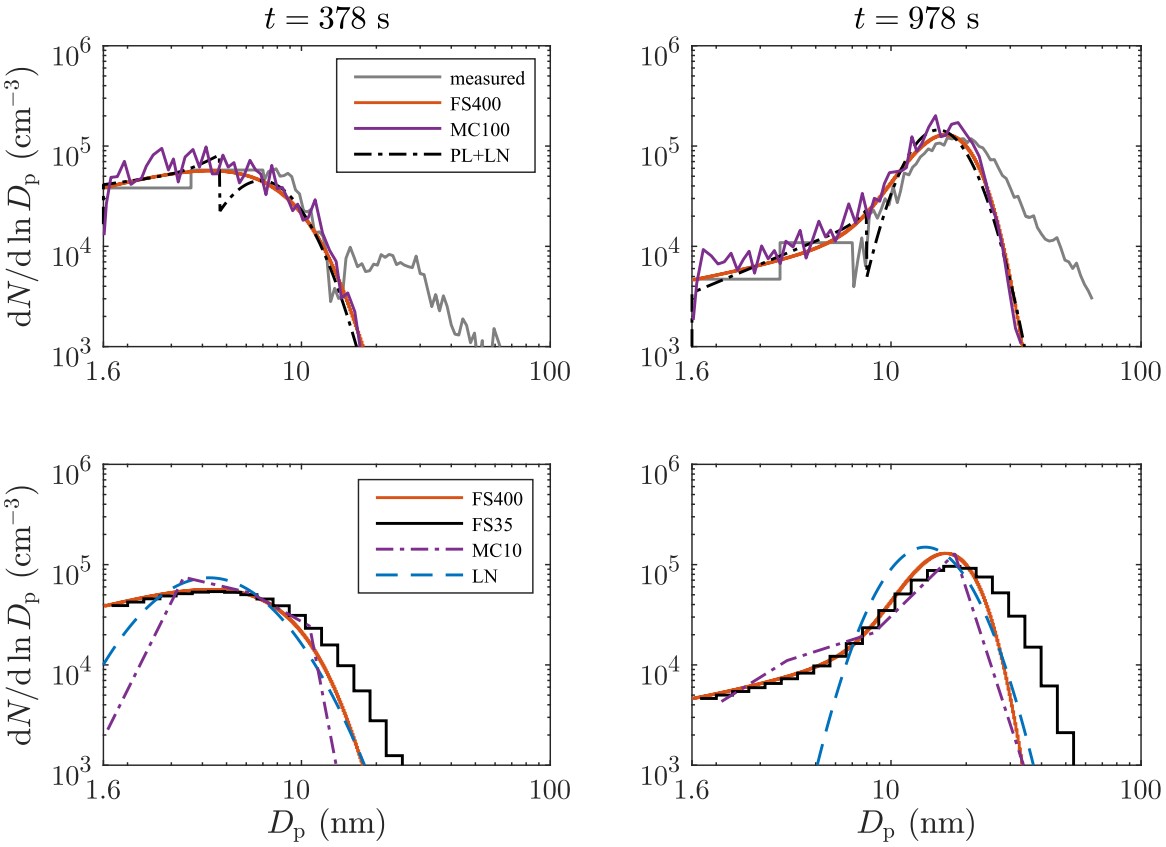

**Figure 12.** Particle size distributions in the chamber event 378 and 978 s after the UV lights were switched on. The top row shows the accurate model outputs together with the measured distribution. The bottom row shows the less accurate model outputs together with the accurate FS400 model output. The measured distributions include also the background distributions around 30 and 50 nm which were excluded from the simulations. The abbreviations are explained in Tab. 3.

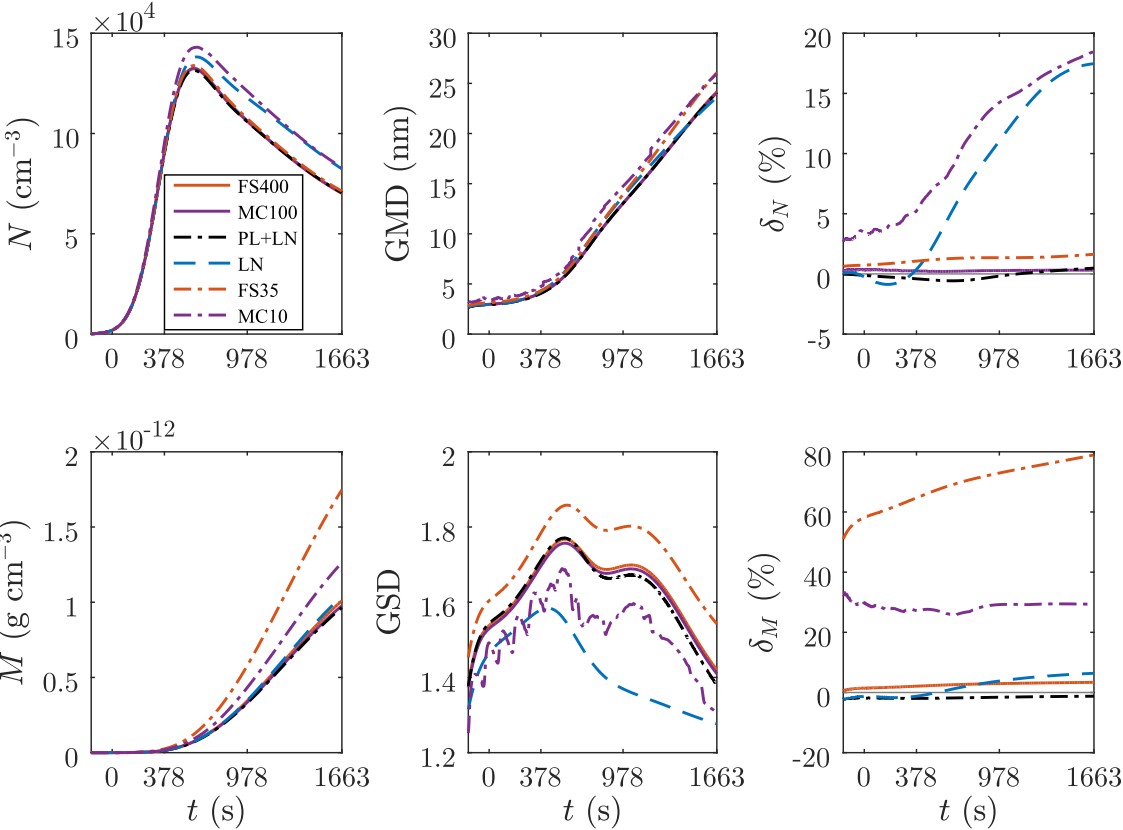

**Figure 13.** Number ($N$) and mass ($M$) concentrations, GMD, and GSD of the nucleation mode and the relative errors of the concentrations ($\delta_N$ and $\delta_M$) in the chamber event, produced by different models. The outputs of the FS400 and the MC100 models are nearly equal, and thus they are difficult to distinguish in the figure.

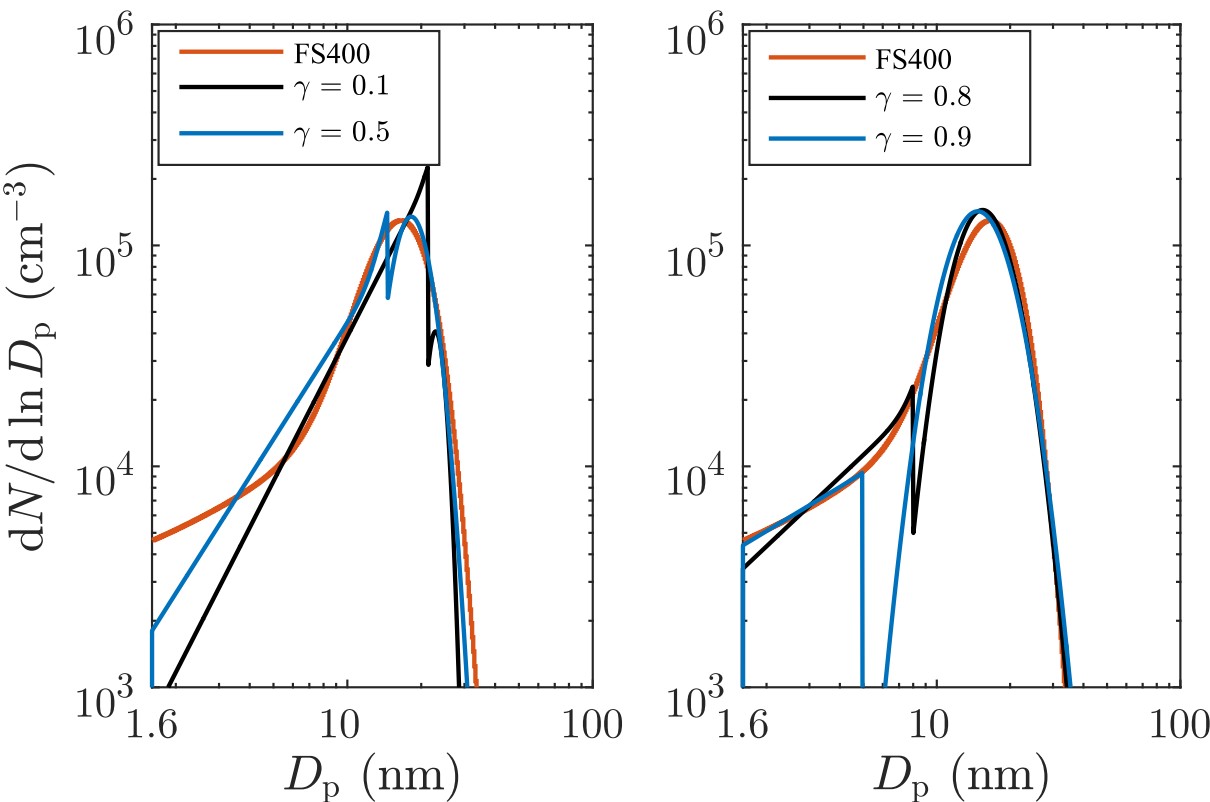

**Figure 14.** Particle size distributions 978 s after the UV lights were switched on, with the different values for the condensational transfer factor $\gamma$ using the PL+LN model compared to the FS400 model.

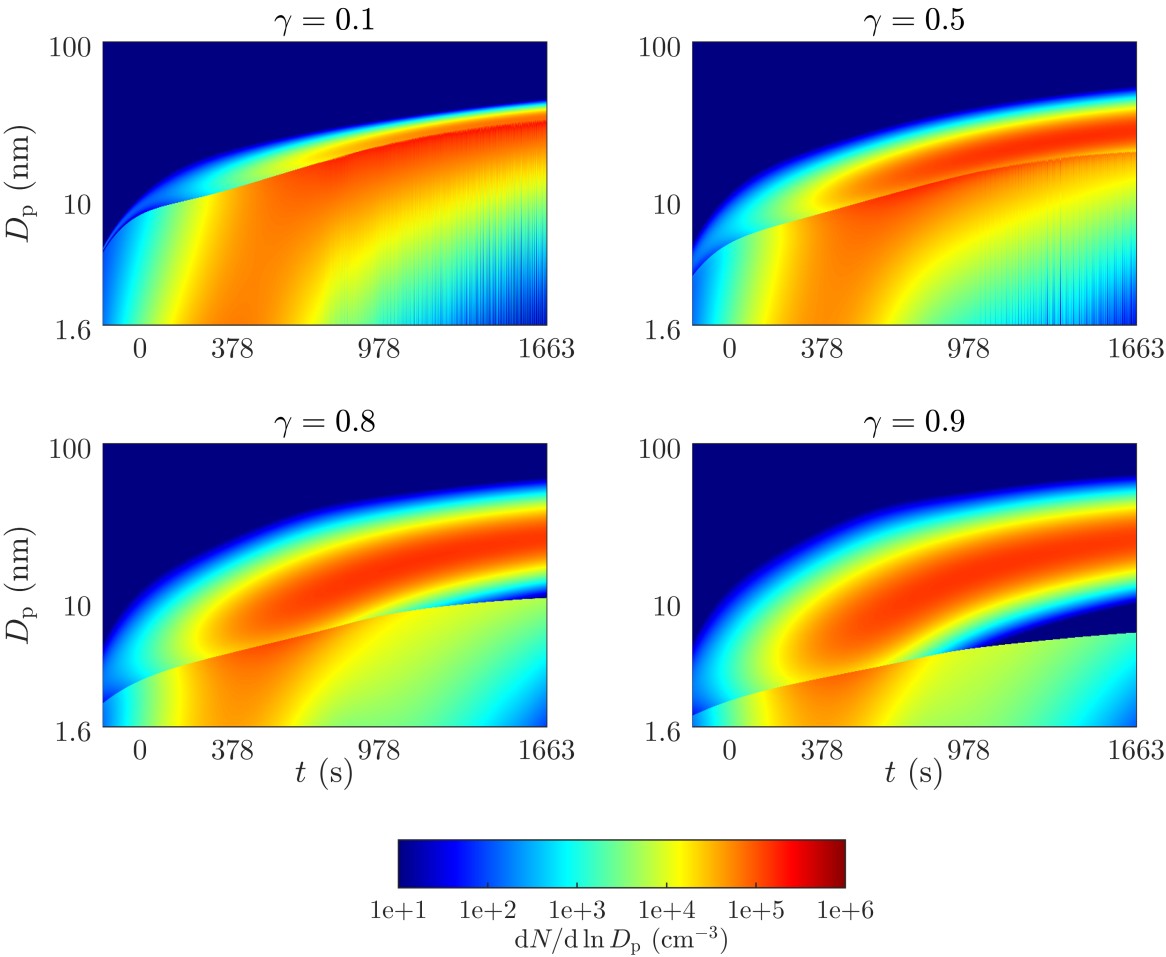

**Figure 15.** Contour plots of particle distributions simulated by the PL+LN models with different values of $\gamma$ in the chamber event.