# Peer review of "Using a combined power law and log-normal distribution model to simulate particle formation and growth in a mobile aerosol chamber"

_Atmospheric Chemistry and Physics, 2015_

## Referee Comment (RC1) · Anonymous Referee #1 · 16 Feb 2016

Review of the manuscript:

In their manuscript, "Using a combined power law and log-normal distribution model to simulate particle formation and growth in a mobile aerosol chamber", M. Olin, T. Anttila and M. Dal Maso have described a novel method for simulating the dynamics of aerosol particles, with the emphasis on simulating the early growth of a freshly-formed particle mode in a computationally cost-efficient manner. The authors have tested their model against previous aerosol dynamics models in a few simplified test scenarios to estimate the accuracy of the novel model, and also to demonstrate both the computational efficiency of the novel model in comparison to more accurate models and the accuracy of the novel model in comparison to other models with similar computational burden. Finally, the authors have used the model to reproduce a new particle formation and growth event as observed in aerosol chamber measurements. The main conclusions of the paper are that the newly-developed model is able to provide concentration, surface area and mass concentrations within a few percent to those obtained with a highly accurate model, and that the novel model is able to represent simultaneous new particle formation and particle growth, which is beyond purely log-normal model, which is often used for cost-efficient representation of an aerosol population.

Based on the results presented in the manuscript, the new model seems like a useful compromise between accurate and computationally cost-efficient representations of a particle population, and the manuscript is in the scope of ACP journal. A more thorough evaluation of the model is needed, however, and the language of the manuscript should be revised.

General comments:

The main objectives of the manuscript are: 1) to describe the new model in detail, 2) to evaluate the accuracy and computational cost-efficiency of the model, and 3) to demonstrate the applicability of the model using a real life example. I find that the first objective is covered quite well in the manuscript. The second objective, however, is not covered sufficiently: the new model is evaluated against more accurate models in a handful of scenarios, but a more thorough examination is needed. The main issue is that only size-independent growth rates are considered. Another issue is the parameter $\gamma$, which is left as a free parameter in the model, but relatively little consideration is given to how sensitive the model results are to the choice of $\gamma$, or, especially, how to choose the value of $\gamma$ for given simulation conditions. These, and some other issues, are described in more detail in the "Specific comments", below. Regarding the third objective, a single example of reproducing the time evolution of a particle size distribution during a new particle formation and growth event as observed in a chamber experiment is provided. A few different examples would probe the capabilities of the model much more comprehensively. Even more problematic, however, is that in this example, the particle formation and growth rates representing the measurement conditions are obtained with inverse modelling using another, more accurate, model, and those values are used as input in the novel model with the results being compared to the results from the more accurate model. In other words, the only connection to the real life measurement is that the formation and growth rates used as inputs in the models represent the measurement conditions, but otherwise there is not much difference to the scenarios used to evaluate the model. A more suitable demonstration of the usage of the model would be, for example, to estimate the formation and growth rates

representing the measurements using inverse modelling with both the accurate model and the novel model, and then compare those values. Finally, the language of the manuscript is not very good at times. The grammatical errors are not exhaustively listed in the "Technical corrections", and a greater care for punctuation, for example, would be needed.

Specific comments:

1. The title of the manuscript implies that the new model is able to simulate the actual process of particle formation, which is misleading, as the formation rate is used as an input in the model, and also that the emphasis of the manuscript would be on simulations related to the mobile aerosol chamber, but only one example is given. The title should be revised.

2. On page 7, lines 19-23, it is stated that the mass growth rate is assumed to be proportional to $D_p^2$ in the model, even though only one of the three conditions justifying that assumption is met. Then, on page 8, line 1, it is stated that this assumption of proportionality results to condensational growth rate being size-independent. The growth rates observed in the atmosphere, however, are not size-independent for the particle size range considered in the model (e.g., Kuang et al., 2012). Furthermore, in all of the simulations used to evaluate the new model, the growth rate is assumed to be size-independent. In other words, the model includes an approximation that is in contradiction to observation, but the error caused by this approximation is not probed at all when evaluating the model accuracy. This is a considerable omission, and additional simulations with size-dependent growth rates should be added. The sectional models used as reference obviously do not suffer from this approximation.

3. As stated on page 16, lines 7-14, the size distribution observed in the chamber measurement is combined from data from three separate instruments, but there is no information to how they are combined. For example, the information from all the three instruments could be used simultaneously (e.g., Viskari et al., 2012) to provide the size distribution, but this seems unlikely based on Figure 4. Another option is that the raw data are inverted separately, which raises the question, how a d$N$/dlog$D_p$ values are calculated for PSM and CPC, which only measure total concentration above a given cut diameter? Without the information of how the observed distribution is obtained, one cannot properly evaluate the accuracy of the new model against those measurements, especially in the sizes below the range of SMPS measurements. A proper explanation of how the measured size distributions are obtained should be given.

4. Beginning from page 13, line 6, until page 14, line 3, it is described how the particles are transferred from PL distribution to LN distribution due to condensational growth. Based on Figure 10, it seems that the value of $D_2$ is affected by the choice of $\gamma$, so this relation should be given explicitly. Furthermore, it is inconvenient for any practical use of the model that $\gamma$ is a free parameter, but very little information is given as to how to choose that value for given conditions to be simulated. In the manuscript, only one example case is provided, which makes it difficult to assess if the optimal value of $\gamma$ for that case can be generalized to other cases or not. For example, does the optimal value of $\gamma$ depend on the particle growth rate or formation rate? In order to facilitate any future use of the model, the authors should provide some advice for choosing the

value of γ, preferably with added examples of comparisons to measurements with different conditions.

5. Page 1, lines 12-16: the first paragraph of the manuscript feels like a few bullet points collected to give some background, and then the rest of the introduction deals with various approaches to modeling particle size distribution. It would serve the reader to have a little longer and more coherent description of the framework and motivation of modelling size distributions.

6. Page 5, line 27: Is the intramodal coagulation really the only process initiating the formation of the LN distribution, or can the condensational transfer initiate it also? According to Eq. (50) the amount of condensational transfer does not depend on the LN distribution, or on the existence of one. Furthermore, on page 17, line 10, it is stated that the coagulational transfer was neglected when simulating conditions of chamber experiment, but LN distribution is still seen in the results. This needs to be clarified.

7. Page 7, lines 5-11: Does this part have something to do with the current work? As far as I understand, the growth rate is used as an input value in the model in all of the simulations, and the condensation process is not really simulated. If this paragraph is important, then it should be made clear why, and if not, it would clarify the article to remove it.

8. Page 7, lines 12-14: It is unclear why the parameters in Eq. (16) need to be considered to vary with $t$ and $D_p$ for the mass growth rate to vary with $t$ and $D_p$. According to Eq. (16), the mass growth rate depends explicitly on $D_p$, and if any term depends on $t$, then the mass growth rate should depend on both $t$ and $D_p$.

9. Page 7, lines 23-25: I do not understand why the new particle formation rate is in this sentence, please clarify.

10. Page 10, lines 18-20: It is stated that the degrees of quadratures are low, but it would serve the reader to provide some examples, how much the simulation results would change due to lower or higher degree of the quadratures used in the model.

11. Page 14, line 16: It is stated that the diameter of newly-formed particles was assumed to be 1.6 nm. Would the results and/or conclusions change with another choice of this diameter? If so, it should be presented, and if not, then that should be mentioned.

12. Page 15, lines 22-23: What is the reason for only considering coagulational transfer, but not condensational transfer?

13. Page 16, lines 13-14: It would serve the reader to explain with a few words how the EEPS and ELPI+ are used to ensure the stability of the aerosol distribution.

14. Page 16, lines 28-30: The PL+LN model was used to estimate $J(t)$ and $g(t)$ via inverse modelling, before those estimates were fine-tuned with FS model. Comparison of the best estimates of $J$ and $g$ from inverse modelling using both the PL+NL and FS models would be an excellent way to demonstrate the capability of the new model. Adding such comparison would increase the

practical use of the manuscript. Performing such comparison manually might not be the most robust approach, though.

15. Page 17, line 15: Only the distributions at the end of the simulations are shown, but it would serve the reader to also provide an example in which the distributions from the PL+NL model and FS1000 model would be compared at other times. This could be done, for example, by showing a surface plot of the relative difference between the two as a function of time and diameter. Such plot would make it possible to evaluate the accuracy of the PL+NL model at all particle sizes and stages of a new particle formation event, instead of just the end distribution.

16. Page 18, lines 31-32: Even if the resolution of the measured distribution is poor below the size-range of the Nano-SMPS measurement, the measured GMD and GSD values would provide a valuable comparison to those from the simulations, especially towards the end of the time domain. I suggest showing also the measured GMD and GSD values in Figures 8 and 9, respectively.

17. Page 19, lines 1-6: This paragraph is a little confusing. I understand that the inverse modelling using the most accurate model produces the best estimates for the $J(t)$ and $g(t)$ representing the actual measurement conditions, and those values are then used as input in the other models, instead of some other values of $J(t)$ and $g(t)$ that would produce better correspondence between those simulation results and measurements. Choosing against what these simulation results are compared to, however, depends on the motivation of the comparison: If the point of interest is, how similar are the distributions simulated with a simple model and a more accurate model, when the same $J(t)$ and $g(t)$ values are used as input in both models, then the distributions from those models should be compared against each other. On the other hand, if the point of interest is, how well does the simple model reproduce the measured size distribution, when the best estimates for $J(t)$ and $g(t)$ representing the measurement conditions are used as input in the model, then the comparison should be against the measurement data, not the accurate model. The authors should revise the paragraph and make it clear what they want to say with it.

Technical corrections:

1. Page 1, line 4: the word "validate" refers to something being labeled as valid, which is not really a proper metric in case of an aerosol dynamics model. I would suggest changing it to "evaluate" here and on other instances the word "validate" is used.

2. Page 1, line 22: $D_p$ and $t$ are used, but they have not been defined yet.

3. Page 2, line 1: there should be a comma on both sides of "e.g.". This issue occurs repeatedly in the manuscript.

4. Page 2, line 4: should read "in which" instead of "of which".

5. Page 2, line 5: consider changing "changed" to "are allowed to vary".

6. Page 2, line 5: One should avoid starting a sentence with "however" when the meaning is "nevertheless". The same issue occurs repeatedly in the manuscript.

7. Page 2, line 10: it is unclear what "too" means in this context.

8. Page 2, line 14: consider changing to "…the effect of numerical diffusion to their results is unknown.".

9. Page 4, line 2: consider using a comma on both sides of the defined variables.

10. Page 4, line 13: there should be a comma on both sides of "i.e.". This issue occurs repeatedly in the manuscript.

11. Page 5, line 15: consider changing "represents" to "presents".

12. Page 6, line 11: consider changing to "where terms on the right hand side denote…".

13. Page 7, line 2: it would serve the reader to clarify that "$i$" is just a shorthand notation for either PL or LN.

14. Page 7, line 5: variables longer than a single letter should not be in italics, "Kn". This error occurs multiple times in the manuscript.

15. Page 7, line 6: add "where" to the beginning of the line.

16. Page 7, line 21: the word "latter" is confusing in the case of more than two items.

17. Page 14, line 15: the FS models are referred to in plural while only a single model was mentioned earlier in the paragraph, please clarify.

18. Page 16, line 1: the "254 nm" is probably the UV wavelength, but it should be made clear.

19. Page 16, lines 30-32: please revise the sentence, it is grammatically incorrect.

20. Page 16, line 33, until page 17, line 1: This sentence is grammatically incorrect, and also somewhat ambiguous about how the deposition coefficient was obtained, which needs to made clear.

21. Page 17, line 16: should read "equal to".

22. Page 17, line 18: should read "the highest".

23. Page 17, lines 24-25: I eventually understood it, but this sentence is quite confusing, consider revising.

24. Page 18, line 15: consider changing "represents" to "presents".

25. Page 18, lines 28-30: I understand the message, but the sentence is poorly worded, consider revising the sentence.

26. Page 19, line 7: consider changing "represents" to "presents".

27. Page 19, lines 8-9: consider changing to "Conversely, using a high value of $\gamma$ produces a more log-normal like form…"

28. Page 19, lines 21-22: "…obtaining input parameters as the model output through inverse modelling" is quite unintuitive way of saying "… using inverse modelling to obtain the best estimates for parameters used as input in the model". Consider revising.

29. Page 19, line 31: The word "conservation" is misleading in this context, as it is commonly used to describe what parameters are conserved in a model, instead of how similar certain parameter values in results from different models are.

30. Page 20, line 8: Saying "simple log-normal distribution model output GMD relatively well" seems to imply that there were rarely any issues with getting the GMD value from the model. Consider revising.

31. Page 23, Table 1: The value of the deposition coefficient is given in nm/s in the text, but in nm/h or nm/s in the table. Using $nms^{-1}$ consistently is advised.

32. Page 24, Table 3: The first sentence "Computational costs and relative errors of different models using the chamber data." is problematic as the table shows errors in certain model outputs, not in the models themselves, and, furthermore, no chamber data per se was used in the simulations, only some input values representing the conditions of the chamber experiment.

33. Page 25, Figure 1: The y-label refers to size distribution as "dN/dlogDp", while "dN/dlnDp" is used in the equations. Consistent notation is suggested to avoid confusion.

34. Page 30, Figure 6: It is often a good practice to provide ticks for the minimum and maximum values of both the x- and y-axis in figures, as it facilitates more accurate comparison to the presented values. In this case, for example, the reader might want to reproduce the model described in this manuscript, and then compare the results from that model to results presented in the manuscript using the same input values.

35. Page 31, Figure 7, caption: It should read "…after the UV lights were switched on…".

References

Kuang, C., Chen, M., Zhao, J., Smith, J., McMurry, P. H., and Wang, J.: Size and time-resolved growth rate measurements of 1 to 5 nm freshly formed atmospheric nuclei, Atmos. Chem. Phys., 12, 3573-3589, doi:10.5194/acp-12-3573-2012, 2012.

Viskari, T., Asmi, E., Virkkula, A., Kolmonen, P., Petäjä, T., and Järvinen, H.: Estimation of aerosol particle number distribution with Kalman Filtering – Part 2: Simultaneous use of DMPS, APS and nephelometer measurements, Atmos. Chem. Phys., 12, 11781-11793, doi:10.5194/acp-12-11781-2012, 2012.

---

## Referee Comment (RC2) · Anonymous Referee #2 · 20 Apr 2016

The authors present a new moment-type method for simulating aerosol microphysics, in which the size distribution is assumed a sum of a power-law part for the smallest particles and a lognormal part for larger ones. The performance is tested against a high resolution sectional model, and additionally compared with low resolution sectional methods as well as a pure lognormal method. The topic is in the scope of ACP but some more work is needed before publication.

In my view there are two major (and necessary!) issues to work on a bit more:

1. The new method is tested only in 0-D. For direct 0-D simulations we do not need

faster methods - even PC:s can handle as many sections as are needed for very good accuracy. However, as the authors state in the conclusions, page 19, lines 19-23: "The PL+LN model is useful in simulations involving the initial steps of aerosol formation where a sectional representation of the size distribution causes too high computational cost, such as in multidimensional simulations or in the case of obtaining input parameters as the model output through inverse modelling." I suggest that the authors show the usefulness of their method in either of these cases: a spatially multidimensional system or an inverse modelling case! The authors have already done some inverse modelling with the method, however it was mentioned to be simple trial and error and not many details of this work were shown. E.g. applying the method with some automatic 'fitting' routine to search for some unknown parameters (such as nucleation rate, condensable vapour concentration or growth rate) would be much more impresssive.

2. The role of the fitting parameter gamma is a little bit troublesome. The authors show that their choice of gamma produces very nice results in the chosen example case. How about other systems? Is a high resolution sectional method always needed to complement the new method? And what about spatially multi-dimensional cases or inverse modelling? Will a single value work in a spatially multidimensional simulation with varying conditions? And, will results of inverse modelling be sensitive to the choice of gamma? Some more advice to the readers/model users would be very valuable! In addition, I would very much like to see figure 4 also for other selections for gamma.

Minor comments:

3. Page 1, line 17: Why is the sectional method called 'shape-preserving'? In my view, e.g. lognormal models are shape-preserving and in contrast, sectional methods allow the shape to evolve according to the dynamics.

4. Page 2, lines 3-4: In addition to moving centre method, similar so-called two-moment methods suffer also from less numerical diffusion than 'regular' fixed sectional methods (e.g. Adams and Seinfeld, JGR 107, D19).

[Figure]

5. Is equation 1 correct? Shouldn't there be delta_D_(j-1) in the denominator of the first term on the right hand side of the lower equation?

6. Page 5, line 30: The lognormal features can arise also because of intramodal coagulation, resulting in a self-preserving distribution that resembles lognormal shape. (Friedlander, Smoke Dust and Haze)

7. Pages 9-10, gaussian quadratures: Such details of numerical integration would be more suitable in an Appendix?

8. Page 16, lines 8-14: Please explain in detail how the size distributions of the different instruments are combined.

9. Page 16, line 15: 'diameters around 15 nm' is very vague.

10. Page 17, line 15: To get a nice result as in figure 6, is the LN distribution needed at all? If not, then it is no wonder that the LN result is much worse than the PL (which is in accordance with theory)?
* * *

---

## Author Comment (AC1) · 27 Apr 2016

The authors' response and the supplement video are included in the supplement package.

Please also note the supplement to this comment:
http://www.atmos-chem-phys-discuss.net/acp-2015-1007/acp-2015-1007-AC1-supplement.zip

---

## Author Response (AR1)

**Response to reviewers' comments of Olin et al.: "Using a combined power law and log-normal distribution model to simulate particle formation and growth in a mobile aerosol chamber"**

We thank the reviewers for their detailed and very useful comments, and have corrected the manuscript according to them.

Referee reports are in *black italic* and authors' responses are in blue roman font. The changes to the manuscript provided as the marked-up manuscript and the changed manuscript are included at the end of this file.

**Referee 1 comments:**

*In their manuscript, "Using a combined power law and log-normal distribution model to simulate particle formation and growth in a mobile aerosol chamber", M. Olin, T. Anttila and M. Dal Maso have described a novel method for simulating the dynamics of aerosol particles, with the emphasis on simulating the early growth of a freshly-formed particle mode in a computationally cost-efficient manner. The authors have tested their model against previous aerosol dynamics models in a few simplified test scenarios to estimate the accuracy of the novel model, and also to demonstrate both the computational efficiency of the novel model in comparison to more accurate models and the accuracy of the novel model in comparison to other models with similar computational burden. Finally, the authors have used the model to reproduce a new particle formation and growth event as observed in aerosol chamber measurements. The main conclusions of the paper are that the newly- developed model is able to provide concentration, surface area and mass concentrations within a few percent to those obtained with a highly accurate model, and that the novel model is able to represent simultaneous new particle formation and particle growth, which is beyond purely log- normal model, which is often used for cost-efficient representation of an aerosol population.*

*Based on the results presented in the manuscript, the new model seems like a useful compromise between accurate and computationally cost-efficient representations of a particle population, and the manuscript is in the scope of ACP journal. A more thorough evaluation of the model is needed, however, and the language of the manuscript should be revised.*

The evaluation of the model is now improved, and the language of the manuscript is checked once more.

*General comments:*

*The main objectives of the manuscript are: 1) to describe the new model in detail, 2) to evaluate the accuracy and computational cost-efficiency of the model, and 3) to demonstrate the applicability of the model using a real life example. I find that the first objective is covered quite well in the manuscript. The second objective, however, is not covered sufficiently: the new model is evaluated against more accurate models in a handful of scenarios, but a more thorough examination is needed. The main issue is that only size-independent growth rates are considered. Another issue is the parameter $\gamma$, which is left as a free parameter in the model, but relatively little consideration is given to how sensitive the model results are to the choice of $\gamma$, or, especially, how to choose the value of $\gamma$ for given simulation conditions. These, and some other issues, are described in more detail in the "Specific comments", below. Regarding the third objective, a single example of reproducing the time evolution of a particle size distribution during a new particle formation and growth event as observed in a chamber experiment is provided. A few different examples would probe the capabilities of the model much more comprehensively. Even more problematic, however, is that in this example, the particle formation and growth rates representing the measurement conditions are obtained with inverse modelling using another, more accurate, model, and those values are used as input in the novel model with the results being compared to the results from the more accurate model. In other words, the only connection to the real life measurement is that the formation and growth rates used as inputs in the models represent the measurement conditions, but otherwise there is not much difference to the scenarios used to evaluate the model. A more suitable demonstration of the usage of the model would be, for example, to estimate the formation and growth rates representing the measurements using inverse modelling with both the accurate model and the novel model, and then compare those values. Finally, the language of the manuscript is not very good at times. The grammatical errors are not exhaustively listed in the "Technical corrections", and a greater care for punctuation, for example, would be needed.*

A more thorough examination to the model is now done. An additional case, where the condensational growth rate is size-dependent, is added to the test cases. Information on how to choose the value of the condensational transfer factor $\gamma$ is added.

The part of inverse modelling is now separated from the part in which the accuracy and the computational cost of the model is examined. Inverse modelling is done using different models, the combined power law and log-normal distribution (PL+LN) model, a simple log-normal (LN) model, and highly accurate models, a fixed-sectional (FS400) and a moving-center fixed sectional (MC100) model. Additionally, the text describing the manually performed trial and error method in finding the best estimates for the functions of the new particle formation ($J$) and condensational growth rates ($g$) is now removed. Instead, a computationally executed least squares method is used to find the best estimates that the most accurately provide the measured number ($N$), surface area ($S$), and mass ($M$) concentrations at every time moment of the centers of the Nano-SMPS scans. In this way, the performance of the PL+LN model in inverse modelling is also examined by comparing the different time series of $J$ and $g$. In other words, all four models have now straight connection to the real-world data. The associated computing time of the automatic inverse modelling procedure using the PL+LN model was approximately 2 orders of magnitude shorter than by using the FS400 or the MC100 model, which implies that a significant improvement in the computing time can be obtained using the PL+LN model in the case of inverse modelling.

The time series of $J$ and $g$ obtained from the FS400 model are then used in the examination of the accuracy and the computational cost of the models, because the comparison of the model outputs is best done by assuming that the FS400 model output is the correct one, not the measured data. In this way, the errors of the model outputs are best seen because the differences of the model outputs are mainly caused by numerical diffusion and the assumptions included in the models.

The language of the manuscript is checked once more.

*Specific comments:*

*1. The title of the manuscript implies that the new model is able to simulate the actual process of particle formation, which is misleading, as the formation rate is used as an input in the model, and also that the emphasis of the manuscript would be on simulations related to the mobile aerosol chamber, but only one example is given. The title should be revised.*

The PL+LN model is now able to simulate the actual process of particle formation, which is shown by performing the inverse modelling using the PL+LN model straightly to the measured data. The manuscript also includes the equations with which the condensational growth rate can be calculated from the vapor concentrations. With those equations, the actual process of particle formation could be simulated with the model if the vapors participating in particle formation and growth were known. The particle formation event measured with the mobile aerosol chamber has now a bigger role in the manuscript, because the inverse modelling part is widened. Therefore, the title is kept unchanged.

*2. On page 7, lines 19-23, it is stated that the mass growth rate is assumed to be proportional to $D_{\mathrm{p}}^2$ in the model, even though only one of the three conditions justifying that assumption is met. Then, on page 8, line 1, it is stated that this assumption of proportionality results to condensational growth rate being size-independent. The growth rates observed in the atmosphere, however, are not size- independent for the particle size range considered in the model (e.g., Kuang et al., 2012). Further-more, in all of the simulations used to evaluate the new model, the growth rate is assumed to be size-independent. In other words, the model includes an approximation that is in contradiction to observation, but the error caused by this approximation is not probed at all when evaluating the model accuracy. This is a considerable omission, and additional simulations with size-dependent growth rates should be added. The sectional models used as reference obviously do not suffer from this approx-imation.*

It is true that the growth rates for the particle size range considered in this manuscript can be size-dependent. However, because the exact function for the condensational growth rate is usually unknown due to unknown vapors participating in the growth process, and because this manuscript concentrates on the description of the PL+LN model, the main focus is on size-independent condensational growth rates.

An additional test case having size-dependent $g$ is, however, included in the test cases now to probe the model more exten-sively. The function chosen for $g(D_{\mathrm{p}})$ is calculated by assuming condensational growth due to sulfuric acid and water vapors in

an atmospheric environment. The resultant function is shown in Fig. 1, from which it can be seen that $g$ increases very steeply at small particle diameters and then remains nearly a constant. Particle diameters in this case lie mainly in the range between 1.6 and 7 nm. In that range, $g$ varies significantly, and therefore, the applicability of the PL+LN model with size-dependent $g$ is probed well with this case. Figure 2 shows the produced size distributions. Due to steeply increasing $g$ at the smallest particle sizes, the size distribution is of a different form, which can be seen in the size distribution produced by the FS1000 model. This kind of size distribution does not follow a power law form, and therefore the PL+LN model is not able to express the distribution correctly at very small particle sizes. The relative errors of the concentrations produced by the PL+LN model compared to the FS1000 model are up to 17 % in this case. Nevertheless, the LN model behaves even more unsatisfactorily; the relative errors rise up to 90 %.

[Figure]

**Figure 1.** Size-dependent condensational growth rate of sulfuric acid-water particles with the sulfuric acid vapor concentration of $0.8 \times 10^7 \, \mathrm{cm}^{-3}$, temperature of 280 K, and relative humidity of 60 % as a function of the particle diameter.

[Figure]

**Figure 2.** Particle size distribution modelled with size-dependent condensational growth rate using a very accurate fixed-sectional model (FS1000), the simple log-normal model (LN), and the combined power law and log-normal distribution model (PL+LN).

*3. As stated on page 16, lines 7-14, the size distribution observed in the chamber measurement is combined from data from three separate instruments, but there is no information to how they are combined. For example, the information from all the three instruments could be used simultaneously (e.g., Viskari et al., 2012) to provide the size distribution, but this seems unlikely based on Figure 4. Another option is that the raw data are inverted separately, which raises the question, how a $\mathrm{d}N/\mathrm{d}\log D_\mathrm{p}$ values are calculated for PSM and CPC, which only measure total concentration above a given cut diameter? Without the information of how the observed distribution is obtained, one cannot properly evaluate the accuracy of the new model against those measurements, especially in the sizes below the range of SMPS measurements. A proper explanation of how the measured size distributions are obtained should be given.*

An explanation of how the data from different devices are combined is now added. The approximated cut diameters of the devices, $D_\mathrm{PSM} = 1.6\,\mathrm{nm}$, $D_\mathrm{CPC} = 3.6\,\mathrm{nm}$, and $D_\mathrm{Nano\text{-}SMPS} = 7\,\mathrm{nm}$, are used to combine the data of PSM, CPC, and Nano-SMPS to obtain total aerosol size distributions for the diameter range of $1.6 - 64\,\mathrm{nm}$ with

$$\left.\frac{\mathrm{d}N}{\mathrm{d}\ln D_\mathrm{p}}\right|_\mathrm{measured} = \begin{cases} \frac{\max\{N_\mathrm{PSM}-N_\mathrm{CPC},0\}}{\ln\left(D_\mathrm{CPC}/D_\mathrm{PSM}\right)}, & D_\mathrm{PSM} \leq D_\mathrm{p} < D_\mathrm{CPC} \\[2mm] \frac{\max\{N_\mathrm{CPC}-N_\mathrm{Nano\text{-}SMPS},0\}}{\ln\left(D_\mathrm{Nano\text{-}SMPS}/D_\mathrm{CPC}\right)}, & D_\mathrm{CPC} \leq D_\mathrm{p} < D_\mathrm{Nano\text{-}SMPS} \,, \\[2mm] \left.\frac{\mathrm{d}N}{\mathrm{d}\ln D_\mathrm{p}}\right|_\mathrm{Nano\text{-}SMPS}, & D_\mathrm{p} \geq D_\mathrm{Nano\text{-}SMPS} \end{cases} \tag{1}$$

where $N_{\text{PSM}}$, $N_{\text{CPC}}$, and $N_{\text{Nano-SMPS}}$ are the total number concentrations measured by the devices, and $\frac{\mathrm{d}N}{\mathrm{d}\ln D_{\text{p}}}\Big|_{\text{Nano-SMPS}}$ is the particle size distribution measured by Nano-SMPS.

*4. Beginning from page 13, line 6, until page 14, line 3, it is described how the particles are transferred from PL distribution to LN distribution due to condensational growth. Based on Figure 10, it seems that the value of $D_2$ is affected by the choice of $\gamma$, so this relation should be given explicitly. Furthermore, it is inconvenient for any practical use of the model that $\gamma$ is a free parameter, but very little information is given as to how to choose that value for given conditions to be simulated. In the manuscript, only one example case is provided, which makes it difficult to assess if the optimal value of $\gamma$ for that case can be generalized to other cases or not. For example, does the optimal value of $\gamma$ depend on the particle growth rate or formation rate? In order to facilitate any future use of the model, the authors should provide some advice for choosing the value of $\gamma$, preferably with added examples of comparisons to measurements with different conditions.*

The value of $D_2$ is affected by the choice of $\gamma$, but not with an explicit relation, because the value of $D_2$ is not a direct consequence of the value of $\gamma$. The value of $\gamma$ affects on the amount of particles transferred from the PL distribution to the LN distribution, which affects on the value of $D_2$ through the transferred amount of the largest particles in the PL distribution.

Information on how to choose the value of the condensational transfer factor $\gamma$ is now added. The value $\gamma = 0$ produces a distribution that will be mainly in a power law form; the value $\gamma = 1$ produces a log-normal distribution only. To choose a suitable value for $\gamma$ for a simulation, the user should consider how well does the aerosol formation event follow the approximations of the theory described in the manuscript. The value 0 is suitable only when the aerosol processes follow the theory exactly. To simulate a realistic particle formation event, the value has to be increased towards unity using the following guidelines. The more the following conditions are met, the higher $\gamma$ should be used: (1) particle formation or growth are multicomponent processes, (2) the particle formation rate or the condensation growth rate vary significantly with time, (3) the condensational growth rate varies significantly with the particle size, (4) the background aerosol acting as a coagulation sink does not remain in a nearly constant state during the time domain of the simulation, (5) particle sizes in the background aerosol are not significantly higher than in the PL+LN distribution, (6) the depositional losses cannot be approximated with as simple form as described in the manuscript, e.g., in the case of complex geometry or turbulent flow. In real atmospheric particle formation events, $\gamma$ should rarely has the value of less than 0.5, which can also be used as an initial guess if figuring the previous guidelines is problematic. If the shapes of the distributions to be modelled are initially known, the value of $\gamma$ can be adjusted to obtain a proper model output, e.g., in the case of inverse modelling. The factor $\gamma$ can also be considered a time-dependent function, but in the manuscript we concentrate only to constant values of $\gamma$.

*5. Page 1, lines 12-16: the first paragraph of the manuscript feels like a few bullet points collected to give some background, and then the rest of the introduction deals with various approaches to modeling particle size distribution. It would serve the reader to have a little longer and more coherent description of the framework and motivation of modelling size distributions.*

Introduction is now extended to contain description about the modelling of size distributions.

*6. Page 5, line 27: Is the intramodal coagulation really the only process initiating the formation of the LN distribution, or can the condensational transfer initiate it also? According to Eq. (50) the amount of condensational transfer does not depend on the LN distribution, or on the existence of one. Furthermore, on page 17, line 10, it is stated that the coagulational transfer was neglected when simulating conditions of chamber experiment, but LN distribution is still seen in the results. This needs to be clarified.*

The original sentence "The intramodal coagulation of the PL distribution remains the only process initiating the formation the LN distribution" was lacking an important part "if the condensational transfer is neglected" which is now added. It is true that both the coagulational transfer and the condensational transfer can alone initiate the LN distribution and these processes do not require the existence of the LN distribution.

*7. Page 7, lines 5-11: Does this part have something to do with the current work? As far as I understand, the growth rate is used as an input value in the model in all of the simulations, and the condensation process is not really simulated. If this paragraph is important, then it should be made clear why, and if not, it would clarify the article to remove it.*

The part, in which the condensation term is expressed using the properties and concentrations of the vapors participating in the growth process, is now relevant in the manuscript because the condensation process is now simulated in the case with size-dependent growth rate. Therefore, it is not removed from the manuscript.

*8. Page 7, lines 12-14: It is unclear why the parameters in Eq. (16) need to be considered to vary with $t$ and $D_\mathrm{p}$ for the mass growth rate to vary with $t$ and $D_\mathrm{p}$. According to Eq. (16), the mass growth rate depends explicitly on $D_\mathrm{p}$, and if any term depends on $t$, then the mass growth rate should depend on both $t$ and $D_\mathrm{p}$.*

The meaning of the sentence should have been that the parameters in Eq. (16) need to be considered to vary with $t$ and $D_\mathrm{p}$ only, not with the spatial location. Thus, if the parameters do not vary with the spatial location, the mass growth rate will vary neither. The sentence is now clarified.

*9. Page 7, lines 23-25: I do not understand why the new particle formation rate is in this sentence, please clarify.*

There was nothing to do with the new particle formation rate in that sentence and it is now removed.

*10. Page 10, lines 18-20: It is stated that the degrees of quadratures are low, but it would serve the reader to provide some examples, how much the simulation results would change due to lower or higher degree of the quadratures used in the model.*

The effect of the degrees of the quadratures are now examined more thoroughly. Quadratures are now used also with the condensation term because it cannot be calculated analytically with a size-dependent growth rate. The examination of the quadrature developed here revealed that it should be used only when $\alpha > 0.5$, in the case of condensation, and when $D_2/D_1 < 3$, in the case of coagulation. In those regions, the absolute relative errors of the terms are less than $10^{-2}$ when calculated with the quadrature. In the remaining regions, numerical integration is used. The equation for the numerical integration is now added. The computing time associated in calculating the condensation term is 1 or 2 orders shorter with the quadrature compared to numerical integration. The degree of 5 used with the Hermite-Gauss quadrature provides the absolute relative errors of the terms less than $10^{-4}$.

*11. Page 14, line 16: It is stated that the diameter of newly-formed particles was assumed to be 1.6 nm. Would the results and/or conclusions change with another choice of this diameter? If so, it should be presented, and if not, then that should be mentioned.*

The choice of $D_1$ does not affect on the results of the test cases and not on the examination of the accuracy and the computational cost of the model using the chamber event simulation, because sectional models have the diameter of newly-formed particles too. However, it has an effect in inverse modelling because it determines the sizes of the smallest particles. The smaller the particles are the higher their losses are; therefore, lower values of $D_1$ result in small particles with high losses, and higher new particle formation rate is needed to obtain the measured concentrations. Approximately $40\%$ higher values for $J$ is needed if $D_1 = 1\,\mathrm{nm}$ is used compared to $D_1 = 3\,\mathrm{nm}$, in the chamber event simulation.

*12. Page 15, lines 22-23: What is the reason for only considering coagulational transfer, but not condensational transfer?*

Because the test cases are purely theoretical, the need of constructing log-normal features to the distributions through the condensational transfer artificially is minimal. This sentence is now added to the manuscript.

*13. Page 16, lines 13-14: It would serve the reader to explain with a few words how the EEPS and ELPI+ are used to ensure the stability of the aerosol distribution.*

EEPS and ELPI+ having time resolutions of only $1\,\mathrm{s}$ were used to ensure the stability of the aerosol distribution during a Nano-SMPS scan lasting $150\,\mathrm{s}$: no rapid changes in the aerosol distribution were observed in the time scales shorter than $150\,\mathrm{s}$. This sentence is now added to the manuscript.

*14. Page 16, lines 28-30: The PL+LN model was used to estimate $J(t)$ and $g(t)$ via inverse modelling, before those estimates were fine-tuned with FS model. Comparison of the best estimates of J and g from inverse modelling using both the PL+NL and FS models would be an excellent way to demonstrate the capability of the new model. Adding such comparison would increase the practical use of the manuscript. Performing such comparison manually might not be the most robust approach, though.*

Inverse modelling is now done using the PL+LN, the LN, the FS400, and the MC100 models. Figures 3 and 4 show the time series of $J(t)$ and $g(t)$ obtained using different models. It can be seen that there are no significant differences between the time series obtained using the PL+LN model compared to the highly accurate models. Therefore, the PL+LN model is capable in inverse modelling.

[Figure]

**Figure 3.** Time series for the new particle formation rates in the chamber event that produce the measured concentrations, $N$, $S$, and $M$, the most accurately compared to the measured ones, using different models. The fit denotes a bell-shaped function fitted to the values from the FS400 model.

[Figure]

**Figure 4.** Time series for the condensational growth rates in the chamber event that produce the measured concentrations, $N$, $S$, and $M$, the most accurately compared to the measured ones, using different models. The fit denotes a linear function fitted to the values from the FS400 model.

*15. Page 17, line 15: Only the distributions at the end of the simulations are shown, but it would serve the reader to also provide an example in which the distributions from the PL+NL model and FS1000 model would be compared at other times. This could be done, for example, by showing a surface plot of the relative difference between the two as a function of time and diameter. Such plot would make it possible to evaluate the accuracy of the PL+NL model at all particle sizes and stages of a new particle formation event, instead of just the end distribution.*

The relative errors of the moments are now presented as time series (Fig. 5). A surface plot of the relative errors does not work well with particle size distributions because they have values in several different orders of magnitude. Therefore, particle size distributions of the test cases are now presented at every time moment using a video supplement.

[Figure]

**Figure 5.** The relative errors of the moments ($\delta_X$) in the test cases produced by the PL+LN model.

*16. Page 18, lines 31-32: Even if the resolution of the measured distribution is poor below the size- range of the Nano-SMPS measurement, the measured GMD and GSD values would provide a valuable comparison to those from the simulations, especially towards the end of the time domain. I suggest showing also the measured GMD and GSD values in Figures 8 and 9, respectively.*

The measured GMD and GSD are now shown in a figure.

*17. Page 19, lines 1-6: This paragraph is a little confusing. I understand that the inverse modelling using the most accurate model produces the best estimates for the $J(t)$ and $g(t)$ representing the actual measurement conditions, and those values are then used as input in the other models, instead of some other values of $J(t)$ and $g(t)$ that would produce better correspondence between those simulation results and measurements. Choosing against what these simulation results are compared to, however, depends on the motivation of the comparison: If the point of interest is, how similar are the distributions simulated with a simple model and a more accurate model, when the same $J(t)$ and $g(t)$ values are used as input in both models, then the distributions from those models should be compared against each other. On the other hand, if the point of interest is, how well does the simple model reproduce the measured size distribution, when the best estimates for $J(t)$ and $g(t)$ representing the measurement conditions are used as input in the model, then the comparison should be against the measurement data, not*

*the accurate model. The authors should revise the paragraph and make it clear what they want to say with it.*

The whole paragraph is now removed and the same message is said in different form in different part of the text because there is now an own section for inverse modelling.

*Technical corrections:*

*1. Page 1, line 4: the word "validate" refers to something being labeled as valid, which is not really a proper metric in case of an aerosol dynamics model. I would suggest changing it to "evaluate" here and on other instances the word "validate" is used.*

All validations are now replaced with evaluations.

*2. Page 1, line 22: $D_\mathrm{p}$ and $t$ are used, but they have not been defined yet.*

$D_\mathrm{p}$ and $t$ are now moved to a later point, where they are also defined.

*3. Page 2, line 1: there should be a comma on both sides of "e.g.". This issue occurs repeatedly in the manuscript.*

Commas to both sides of all occurrences of "e.g." are now added.

*4. Page 2, line 4: should read "in which" instead of "of which".*

It is corrected now.

*5. Page 2, line 5: consider changing "changed" to "are allowed to vary".*

It is changed now.

*6. Page 2, line 5: One should avoid starting a sentence with "however" when the meaning is "nevertheless". The same issue occurs repeatedly in the manuscript.*

All occurrences of "however" are now checked and replaced with "nevertheless" when relevant.

*7. Page 2, line 10: it is unclear what "too" means in this context.*

"due to the requirement of storing the center values of the sections too" is now replaced with "due to the requirement of storing also the center values of the sections".

*8. Page 2, line 14: consider changing to "...the effect of numerical diffusion to their results is unknown.".*

It is corrected now.

*9. Page 4, line 2: consider using a comma on both sides of the defined variables.*

It is corrected now.

*10. Page 4, line 13: there should be a comma on both sides of "i.e.". This issue occurs repeatedly in the manuscript.*

Commas to both sides of all occurrences of "i.e." are now added.

*11. Page 5, line 15: consider changing "represents" to "presents".*

It is corrected now.

*12. Page 6, line 11: consider changing to "where terms on the right hand side denote...".*

It is corrected now.

*13. Page 7, line 2: it would serve the reader to clarify that "i" is just a shorthand notation for either PL or LN.*

It is clarified now.

*14. Page 7, line 5: variables longer than a single letter should not be in italics, "Kn". This error occurs multiple times in the manuscript.*

All occurrences of "Kn" are now changed to roman font.

*15. Page 7, line 6: add "where" to the beginning of the line.*

It is added now.

*16. Page 7, line 21: the word "latter" is confusing in the case of more than two items.*

The word "latter" is now replaced with "last".

*17. Page 14, line 15: the FS models are referred to in plural while only a single model was mentioned earlier in the paragraph, please clarify.*

It is clarified now.

*18. Page 16, line 1: the "254 nm" is probably the UV wavelength, but it should be made clear.*

It is clarified now.

*19. Page 16, lines 30-32: please revise the sentence, it is grammatically incorrect.*

Trial and error method is now cancelled; therefore, the sentence does not exist in the manuscript any more.

*20. Page 16, line 33, until page 17, line 1: This sentence is grammatically incorrect, and also somewhat ambiguous about how the deposition coefficient was obtained, which needs to made clear.*

It is clarified now.

*21. Page 17, line 16: should read "equal to".*

It is corrected now.

*22. Page 17, line 18: should read "the highest".*

It is corrected now.

*23. Page 17, lines 24-25: I eventually understood it, but this sentence is quite confusing, consider revising.*

It is clarified now.

*24. Page 18, line 15: consider changing "represents" to "presents".*

It is corrected now.

*25. Page 18, lines 28-30: I understand the message, but the sentence is poorly worded, consider revising the sentence.*

The sentence is now revised.

*26. Page 19, line 7: consider changing "represents" to "presents".*

It is corrected now.

*27. Page 19, lines 8-9: consider changing to "Conversely, using a high value of $\gamma$ produces a more log-normal like form..."*

It is revised now.

*28. Page 19, lines 21-22: "...obtaining input parameters as the model output through inverse modelling" is quite unintuitive way of saying "... using inverse modelling to obtain the best estimates for parameters used as input in the model". Consider revising.*

It is revised now.

*29. Page 19, line 31: The word "conservation" is misleading in this context, as it is commonly used to describe what parameters are conserved in a model, instead of how similar certain parameter values in results from different models are.*

The word "conservation" is now replaced with "accuracy".

*30. Page 20, line 8: Saying "simple log-normal distribution model output GMD relatively well" seems to imply that there were rarely any issues with getting the GMD value from the model. Consider revising.*

The GMD values were obtained well using the LN model in this case, and "in this case" is now added to the end of the sentence.

*31. Page 23, Table 1: The value of the deposition coefficient is given in nm/s in the text, but in nm/h or nm/s in the table. Using nms$^{-1}$ consistently is advised.*

The units of the deposition coefficient are now changed consistently to $\mathrm{nm\,h^{-1}}$. However, in the case of the test case with the parameters reflecting vehicle exhaust, the units are $\mathrm{nm\,s^{-1}}$ due to the time scale of seconds instead of hours.

*32. Page 24, Table 3: The first sentence "Computational costs and relative errors of different models using the chamber data." is problematic as the table shows errors in certain model outputs, not in the models themselves, and, furthermore, no chamber data per se was used in the simulations, only some input values representing the conditions of the chamber experiment.*

It is now clarified that the relative errors in the table represent the values at the end of the simulation, and "chamber data" is replaced with "chamber simulation".

*33. Page 25, Figure 1: The y-label refers to size distribution as "$\mathrm{d}N/\mathrm{d}\log D_{\mathrm{p}}$", while "$\mathrm{d}N/\mathrm{d}\ln D_{\mathrm{p}}$" is used in the equations. Consistent notation is suggested to avoid confusion.*

All size distributions in the text and in the figures are now expressed with "$\mathrm{d}N/\mathrm{d}\ln D_{\mathrm{p}}$".

*34. Page 30, Figure 6: It is often a good practice to provide ticks for the minimum and maximum values of both the x- and y-axis in figures, as it facilitates more accurate comparison to the presented values. In this case, for example, the reader might want to reproduce the model described in this manuscript, and then compare the results from that model to results presented in the manuscript using the same input values.*

The minimum and maximum values of the both axes are now shown in figures, with the exception of some time axes, because they would disrupt with the time ticks of $0\,\mathrm{s}$ that should be visible because it is the time of switching the UV lights on.

*35. Page 31, Figure 7, caption: It should read "...after the UV lights were switched on...".*

It is corrected now

The LN distribution is not needed in those cases to achieve as accurate results for the variables (the number, the surface area, and the mass concentrations, GMD, and GSD) as with both the PL and the LN distribution, but by visual inspection, the LN distribution is needed to obtain distributions that have the correct shapes in the highest particle sizes. This information is now added to the manuscript. It is in accordance with theory that the distribution will be in a purely power law form if coagulation is totally neglected, $J$ and $g$ are constants, and particle losses are modelled with the equations used in the manuscript. In that case, the LN distribution is not needed at all. However, by including coagulation and modelling other processes with more complex equations, the need for the LN distribution rises.

[revised manuscript text omitted]

$$\frac{\mathrm{d}m_{\text{p}}}{\mathrm{d}t}(t, D_{\text{p}}) = \frac{\mathrm{d}m_{\text{p}}}{\mathrm{d}D_{\text{p}}} \cdot \frac{\mathrm{d}D_{\text{p}}}{\mathrm{d}t}(t, D_{\text{p}}) = \frac{\pi}{2}\rho D_{\text{p}}^2 g(t, D_{\text{p}}). \tag{18}$$

Hence, the condensation rate for a particle distribution becomes

$$\text{cond}_{M_i} = \frac{\pi}{2}\rho \int_{-\infty}^{\infty} D_{\text{p}}^2 g(t, D_{\text{p}}) \left.\frac{\mathrm{d}N}{\mathrm{d}\ln D_{\text{p}}}\right|_i \mathrm{d}\ln D_{\text{p}}, \tag{19}$$

10   which has an analytical solution for  the both distributions, PL and LN, when $g(t, D_{\text{p}})$ can be expressed with a polynomial of $D_{\text{p}}$. The mass growth rate is proportional to $D_{\text{p}}^2$ if the following conditions are met: 1) the particle size is in free-molecular regime, 2) $D_p \gg D_v$, 3) $C_{v,\infty} \gg C_{v,\text{p}}$. The  last one applies when the particle size is large or when the vapor has low saturation vapor pressure. Since particle sizes near the molecular size are modelled in this article, only the first condition applies satisfactorily.  Nevertheless, this article concentrates mainly in cases

15   where the mass growth rate is assumed to be proportional to $D_{\text{p}}^2$.  Additionally, a single test case, where the mass growth rate is calculated using Eqs. (16) – (17), is presented. The main point in this article is not to provide the correct formulation for $g(t, D_{\text{p}})$, but to compare different models, and additionally to perform inverse modelling to obtain $g(t)$ from the time evolution of measured aerosol size distributions. Due to the assumption of the proportionality of the mass growth rate, the condensational growth rate becomes size-independent, and finally, the condensation terms used in

20   Eq. (13) become

$$\text{cond}_X = \begin{cases} 0, & X = N_i \\ 2\pi\, g(t) \int_{-\infty}^{\infty} D_{\text{p}}\, \mathrm{d}N_i, & X = S_i \\ \frac{\pi}{2}\rho\, g(t) \int_{-\infty}^{\infty} D_{\text{p}}^2\, \mathrm{d}N_i, & X = M_i \end{cases} \tag{20}$$

where $\mathrm{d}N_i$ is an abbreviation of

$$\left.\frac{\mathrm{d}N}{\mathrm{d}\ln D_{\text{p}}}\right|_i \mathrm{d}\ln D_{\text{p}}. \tag{21}$$

The analytical solution for Eq. (20) is

$$
\mathrm{cond}_X = X\,g(t) \cdot
\begin{cases}
0, & X = N_{\mathrm{PL}} \\
\frac{2}{D_1}\left(\frac{\alpha+2}{\alpha+1}\right)\left(\frac{d^{\alpha+1}-1}{d^{\alpha+2}-1}\right), & X = S_{\mathrm{PL}} \\
\frac{3}{D_1}\left(\frac{\alpha+3}{\alpha+2}\right)\left(\frac{d^{\alpha+2}-1}{d^{\alpha+3}-1}\right), & X = M_{\mathrm{PL}} \\
0, & X = N_{\mathrm{LN}} \\
\frac{2}{D_{\mathrm{g}}}\exp\left(-\frac{3}{2}\ln^2\sigma\right), & X = S_{\mathrm{LN}} \\
\frac{3}{D_{\mathrm{g}}}\exp\left(-\frac{5}{2}\ln^2\sigma\right), & X = M_{\mathrm{LN}}
\end{cases}
\tag{22}
$$

when $\alpha$ is not -3, -2, or -1.

When the mass growth rate is calculated from the vapor concentrations and the properties of the vapor and the particles using Eqs. (16) – (17), it rarely can be expressed with a polynomial of $D_{\mathrm{p}}$, unless polynomial fits are done for the function. However, if the vapor concentrations or the other properties are allowed to vary during the simulation, fits for the mass growth rate function may become inconvenient. In that case, the integral in Eq. (19) cannot be solved analytically. Therefore, numerical integration is required, in which Eq. (19) is calculated in a form of

$$
\mathrm{cond}_{M_i} = \frac{\pi}{2}\rho \sum_{j=1}^{n} D_j^2\, g(t, D_j)\,\left.\frac{\mathrm{d}N}{\mathrm{d}\ln D_{\mathrm{p}}}\right|_i \ln\frac{D_{j+1}}{D_j},
\tag{23}
$$

where $D_j$ is the particle diameter of the size section $j$ used in numerical integration when the particle diameter range is split into $n$ sections. Computational cost of numerical integration is, however, higher compared to analytical solution of the integrals. Therefore, Gaussian quadratures are used here to reduce the associated computing time; they provide the optimal particle diameters and their weights for efficient evaluation of the integrals. The details of the Gaussian quadratures are described in Appendix A.

**2.2.3 Coagulation**

Coagulation is modelled as intramodal coagulation within the PL distribution and within the LN distribution, and as intermodal coagulation from the PL distribution to the LN distribution. The coagulation terms derived from the equations of Whitby and

McMurry (1997) are

$$\mathrm{coag}_{N_{\mathrm{PL}}} = -\frac{1}{2}\int_{-\infty}^{\infty}\int_{-\infty}^{\infty}\beta(D_{\mathrm{p}},D_{\mathrm{p}}')\,\mathrm{d}N_{\mathrm{PL}}\,\mathrm{d}N_{\mathrm{PL}}'$$

$$-\int_{-\infty}^{\infty}\int_{-\infty}^{\infty}\beta(D_{\mathrm{p}},D_{\mathrm{p}}')\,\mathrm{d}N_{\mathrm{PL}}\,\mathrm{d}N_{\mathrm{LN}}' \tag{24}$$

$$\mathrm{coag}_{S_{\mathrm{PL}}} = -\frac{1}{2}\int_{-\infty}^{\infty}\int_{-\infty}^{\infty}\left[2s_{\mathrm{p}}-\left(s_{\mathrm{p}}^{\frac{3}{2}}+s_{\mathrm{p}}'^{\frac{3}{2}}\right)^{\frac{2}{3}}\right]\beta(D_{\mathrm{p}},D_{\mathrm{p}}')\,\mathrm{d}N_{\mathrm{PL}}\,\mathrm{d}N_{\mathrm{PL}}'$$

$$-\int_{-\infty}^{\infty}\int_{-\infty}^{\infty}s_{\mathrm{p}}\,\beta(D_{\mathrm{p}},D_{\mathrm{p}}')\,\mathrm{d}N_{\mathrm{PL}}\,\mathrm{d}N_{\mathrm{LN}}' \tag{25}$$

$$\mathrm{coag}_{M_{\mathrm{PL}}} = -\int_{-\infty}^{\infty}\int_{-\infty}^{\infty}m_{\mathrm{p}}\,\beta(D_{\mathrm{p}},D_{\mathrm{p}}')\,\mathrm{d}N_{\mathrm{PL}}\,\mathrm{d}N_{\mathrm{LN}}' \tag{26}$$

$$\mathrm{coag}_{N_{\mathrm{LN}}} = -\frac{1}{2}\int_{-\infty}^{\infty}\int_{-\infty}^{\infty}\beta(D_{\mathrm{p}},D_{\mathrm{p}}')\,\mathrm{d}N_{\mathrm{LN}}\,\mathrm{d}N_{\mathrm{LN}}' \tag{27}$$

$$\mathrm{coag}_{S_{\mathrm{LN}}} = -\frac{1}{2}\int_{-\infty}^{\infty}\int_{-\infty}^{\infty}\left[2s_{\mathrm{p}}-\left(s_{\mathrm{p}}^{\frac{3}{2}}+s_{\mathrm{p}}'^{\frac{3}{2}}\right)^{\frac{2}{3}}\right]\beta(D_{\mathrm{p}},D_{\mathrm{p}}')\,\mathrm{d}N_{\mathrm{LN}}\,\mathrm{d}N_{\mathrm{LN}}'$$

$$+\int_{-\infty}^{\infty}\int_{-\infty}^{\infty}\left[\left(s_{\mathrm{p}}^{\frac{3}{2}}+s_{\mathrm{p}}'^{\frac{3}{2}}\right)^{\frac{2}{3}}-s_{\mathrm{p}}'\right]\beta(D_{\mathrm{p}},D_{\mathrm{p}}')\,\mathrm{d}N_{\mathrm{PL}}\,\mathrm{d}N_{\mathrm{LN}}' \tag{28}$$

$$\mathrm{coag}_{M_{\mathrm{LN}}} = \int_{-\infty}^{\infty}\int_{-\infty}^{\infty}m_{\mathrm{p}}\,\beta(D_{\mathrm{p}},D_{\mathrm{p}}')\,\mathrm{d}N_{\mathrm{PL}}\,\mathrm{d}N_{\mathrm{LN}}', \tag{29}$$

where $\beta(D_{\mathrm{p}},D_{\mathrm{p}}')$ is the coagulation coefficient of particles with  the diameters of $D_{\mathrm{p}}$ and $D_{\mathrm{p}}'$ calculated with the equation

$$\beta(D_{\mathrm{p}},D_{\mathrm{p}}') = 2\pi(D_{\mathrm{p}}+D_{\mathrm{p}}')(\mathcal{D}_{\mathrm{p}}+\mathcal{D}_{\mathrm{p}}')f(\underline{Kn}\mathrm{Kn}_{\mathrm{coag}}), \tag{30}$$

where $\cancel{f(Kn_{\mathrm{coag}})}\,f(\mathrm{Kn}_{\mathrm{coag}})$ is the transition regime function of Dahneke (1983)

$$f(\underline{Kn}\mathrm{Kn}_{\mathrm{coag}}) = \frac{1+Kn_{\mathrm{coag}}}{1+2Kn_{\mathrm{coag}}+2Kn_{\mathrm{coag}}^2}\,\frac{1+\mathrm{Kn}_{\mathrm{coag}}}{1+2\mathrm{Kn}_{\mathrm{coag}}+2\mathrm{Kn}_{\mathrm{coag}}^2}, \tag{31}$$

where $\cancel{Kn_{\mathrm{coag}}}\,\mathrm{Kn}_{\mathrm{coag}}$ is the Knudsen number for coagulation

$$\underline{Kn}\mathrm{Kn}_{\mathrm{coag}} = \frac{4(\mathcal{D}_{\mathrm{p}}+\mathcal{D}_{\mathrm{p}}')}{(D_{\mathrm{p}}+D_{\mathrm{
[revised manuscript text omitted]

$$\mathrm{transfer}_{N_{\mathrm{PL}}} = -\,\mathrm{transfer}_{N_{\mathrm{LN}}} = -\frac{1}{2} \int_{-\infty}^{\infty} \int_{\ln D_{\mathrm{coag}}}^{\infty} \beta(D_{\mathrm{p}}, D_{\mathrm{p}}')\,\mathrm{d}N_{\mathrm{PL}}\,\mathrm{d}N_{\mathrm{PL}}' \tag{41}$$

$$\mathrm{transfer}_{S_{\mathrm{PL}}} = -\,\mathrm{transfer}_{S_{\mathrm{LN}}} = -\frac{1}{2} \int_{-\infty}^{\infty} \int_{\ln D_{\mathrm{coag}}}^{\infty} \left(s_{\mathrm{p}}^{\frac{3}{2}} + s_{\mathrm{p}}'^{\frac{3}{2}}\right)^{\frac{2}{3}} \beta(D_{\mathrm{p}}, D_{\mathrm{p}}')\,\mathrm{d}N_{\mathrm{PL}}\,\mathrm{d}N_{\mathrm{PL}}' \tag{42}$$

$$\mathrm{transfer}_{M_{\mathrm{PL}}} = -\,\mathrm{transfer}_{M_{\mathrm{LN}}} = -\frac{1}{2} \int_{-\infty}^{\infty} \int_{\ln D_{\mathrm{coag}}}^{\infty} \left(m_{\mathrm{p}} + m_{\mathrm{p}}'\right) \beta(D_{\mathrm{p}}, D_{\mathrm{p}}')\,\mathrm{d}N_{\mathrm{PL}}\,\mathrm{d}N_{\mathrm{
[revised manuscript text omitted]
{dm_{p,H_2O}}{dt} = \left[\frac{1}{Y_{H_2SO_4}(D_p)} - 1\right] \cdot \frac{dm_{p,H_2SO_4}}{dt}, \qquad (51)$$

where $Y_{H_2SO_4}(D_p)$ is the mass fraction of sulfuric acid in a particle in water equilibrium, i.e., a particle having the composition with which no condensation or evaporation of water vapor occurs in temperature of $280\,K$ and relative humidity of $60\,\%$
5  when the particle diameter is $D_p$. The approximation of water equilibrium is reasonable because, with these environmental values, $\sim 2 \times 10^{10}$ times more water molecules than sulfuric acid molecules exist and thus there are probably a sufficient amount of water molecules to condense on the particle to reach the equilibrium state before the next sulfuric acid molecule condenses on it. The properties of the vapors and the particles were calculated, using the equilibrium composition, as described in Olin et al. (2015). These environmental values were chosen because they are relevant values met in the atmosphere and they
10  cause the condensational growth rate function that is far beyond a constant value in the particle diameter range of this case (from $1.6\,nm$ to $8\,nm$), as seen in Fig. 4, which provides a beneficial test to examine how the model behaves with size-dependent $g$.

The Exh case represents simultaneous new particle formation, condensation, intramodal- and intermodal coagulation, coagulational losses, and depositional losses occurring in diesel vehicle exhaust inside the ageing chamber of a laboratory sampling system. The values $N_{bg} = 10^6\,cm^{-3}$ and $CMD_{bg} = 60\,nm$ were obtained from the measurements of Rönkkö et al.
15  (2013) and the corresponding $l_{bg} = -1.5$ from Lehtinen et al. (2007) using $CMD_{bg} = 60\,nm$. The deposition coefficient $k_{dep} = 7 \times 10^{-2}\,nm/s$ $k_{dep} = 0.07\,
[revised manuscript text omitted]
_{\mathrm{p}}, D_{\mathrm{p}}') \, \mathrm{d}N_{\mathrm{PL}} \, \mathrm{d}N_{\mathrm{LN}}' \tag{24}
$$

$$
\mathrm{coag}_{S_{\mathrm{PL}}} = -\frac{1}{2} \int_{-\infty}^{\infty} \int_{-\infty}^{\infty} \left[ 2s_{\mathrm{p}} - \left( s_{\mathrm{p}}^{\frac{3}{2}} + s_{\mathrm{p}}'^{\frac{3}{2}} \right)^{\frac{2}{3}} \right] \beta(D_{\mathrm{p}}, D_{\mathrm{p}}') \, \mathrm{d}N_{\mathrm{PL}} \, \mathrm{d}N_{\mathrm{PL}}'
$$

$$
- \int_{-\infty}^{\infty} \int_{-\infty}^{\infty} s_{\mathrm{p}} \, \beta(D_{\mathrm{p}}, D_{\mathrm{p}}') \, \mathrm{d}N_{\mathrm{PL}} \, \mathrm{d}N_{\mathrm{LN}}' \tag{25}
$$

$$
\mathrm{coag}_{M_{\mathrm{PL}}} = - \int_{-\infty}^{\infty} \int_{-\infty}^{\infty} m_{\mathrm{p}} \, \beta(D_{\mathrm{p}}, D_{\mathrm{p}}') \, \mathrm{d}N_{\mathrm{PL}} \, \mathrm{d}N_{\mathrm{LN}}' \tag{26}
$$

$$
\mathrm{coag}_{N_{\mathrm{LN}}} = -\frac{1}{2} \int_{-\infty}^{\infty} \int_{-\infty}^{\infty} \beta(D_{\mathrm{p}}, D_{\mathrm{p}}') \, \mathrm{d}N_{\mathrm{LN}} \, \mathrm{d}N_{\mathrm{LN}}' \tag{27}
$$

$$
\mathrm{coag}_{S_{\mathrm{LN}}} = -\frac{1}{2} \int_{-\infty}^{\infty} \int_{-\infty}^{\infty} \left[ 2s_{\mathrm{p}} - \left( s_{\mathrm{p}}^{\frac{3}{2}} + s_{\mathrm{p}}'^{\frac{3}{2}} \right)^{\frac{2}{3}} \right] \beta(D_{\mathrm{p}}, D_{\mathrm{p}}') \, \mathrm{d}N_{\mathrm{LN}} \, \mathrm{d}N_{\mathrm{LN}}'
$$

$$
+ \int_{-\infty}^{\infty} \int_{-\infty}^{\infty} \left[ \left( s_{\mathrm{p}}^{\frac{3}{2}} + s_{\mathrm{p}}'^{\frac{3}{2}} \right)^{\frac{2}{3}} - s_{\mathrm{p}}' \right] \beta(D_{\mathrm{p}}, D_{\mathrm{p}}') \, \mathrm{d}N_{\mathrm{PL}} \, \mathrm{d}N_{\mathrm{LN}}' \tag{28}
$$

$$
\mathrm{coag}_{M_{\mathrm{LN}}} = \int_{-\infty}^{\infty} \int_{-\infty}^{\infty} m_{\mathrm{p}} \, \beta(D_{\mathrm{p}}, D_{\mathrm{p}}') \, \mathrm{d}N_{\mathrm{PL}} \, \mathrm{d}N_{\mathrm{LN}}', \tag{29}
$$

where $\beta(D_{\mathrm{p}}, D_{\mathrm{p}}')$ is the coagulation coefficient of particles with the diameters of $D_{\mathrm{p}}$ and $D_{\mathrm{p}}'$ calculated with the equation

$$
\beta(D_{\mathrm{p}}, D_{\mathrm{p}}') = 2\pi (D_{\mathrm{p}} + D_{\mathrm{p}}')(\mathcal{D}_{\mathrm{p}} + \mathcal{D}_{\mathrm{p}}') f(\mathrm{Kn}_{\mathrm{coag}}), \tag{30}
$$

where $f(\mathrm{Kn_{coag}})$ is the transition regime function of Dahneke (1983)

$$f(\mathrm{Kn_{coag}}) = \frac{1 + \mathrm{Kn_{coag}}}{1 + 2\mathrm{Kn_{coag}} + 2\mathrm{Kn_{coag}^2}}, \tag{31}$$

where $\mathrm{Kn_{coag}}$ is the Knudsen number for coagulation

$$\mathrm{Kn_{coag}} = \frac{4(\mathcal{D}_\mathrm{p} + \mathcal{D}_\mathrm{p}')}{(D_\mathrm{p} + D_\mathrm{
[revised manuscript text omitted]
_{\mathrm{coag}}}^{\infty} \beta(D_{\mathrm{p}}, D_{\mathrm{p}}') \, \mathrm{d}N_{\mathrm{PL}} \, \mathrm{d}N_{\mathrm{PL}}' \tag{41}$$

$$\text{transfer}_{S_{\mathrm{PL}}} = -\text{transfer}_{S_{\mathrm{LN}}} = -\frac{1}{2} \int\limits_{-\infty}^{\infty} \int\limits_{\ln D_{\mathrm{coag}}}^{\infty} \left(s_{\mathrm{p}}^{\frac{3}{2}} + s_{\mathrm{p}}'^{\frac{3}{2}}\right)^{\frac{2}{3}} \beta(D_{\mathrm{p}}, D_{\mathrm{p}}') \, \mathrm{d}N_{\mathrm{PL}} \, \mathrm{d}N_{\mathrm{PL}}' \tag{42}$$

10   $$\text{transfer}_{M_{\mathrm{PL}}} = -\text{transfer}_{M_{\mathrm{LN}}} = -\frac{1}{2} \int\limits_{-\infty}^{\infty} \int\limits_{\ln D_{\mathrm{coag}}}^{\infty} \left(m_{\mathrm{p}} + m_{\mathrm{p}}'\right) \beta(D_{\mathrm{p}}, D_{\mathrm{p}}') \, \mathrm{d}N_{\mathrm{PL}} \, \mathrm{d}N_{\mathrm{
[revised manuscript text omitted]

---

## Author Response (AR2)

**Response to reviewers' comments of Olin et al.: "Using a combined power law and log-normal distribution model to simulate particle formation and growth in a mobile aerosol chamber"**

We thank the reviewers for their comments, and have corrected the manuscript according to them.

Referee reports are in *black italic* and authors' responses are in blue roman font. The changes to the manuscript provided as the marked-up manuscript and the changed manuscript are included at the end of this file.

**Referee 1 comments:**

*I suggest revising the following parts of the manuscript:*

*Page 7, lines 23-24, the following sentence is very unintuitive and should be revised: "In one-dimensional simulation, the mass growth rate can be considered a function of time and the particle diameter only, not of the spatial location, if the other parameters in Eq. (16) are considered also."*

The sentence is now replaced with "If all the parameters in Eq. (16) do not depend on the spatial location, as is the case in a one-dimensional simulation, the mass growth rate can be considered a function of time and the particle diameter only."

*Page 17, lines 3-5 reads: "The data measured after the event were used in the synchronization, using the assumption that the device outputs are equal in that moment, because particle sizes are well within the range of high detection efficiency of all three devices." I do not understand what is meant by "assumption that the device outputs are equal in that moment", please clarify the whole sentence.*

The sentence is now replaced with "Because particle sizes are well within the range of high detection efficiency of all three devices after the particle formation event, the device outputs would be equal in that moment if the maximum detection efficiencies of all the devices were equal. But because there are differences in the maximum detection efficiencies and the time responses of the devices, the output concentrations were multiplied and the output time vectors were synced so that all the time

series of the concentrations are overlapping after the event."

*Page 22, lines 6-7 reads: "The number of size sections in the MC10 model is obviously too low to plot the distributions well." I assume that the sentence tries to say that the correspondence between MC10 and FS400 is not good because of too few size sections used in the MC10 model. The sentence, however, is actually saying that because of the low number of sections in the MC 10 model, there were issues plotting the distribution, as if getting the purple line into the figure was somehow technically challenging.*

The sentence is now replaced with "The number of size sections in the MC10 model is obviously too low to obtain size distributions that are near the reference distributions."

**Referee 2 comments:**

*The authors have done an excellent job in revising their paper! I now can suggest publication in ACP. One thing to consider is to move also section 3.2.1 'Processing the experimental data' to the appendix. It was an important addition (and I was also asking for such details), but is not core material of the manuscript.*

The section is now moved to the Appendix. The moving caused also some reordering of the figures and the locations of the first descriptions of abbreviations.

**Additional corrections:**

There were minus signs missing in the equations of the coagulational and the depositional losses (Eqs. (34), (35), (38), and (39)). They are now added into those equations.

**Marked-up manuscript:**

[revised manuscript text omitted]

where $l_{bg}$ is the exponent depending on $CMD_{bg}$. The value of $l_{bg}$ ranges between -2 and -1 (Lehtinen et al., 2007). The coagulational loss term, e.g., for a number concentration is

$$\mathrm{loss}_{N_i}^{\mathrm{coag}} = \underline{N} - N_{bg} \int_{-\infty}^{\infty} \beta(D_{\mathrm{p}}, CMD_{bg}) \mathrm{d}N_i \approx \underline{N} - N_{bg} \beta(D_1, CMD_{bg}) D_1^{-l_{bg}} \int_{-\infty}^{\infty} D_{\mathrm{p}}^{l_{bg}} \mathrm{d}N_i, \tag{34}$$

in which the last integral can be solved analytically. The analytical solutions for the coagulational loss terms are

$$\mathrm{loss}_X^{\mathrm{coag}} = \underline{X} - X N_{bg} \cdot \begin{cases} \beta(D_1, CMD_{bg}) \left(\frac{\alpha}{\alpha + l_{bg}}\right) \left(\frac{d^{\alpha + l_{bg}} - 1}{d^\alpha - 1}\right), & X = N_{PL} \\[2mm] \beta(D_1, CMD_{bg}) \left(\frac{\alpha + 2}{\alpha + 2 + l_{bg}}\right) \left(\frac{d^{\alpha + 2 + l_{bg}} - 1}{d^{\alpha + 2} - 1}\right), & X = S_{PL} \\[2mm] \beta(D_1, CMD_{bg}) \left(\frac{\alpha + 3}{\alpha + 3 + l_{bg}}\right) \left(\frac{d^{\alpha + 3 + l_{bg}} - 1}{d^{\alpha + 3} - 1}\right), & X = M_{PL} \\[2mm] \beta(D_{\mathrm{g}}, CMD_{bg}) \exp\left[\frac{1}{2} l_{bg}^2 \ln^2 \sigma\right], & X = N_{LN} \\[2mm] \beta(D_{\mathrm{g}}, CMD_{bg}) \exp\left[\left(\frac{1}{2} l_{bg}^2 + 2 l_{bg}\right) \ln^2 \sigma\right], & X = S_{LN} \\[2mm] \beta(D_{\mathrm{g}}, CMD_{bg}) \exp\left[\left(\frac{1}{2} l_{bg}^2 + 3 l_{bg}\right) \ln^2 \sigma\right], & X = M_{
[revised manuscript text omitted]
_p, CMD_{bg}) \approx \beta(D_1, CMD_{bg}) \left( \frac{D_p}{D_1} \right)^{l_{bg}}, \tag{33}$$

where $l_{bg}$ is the exponent depending on $CMD_{bg}$. The value of $l_{bg}$ ranges between -2 and -1 (Lehtinen et al., 2007). The coagulational loss term, e.g., for a number concentration is

$$\text{loss}_{N_i}^{coag} = -N_{bg} \int_{-\infty}^{\infty} \beta(D_p, CMD_{bg}) dN_i \approx -N_{bg} \beta(D_1, CMD_{bg}) D_1^{-l_{bg}} \int_{-\infty}^{\infty} D_p^{l_{bg}} dN_i, \tag{34}$$

in which the last integral can be solved analytically. The analytical solutions for the coagulational loss terms are

$$\text{loss}_X^{coag} = -X N_{bg} \cdot \begin{cases} \beta(D_1, CMD_{bg}) \left( \frac{\alpha}{\alpha + l_{bg}} \right) \left( \frac{d^{\alpha + l_{bg}} - 1}{d^\alpha - 1} \right), & X = N_{PL} \\ \beta(D_1, CMD_{bg}) \left( \frac{\alpha + 2}{\alpha + 2 + l_{bg}} \right) \left( \frac{d^{\alpha + 2 + l_{bg}} - 1}{d^{\alpha + 2} - 1} \right), & X = S_{PL} \\ \beta(D_1, CMD_{bg}) \left( \frac{\alpha + 3}{\alpha + 3 + l_{bg}} \right) \left( \frac{d^{\alpha + 3 + l_{bg}} - 1}{d^{\alpha + 3} - 1} \right), & X = M_{PL} \\ \beta(D_g, CMD_{bg}) \exp\left[ \frac{1}{2} l_{bg}^2 \ln^2 \sigma \right], & X = N_{LN} \\ \beta(D_g, CMD_{bg}) \exp\left[ \left( \frac{1}{2} l_{bg}^2 + 2 l_{bg} \right) \ln^2 \sigma \right], & X = S_{LN} \\ \beta(D_g, CMD_{bg}) \exp\left[ \left( \frac{1}{2} l_{bg}^2 + 3 l_{bg} \right) \ln^2 \sigma \right], & X = M_{
[revised manuscript text omitted]
 | 0.1 | $1\,\text{nm}\,\text{h}^{-1}$ | 280 | intra, inter | 0 | – | – | 0 | 0 | 5 h |
| Atm2 | 0.1 | $1\,\text{nm}\,\text{h}^{-1}$ | 280 | intra, inter | 0 | – | – | $1.8\,\text{nm}\,\text{h}^{-1}$ | 0 | 5 h |
| Atm3 | 0.1 | $1\,\text{nm}\,\text{h}^{-1}$ | 280 | intra, inter, bg | $10^3$ | 100 | -1.6 | $1.8\,\text{nm}\,\text{h}^{-1}$ | 0 | 5 h |
| Atm4 | Eq. (50) | $1\,\text{nm}\,\text{h}^{-1}$ | 280 | intra, inter, bg | $10^3$ | 100 | -1.6 | $1.8\,\text{nm}\,\text{h}^{-1}$ | 0 | 5 h |
| Atm5 | Eq. (50) | Eqs. (16) – (18) | 280 | intra, inter, bg | $10^3$ | 100 | -1.6 | $1.8\,\text{nm}\,\text{h}^{-1}$ | 0.25 | 5 h |
| Exh | $10^8$ | $5\,\text{nm}\,\text{s}^{-1}$ | 500 | intra, inter, bg | $10^6$ | 60 | -1.5 | $0.07\,\text{nm}\,\text{s}^{-1}$ | 0 | 1 s |

[revised manuscript text omitted]